# A novel interface for cortical columnar neuromodulation with multipoint infrared neural stimulation

Feiyan Tian[1,2], Ying Zhang [1,2], Kenneth E. Schriver [1,3], Jia Ming Hu [1,3] ✉ &
Anna Wang Roe [1,2,3,4] ✉

Cutting edge advances in electrical visual cortical prosthetics have evoked perception of shapes, motion, and letters in the blind. Here, we present an alternative optical approach using pulsed infrared neural stimulation. To interface with dense arrays of cortical columns with submillimeter spatial precision, both linear array and 100-fiber bundle array optical fiber interfaces were devised. We deliver infrared stimulation through these arrays in anesthetized cat visual cortex and monitor effects by optical imaging in contralateral visual cortex. Infrared neural stimulation modulation of response to ongoing visual oriented gratings produce enhanced responses in orientation-matched domains and suppressed responses in non-matched domains, consistent with a known higher order integration mediated by callosal inputs. Controls include dynamically applied speeds, directions and patterns of multipoint stimulation. This provides groundwork for a distinct type of prosthetic targeted to maps of visual cortical columns.

Developing visual cortical prosthetics (VCPs) for the blind (and those with low vision and other vision deficits) has long been a driving force behind vision research. From the days of Dobelle[1], to current advanced studies in humans and monkeys, groundbreaking work has demonstrated that electrical stimulation delivered through multicontact arrays can elicit perception of shapes, motion, and letters[2,3]. Advances in electrode technology and stimulation methods have brought significant quantitative and qualitative advances to VCPs. With low to moderate stimulation currents, Utah arrays are now able to evoke perception of fine shapes and features[4,5]. Development of 'current steering' methods in humans demonstrated enhanced saliency of evoked percepts over static stimulation patterns[6]. Evidence from studies in monkeys shows that focal low-current electrical stimulation of the cortex can modulate naturally specific percepts. These include single-site stimulation of (1) direction domains in MT to modulate perception of the direction of a field of moving dots (Salzmann &

Newsome[7]), (2) disparity-tuned (near-to-far) sites in MT to bias depth perception (DeAngelis and colleagues[8]), and (3) presumed color sites in V1 to modulate the visibility of a small color spot in a background color matching task (Tehovnik[9], in humans, see Schmidt et al.[10]). Stimulation of face patches in temporal cortex led to non-specific distorted face percepts (non-specificity perhaps due to the higher electrical current range used 50–300 μA)[11]. Studies that have employed electrical[12–14], optogenetic[15,16], and infrared[17,18] neural stimulation to stimulate single cortical columns led to activation of local functionally matched orientation and color domain networks, further suggesting that single site stimulation impacts a larger network of functionally selective response. Together, these examples support the hypothesis that the lower stimulation intensities have minimal current spread, allowing selective stimulation of single cortical columns and their associated networks, which may lead to perceptual modulation related to the column-encoded feature.

[1]Department of Neurosurgery of the Second Affiliated Hospital, Interdisciplinary Institute of Neuroscience and Technology, School of Medicine, Zhejiang University, Hangzhou 310029, China. [2]Key Laboratory for Biomedical Engineering of Ministry of Education, College of Biomedical Engineering and Instrument Science, Zhejiang University, Hangzhou 310027, China. [3]MOE Frontier Science Center for Brain Science and Brain-machine Integration, School of Brain Science and Brain Medicine, Zhejiang University, Hangzhou 310012, China. [4]National Key Laboratory of Brain and Computer Intelligence, Zhejiang University, Hangzhou 310058, China. ✉e-mail: hujiaming@zju.edu.cn; annawang@zju.edu.cn

**Columnar organization of visual cortex.** The columnar organization of visual cortex has long been known and provides a beautiful roadmap for accessing different featural loci underlying perception. Here, the term 'columnar' is used to refer to the mesoscale functional domains revealed with electrophysiology[19], optical imaging[20], and fMRI[21]. In visual cortex of humans, monkeys, and cats, different cortical areas have distinct spatial layouts representing, for example, features of color, contour orientation, motion, and depth. Previous VCP studies have functionally characterized and stimulated single functional columns[22] and have recognized the importance of incorporating cortical circuitry in VCP design. With the development of ultrahigh field MRI and high-resolution imaging methods, it becomes feasible to obtain columnar maps for targeted stimulation (in humans ~500 μm[23–25]; monkeys ~200–300 μm[26], cat ~500 μm[27]). Another important aspect of columnar organization is that, from area to area, information from 'lower order' columns is integrated into 'higher order' columns in downstream areas, thereby establishing de novo parameter spaces. A columnar VCP may be capable of targeting desired sets of columns, ideally with knowledge of the downstream effects. Here, we conduct a proof of principle demonstration of a column-targeted approach and show that higher-order representations can be predictably activated by columnar stimulation.

**Technological approach.** To enable selective stimulation of sets of cortical columns, our approach is to employ a non-penetrating optical stimulation method (termed infrared neural stimulation, INS, 1875 ± 10 nm)[28] delivered through an array of multiple optical fibers[29]. Optical arrays can be designed to match the size and density of cortical columns, can be placed on cortical windows in different positions, and can be removed when not in use. INS stimulates brain tissue via brief infrared laser pulses causing brief localized heating, which induces membrane capacitance change and subsequent neuronal activation[30]. Within the parameters used, it is non-damaging to tissue[28,31,32] and requires no prior viral transfection, making it amenable for use in cats, primates, and humans[17,18,33,34] (Supplementary Method). Applications in humans are in trial stage[33] (clinical trials for peripheral nerve stimulation and cochlear stimulation: Jansen NCT04601337, Richter NCT05110183). Similar to electrical[5,35] and optogenetic[36,37] stimulation in nonhuman primates, application of INS to primate visual cortex elicits phosphenes to which the animal saccades[28]. Using fiber optics of appropriate size (200 μm-diameter core), we have shown that stimulation leads to focal activation of functionally specific columns and their associated columnar networks within V1[14,15,17,38]. That is, stimulation of single columns leads to enhancement of nearby columns. However, unknown is whether stimulation of multiple columns in an area can lead to higher order response in another area.

**Higher order orientation in cat area 18.** Previous studies have shown that, similar to area V2 in macaque monkeys, cat area 18 is responsive to illusory contours (Fig. 1A)[39] and that one of the sources of illusory contour response is the callosal connections[40] (Fig. 1B, upper schematic). Here, we have designed two fiber optic interfaces (a 5-fiber and a 10 × 10 fiber array) to target cortical columns in cat visual area 18 and have used optical imaging to monitor the contralateral cortex in response to two orientations of linear fiber stimulation (Fig. 1B, bottom). The fiber optic interface is apposed to cortex without penetration and can be moved to different locations on the brain. We show that this interface enables focal, column-based INS stimulation and activates predicted higher-order orientation-selective effects in contralateral area 18. This demonstrates an interface that can activate native columnar organizations and their associated circuits, and provides promise for a targeted, column-based VCP.

## Results
### Hypothesis
Our stimulation paradigm derives from psychophysical studies in which adding a visual cue can accentuate or reduce the perception of a

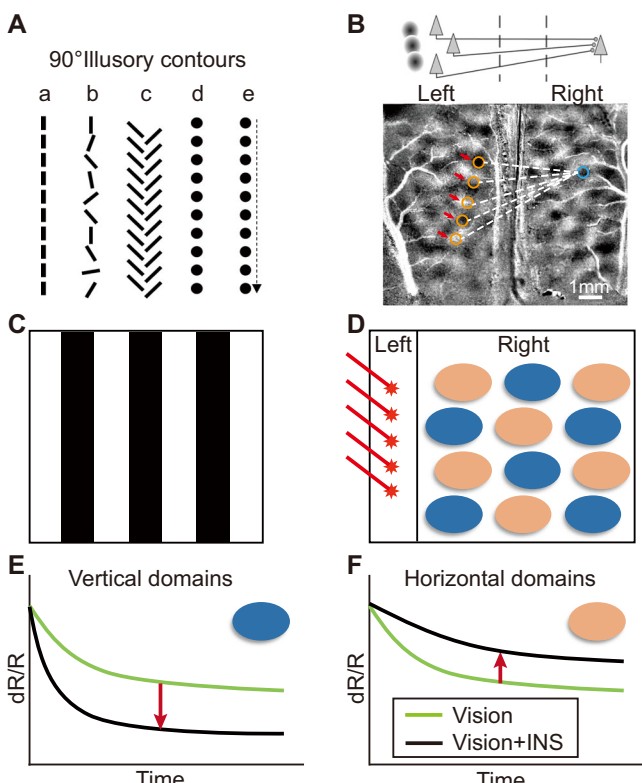

**Fig. 1 | Experimental scheme. A** Perception of 90° contours induced by higher order illusory contours. a. single orientation b. mixed orientation c. abutting line illusory contour d. aligned points e. sequential aligned points. **B** Upper: Schematic of 'high order projection' via callosal connection[22]. Below: Predicted activation in right hemisphere following stimulation of multiple aligned domains in left hemisphere. **C** Example of vertical grating or horizontal grating for visual stimulation. **D** "Vertical" INS applied at five points parallel to the vertical meridian in left hemisphere. Orientation-selective domains in right hemisphere mapped using optical imaging (OI); blue ovals: vertical domains, orange ovals: horizontal domains. **E, F** Schematic of expected OI response. Green lines: OI response curves for visual stimulation alone; black lines: response for vision + INS. Vertical INS added to visual stimulation **E** enhances grating response of vertical domains (downward red arrow), and **F** suppresses grating response of horizontal domains (upward red arrow).

scene (e.g., cross-orientation suppression[41]). Based on previous studies[42], we predict that adding a 'vertical' cue will bias the cortical response towards vertical and away from horizontal. We had previously shown that stimulating a single orientation domain leads to the enhancement of connected 'like' orientation domains and relative suppression of 'orthogonal' orientation domains[14,15]. However, whether stimulating multiple sites can induce higher-order responses is unknown.

Our testbed for this question is area 17 and 18 of cat visual cortex, areas whose topographic organization, functional orientation domain organization, and callosal connectivity are well known[40,43–46]. Callosal projections arise from cells near the 17/18 border and axons terminate broadly in multiple cortical columns over several square millimeters of contralateral cortex[45]. These non-topographic projections also project in a bilaminar pattern avoiding layer 4, supporting an interaction that generates higher order response[40]. Pulsed INS stimulation placed at the cortical surface is known to effectively activate the superficial layers[17,18] where callosally projecting neurons reside[45]. To address the modulatory effect of array stimulation, we tested whether stimulation of linear points on the cortex (similar to illusory contour induced by aligned dots) would evoke differential effects on domains with matching vs non-matching orientation preference. As illustrated in

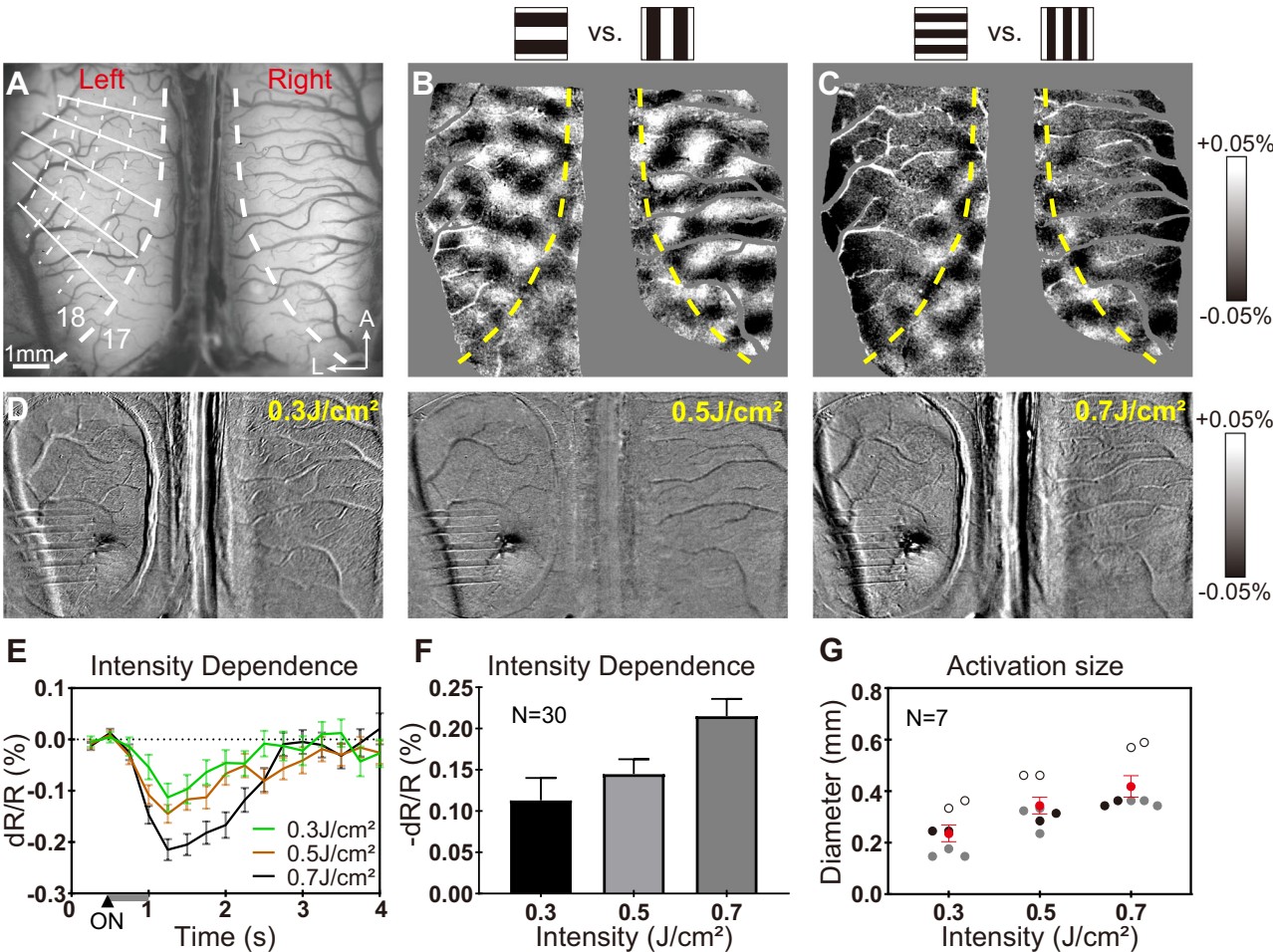

**Fig. 2 | Procedure for determining the 17/18 border and intensity dependence of single-fiber stimulation. A** Vessel map of the cortex in the optical window, area 18 in both left and right brain are exposed and imaged at the same time. Red dashed line: midline; thick white dashed line: vertical meridian (VM); thin white dashed lines: iso-polar lines; thin white solid lines: iso-eccentricity lines. **B** OI response map of 'H' – 'V' visual stimulation with low spatial and high temporal (0.2 cyc per deg, 5 Hz) frequency showing activation of area 18. **C** OI response map of 'H' – 'V' visual stimulation with high spatial and low temporal (0.5 cyc per deg, 2 Hz) frequency showing activation of area 17. Yellow dashed lines mark the 17/18 border (VM). 20 trials per image. Optical imaging field of view: 10.5 mm*8.7 mm; scale bar in **A**: 1 mm, A = anterior. Data from Cat 1. **D** Intensity dependence via a single fiber with radiant exposure from 0.3–0.7 J cm$^{-2}$. **E, F** Time course **E**, amplitude **F** of cortical response to INS stimulation in **D**. Gray bar indicates the duration of INS. $N = 30$ trials in **E, F**, data are presented as mean values ±SEM. **G** Diameter of the activation areas from 7 sessions of 3 cases. Black/gray/white circles: result from different cases. Red circles: average diameter across 3 cases (0.3 J cm$^{-2}$: 0.24 mm, 0.5 J cm$^{-2}$: 0.35 mm, 0.7 J cm$^{-2}$: 0.42 mm). $N = 7$ sessions. Data are presented as mean values ±SEM.

Fig. 1C, D, visual presentation of a vertical grating evokes a response (Fig. 1E, green line, a negative reflectance change detected by intrinsic signal imaging, see Methods) in vertical orientation domains (Fig. 1D, blue domains), while the horizontal grating evokes a response (Fig. 1F, green line) in horizontal orientation domains (Fig. 1D, peach domains). We predicted that adding a line of INS stimulation (Fig. 1D, red asterisks) parallel to the vertical meridian (VM, which is represented at the 17/18 border) will enhance the response of vertical domains (Fig. 1E, black line, downward red arrow indicates increased reflectance change over vision alone), and will suppress the horizontal domain response (Fig. 1F, black line, upward red arrow, reduction of reflectance change).

### Mapping 17/18 border, focal stimulation, and intensity dependence

We begin by identifying the border between areas 17 (cat V1) and 18 (cat V2) bilaterally (Fig. 2A–C). Optical imaging (OI) establishes orientation maps in response to drifting gratings (0° vs. 90°) in area 17 (prefers higher spatial frequency and lower temporal frequency gratings) and area 18 (prefers lower spatial frequency and higher temporal frequency gratings) on both sides (Fig. 2B, C)[47,48]. From this, the border between areas 17 and 18 is determined[47] (yellow dashed line), which is a

1–1.5-mm-wide zone of cytoarchitectonic transition rather than a sharp border[49]. The border represents the vertical meridian (VM) and guides the placement of the optical fiber array in a 'vertical' orientation (parallel to the VM) or in a 'horizontal' orientation (roughly orthogonal to the VM).

We also demonstrate that INS stimulation achieves focal submillimeter-sized activations and intensity-dependent activations. As shown in Fig. 2D, single-fiber INS stimulation (from 0.3–0.7 J cm$^{-2}$) applied to the surface of cat visual cortex induces an intensity-dependent response (Fig. 2E, F). The mean activation sizes for radiant exposures 0.3 J cm$^{-2}$, 0.5 J cm$^{-2}$, 0.7 J cm$^{-2}$ are 240 μm, 350 μm, and 420 μm respectively (Fig. 2G, red dots). This is well within the ~500 μm size of cat orientation domains and indicates that INS can stimulate orientation domains selectively[17].

### Single and multi-fiber stimulation: modulation of contralateral orientation networks

We then asked whether INS coupled with visually evoked activity can be used to modulate cortical responses. This would potentially make INS an important and useful tool for modulating cortical response under natural sensory and behavioral conditions. Our experimental

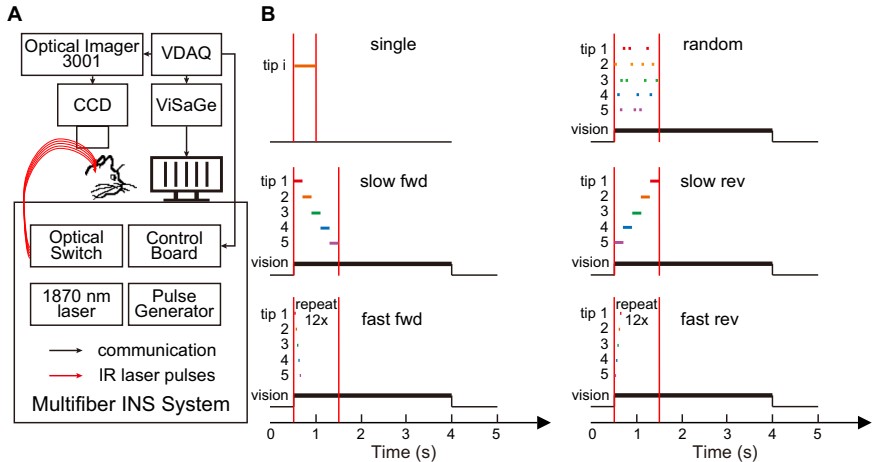

**Fig. 3 | Experimental schematic and pulse sequence paradigms. A** Schematic diagram of system combining OI, visual stimulation, and INS. See methods. **B** Time course for six INS conditions. single: 500 ms pulse train applied to any one optical fiber; random: 8 ms per fiber in random sequence for 1 s total duration; slow fwd: 200 ms per fiber in sequence (1→5) for one cycle, 1.0 s duration; slow rev: 200 ms per fiber in reverse sequence (5→1) for one cycle, 1.0 s duration; fast fwd: 16 ms per

fiber in sequence (1→5) for twelve cycles, 0.96 s duration; fast rev: 16 ms per fiber in reverse sequence (5→1) for twelve cycles, 0.96 s duration. All INS utilized pulse trains with pulse width of 250 μs at 200 Hz; INS initiated coincident with visual stimulation 0.5 s after the onset of OI data acquisition. OI acquisition 0 to 5 s. Black line: visual stimulation ON. Vertical red lines: INS period.

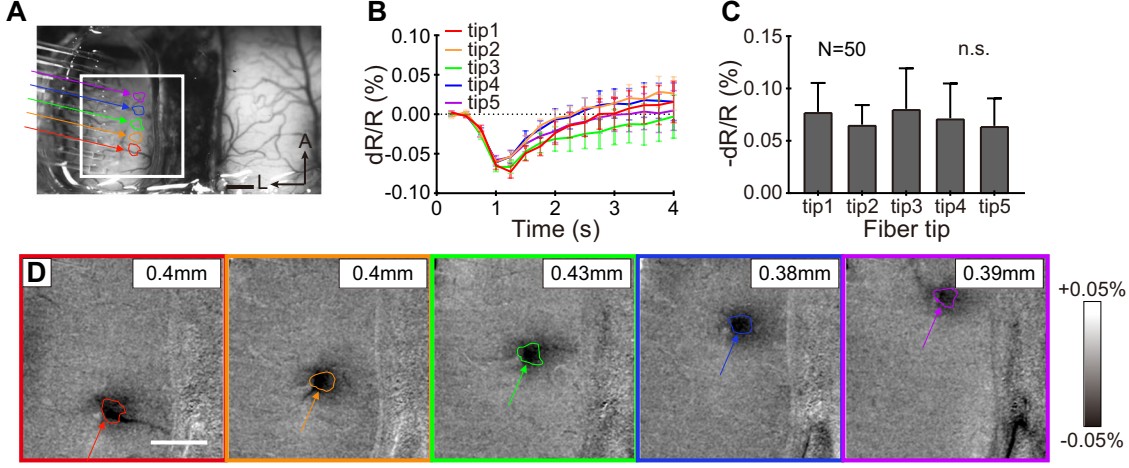

**Fig. 4 | Single-fiber INS activation in cat area 18. A** Linear nine-fiber array placed on left hemisphere; scale bar: 1 mm; A anterior, L left. Single-pulse INS applied at five of nine fibers in the array. Colored arrows mark tips of the five fibers used (tips 1–5 from posterior to anterior). Colored ROIs mark areas of significant responses evoked by INS. **B** Time course of OI signals in response to 0.3 J cm⁻² single-fiber INS for each ROI marked in **A**. Gray bar indicates the duration of INS. Error bar: SEM. **C** Peak magnitude of OI signals plotted

in **B**; calculated p values (p = 0.62) obtained from one-way ANOVA show no statistically significant difference across all five fibers. N = 50 trials in (B-C), data are presented as mean values ±SEM. **D** OI activation maps in response to INS show similar-sized focal activation areas. The diameter of activation size is within 500 μm (Top right corner, activation region: $p < 0.001$). INS: 1870 nm, 200 Hz, 0.3 J cm⁻², pulse width 250 μs, 50 trials. Scale bars in **A**, **D**: 1 mm. Data from Cat 2.

configuration combining OI, visual stimulation, and multi-fiber INS is shown schematically in Fig. 3A. Experimental conditions are shown in Fig. 3B.

Single-Fiber Stimulation (Fig. 3B, Single): We first investigated whether each single optical fiber evoked a local response. Although each fiber is calibrated at the beginning of the experiment to deliver the same radiant exposure (J cm⁻²), we further confirmed this by assessing the similarity of cortical response (OI spatial size and amplitude of reflectance change) across fibers. Figure 4A highlights five fibers evaluated in one experiment (colored arrows). The results of INS at each fiber tip at 0.3 J cm⁻² (without visual stimulation) are shown in Fig. 4D (INS activation maps). The activations were focal, sub-millimeter in size (diameter within 0.5 mm), and similar to activations evoked by visual spot stimulations (Supplementary Fig. 4)[50]. Moreover,

the locations of the activations shifted in topographic sequence, similar to the sequence of activations induced by a moving dot[51,52]. Thus, the activations of single fibers within the array were focal and topographic, indicating the suitability of using single fibers in sequence to mimic a moving point stimulus. The time course curves of the reflectance change for each fiber are shown in Fig. 4B. These are typical INS induced time course curves, peaking 1–1.5 s following stimulation onset followed by a gradual return to baseline. Figure 4C shows the amplitudes of OI response are similar (mean dR/R = −0.07%) across fibers. These single-fiber tests were conducted at the beginning of each experiment to confirm that comparable stimulations were delivered by the fibers in the array.

Sequential Multi-Fiber Stimulation (Fig. 3B, Slow Fwd, Slow Rev, Fast Fwd, Fast Rev, Random): Previous studies have shown that a

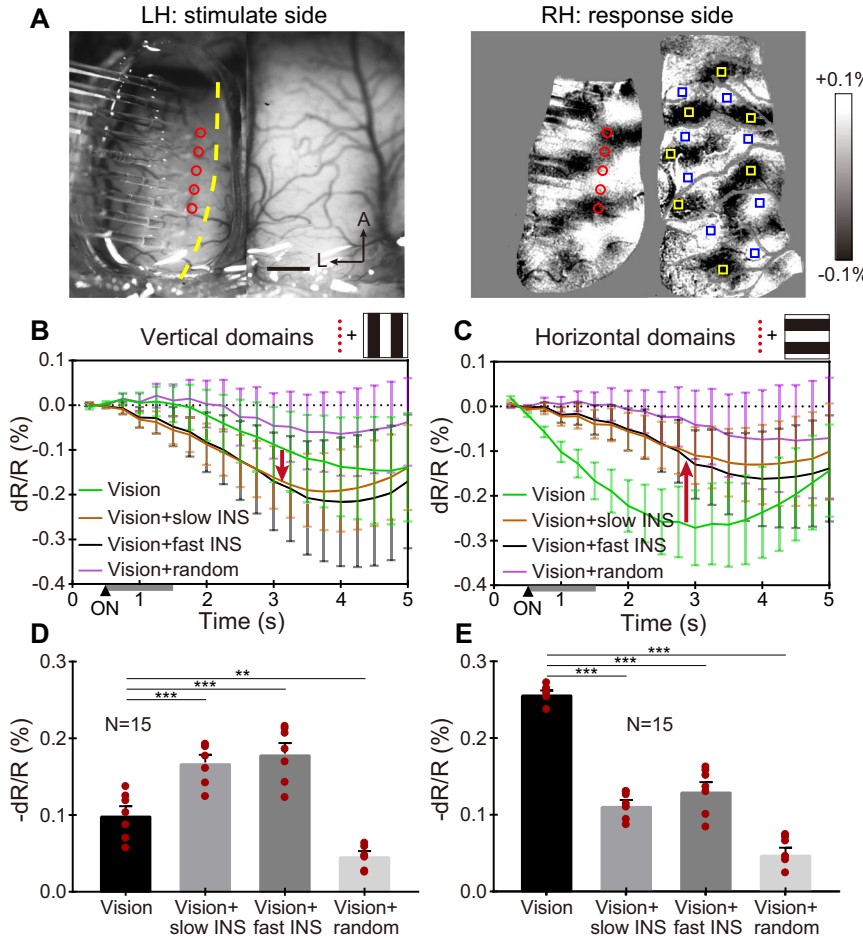

**Fig. 5 | Orientation-selective response: vertically oriented INS fiber array (referred as 'vertical INS') in area 18 of left hemisphere. A** Left panel: blood vessel map with nine-fiber linear array. Red circles: location of five fiber tips used. Right panel: orientation map generated by visual stimulation. Yellow boxes: center of ROIs in horizontal domains; blue boxes: center of ROIs in vertical domains. **B, C** Time course of OI signals for four different visual/INS stimuli combinations in vertical **B** and horizontal **C** domains. Gray bars from 0.5–1.5 s: INS ON. Red arrows indicate that both slow (brown) and fast (black) vertical INS enhance visually evoked OI signals for vertical domains (**B**, down arrow) and suppress OI signals in horizontal domains (**C**, up arrow). Random condition (magenta) reduces OI response in both vertical **B** and horizontal domains **C**. Error bar: SEM. **D, E** Averaged reflectance changes during 2.5–5 s of the time courses in **B, C**. ***$P < 0.001$, **$P = 0.002$ (two-sided Welch's $t$ test, vision vs. vision + slow/fast/random INS). $N = 15$ trials in **B–E**, data are presented as mean values ±SEM, with individual data superimposed (red dots). INS: 1870 nm, 200 Hz, 0.3 J cm$^{-2}$, pulse width 250 µs, pulse train 1 s. Scale bar in **A**: 1 mm, A anterior, L left. Data from Cat 2.

linearly moving dot can evoke the sensation of an oriented contour[53]. To mimic this, we delivered INS across five collinear points in area 18 (at 0.3 J cm$^{-2}$); the effect of two speeds and two directions on oriented visual stimulation was examined. As shown in Fig. 1, we expected to see enhancement of response in orientation-selective domains matched to that of the fiber array orientation, and reduction in response of orientation-selective domains orthogonal to the fiber array orientation. We also investigated whether the direction of the INS sequence affected the response of the orientation-selective domains.

In another case (Cat 2, Fig. 5), we combined sequential 'vertical' array stimulation (Fig. 5A, array parallel to VM in cat area 18), with vertical (Fig. 5B) and horizontal (Fig. 5C) visual grating stimulation. Orientation maps derived from each of these combined 'vision+INS' stimulations were compared to 'vision only' orientation maps. In addition, the vertical array conditions comprised two directions and two speeds (slow: 2 mm cortical distance/s, fast: 24 mm/s; similar to a dot moving at 3 deg/s and ~36 deg/s, respectively). As a control, a non-directional 'random' array stimulus (which induces a much weaker vertical percept) was added (see Fig. 3B). All 13 conditions were randomly interleaved within each trial and each trial was repeated 15 times.

Figure 5B, C show, for each condition, the time course of the intrinsic signal reflectance changes for vertical and horizontal orientation-selective domains, respectively. All conditions show responses that peak around 2.5–3.5 s after stimulation onset, with peak magnitudes ranging from 0.1 to 0.3%, typical of OI signals. Figure 5B shows that vertical visual stimulation alone (green trace) produced a peak response amplitude of -0.15%, and that adding vertical INS (as shown in Fig. 5A) to visual vertical stimulation resulted in an enhanced (-0.05%, downward red arrow) response in vertical domains (Fig. 5B, brown: slow, black: fast). In contrast, adding vertical INS to visual horizontal stimulation led to a reduction (-0.1%, upward red arrow) in response (Fig. 5C, brown: slow, black: fast). This differential effect is quantified in Fig. 5D (vertical domains, Welch's $t$ test, $p < 0.001$ for all comparisons, except vision alone vs. vision + random INS ($p < 0.01$)) and 5E (horizontal domains, Welch's $t$ test, $p < 0.001$ for all comparisons). The effect of sequential INS stimulation in a single direction also produced similar orientation-matched enhancement and non-matched reduction of the same trend (Supplementary Fig. 1). No significant difference was observed between slow and fast INS. The addition of random INS stimulation to visual stimulation resulted in reduced response to both vertical (-0.05%) and horizontal (-0.15%)

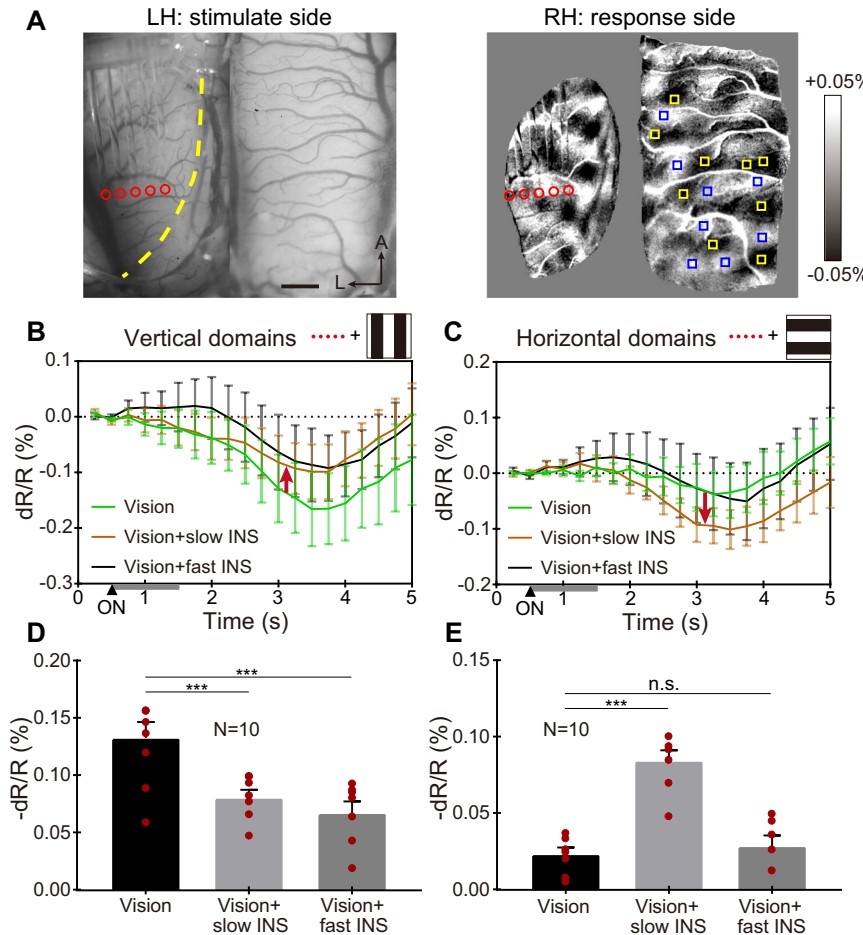

**Fig. 6 | Orientation-selective response: horizontally oriented INS fiber array (referred as 'horizontal INS') in area 18 of left hemisphere. A** Left panel: blood vessel map with the five-fiber linear array. Red circles: location of five fiber tips used. Right panel: orientation map generated by visual stimulation. Yellow boxes: center of ROIs in horizontal domains; blue boxes: center of ROIs in vertical domains. **B, C** Time course of OI signals for three different visual/INS stimuli combinations in vertical **B** and horizontal **C** domains. Gray bars from 0.5–1.5 s: INS ON. Red arrows indicate that both slow (brown) and fast (black) horizontal INS suppresses visually evoked OI signals for vertical domains (**B**, up arrow) and slow horizontal INS enhances OI signals in horizontal domains (**C**, down arrow). Error bar: SEM. **D, E** Averaged reflectance changes during 2.5–5 s of the time courses in **B, C**. ***$P < 0.001$; n.s.: $P = 0.54$ (two-sided Welch's $t$ test, vision vs. vision + slow/fast INS). $N = 10$ trials in **B–E**, data are presented as mean values ±SEM, with individual data superimposed (red dots). INS: 1870 nm, 200 Hz, 0.3 J cm$^{-2}$, pulse width 250 μs, pulse train 1 s. Scale bar in **A**: 1 mm, A anterior, L left. Data from Cat 3.

orientation domains (Fig. 5B, C magenta; Fig. 5D, E last column), suggesting that the scattered incoherent flashes interfere with contour orientation response. We also note that fiber tip locations spanned multiple different orientation domains (Fig. 5A, from anterior to posterior, estimated to be 90°,0°,135°,100°,20°), so the contour orientation effect is not a result of INS stimulation of vertical orientation domains in the left hemisphere, but rather, we infer, a result of topographic multipoint integration.

In yet another case (Cat 3, Fig. 6), we placed the linear fiber array approximately orthogonal to the VM ("horizontal INS"), as shown in Fig. 6A. Eleven of the 13 conditions in case 1 were investigated (random stimulation was not used in this experiment); each trial was repeated 10 times. Here, the results largely mirrored those observed in Cat 2. Sequential horizontal INS suppressed OI response in the vertical orientation-selective domains at both fast and slow speeds (Fig. 6B, black and brown traces). The response of horizontal orientation-selective domains was enhanced by slow INS stimulation (Fig. 6C, brown trace), but fast INS stimulation resulted in no significant change (Fig. 6C, black trace) compared to visual stimulation alone (Fig. 6C, green trace). The effect is further quantified in Fig. 6D (vertical domains, Welch's $t$ test, $p < 0.001$) and 6E (horizontal domains, Welch's $t$ test, $p < 0.001$ between vision alone and vision + slow INS).

We also examined the effects of higher stimulation intensity. Previous studies have shown that increasing stimulation intensity can bias the excitatory/inhibitory balance such that a general suppression is invoked[54]. Our results of adding a higher intensity INS stimulation (0.7 J cm$^{-2}$) to visual stimulation revealed relative suppression of all conditions. These results are shown in Fig. 7. All experimental parameters were the same as described above, except that the radiant energy was increased to 0.7 J cm$^{-2}$. Under these higher fluence conditions, we observed a reduction in the OI reflectance signal irrespective of the orientation of the fiber optic array, the direction of INS stimulation, or the speed of INS stimulation. Suppressive effects ranged from -0.05% to -0.1%, and the effect is further quantified in Fig. 7 (Welch's $t$ test).

To increase the number and flexibility of the stimulation patterns, we designed and constructed a 100-channel (10 × 10 array) fiber bundle. The fibers have a 200-μm-diameter core and 10 μm cladding, resulting in close-packed spacing of 220 μm (see Fig. 8C, inset). As Fig. 8 shows, the fiber bundle was apposed to the cortex on one hemisphere (Fig. 8A, red circle), where we had acquired an orientation map (Fig. 8B), and optical imaging was conducted over the contralateral cortex. We selected two perpendicular rows of fibers in the array (5 per row, every other fiber, spacing 440 μm apart, one vertical

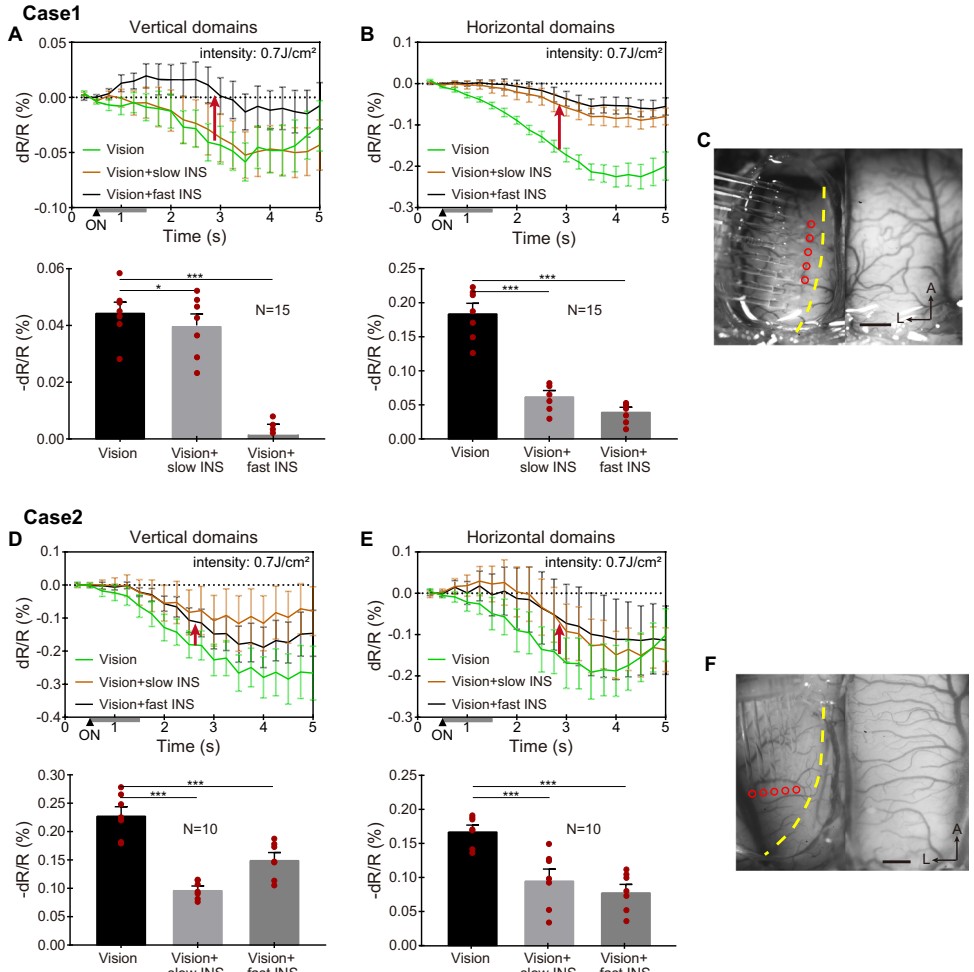

**Fig. 7 | Reduction of OI response at high INS intensity. A, B** Upper row is the time course of OI signal for three vision / INS stimulation conditions, measures at ROIs in vertical **A** and horizontal **B** domains shown in Fig. 5A in manuscript. Bottom row is the average reflectance changes during 2.5–5 s of the time courses in the upper row. ***$P < 0.001$, *$P = 0.04$ (two-sided Welch's $t$ test, vision vs. vision + slow/fast INS). $N = 15$ trials in **A–B**, data are presented as mean values ±SEM, with individual data superimposed (red dots). **C** The location and direction of fiber array placement in case 1. **D, E** Upper row is the time course of OI signal for three vision/INS stimulation conditions (vertical, **D** vs. Horizontal, **E** domains shown in Fig. 5A in manuscript). Bottom row is the average reflectance changes during 2.5–5 s of the time courses in the upper row. ***$P < 0.001$ (two-sided Welch's $t$ test, vision vs. vision + slow/fast INS). $N = 10$ trials in **D, E**, data are presented as mean values ±SEM, with individual data superimposed (red dots). **F** The location and orientation of the fiber array placement in case 2. Gray bars from 0.5–1.5 s: laser ON. INS: 1870 nm, 200 Hz, 0.7 J cm⁻², pulse width 250 μs, pulse train 1 s. Scale bars in **C, F**: 1 mm, A anterior, L left. Data from Cat 2 & 3.

and one horizontal, Fig. 8B) for delivery of INS in combination with the presentation of the visual grating. This enabled stimulation of the cortex in two orientations without moving the array.

We found results similar to those from the linear arrays. That is, we found that on the contralateral visual cortex (Fig. 8D, yellow: horizontal domains, blue: vertical domains), "horizontal INS" stimulation (orthogonal to VM, red dots drawn in Fig. 8B) suppressed the response of vertical orientation-selective domains to vertical gratings (Fig. 8E) and relatively enhanced the horizontal orientation domains response to horizontal gratings (Fig. 8G). The effect is further quantified in Fig. 8F (vertical domains, Welch's $t$ test, $p < 0.001$) and 8H (horizontal domains, Welch's $t$ test, $p < 0.001$). In another case, we conducted a similar preparation as in Fig. 8 (Fig. 9A–D) and compared the cortical modulation effects of two orthogonal INS stimulation patterns with the 100-channel fiber bundle. On the contralateral visual cortex (Fig. 9D, yellow: horizontal domains, blue: vertical domains), "vertical INS" stimulation (parallel to VM, see Fig. 9B) enhanced the response of vertical orientation-selective domains to vertical gratings (Fig. 9E) and relatively suppressed the horizontal orientation domains response to horizontal gratings (Fig. 9G). The effect is further quantified in Fig. 9F (vertical domains, Welch's $t$ test, $p < 0.001$) and 9H (horizontal

domains, Welch's $t$ test, $p < 0.001$). Similarly, on the contralateral visual cortex, "horizontal INS" (orthogonal to VM, see Fig. 9B) suppresses the vertical orientation-selective domains to vertical gratings (Fig. 9I) and enhances the horizontal orientation-selective domains to horizontal gratings (Fig. 9K). The effect is further quantified in Fig. 9J (vertical domains, Welch's $t$ test, $p < 0.001$) and 8K (horizontal domains, Welch's $t$ test, $p < 0.001$). These results further confirm the higher-order orientation-selective effects of INS stimulation and, in addition, demonstrate the delivery of different patterns of stimulation without moving the fiber bundle.

## Discussion

**Summary of key aspects of this visual brain-machine interface**
**Columnar stimulation.** As assessed by OI of cortical response, we found that single-fiber activation was confined to single <0.5 mm loci (Figs. 2 and 4), well within the size of cat orientation columns[27]. Because feature parameters are represented at mesoscale and are intermixed locally (e.g., different orientation columns within a 1–2 mm-sized region), stimulations of higher intensity can lead to activation of greater than 1 mm loci; this would activate multiple types of feature nodes near the stimulation tip, leading to a non-specific 'phosphene-

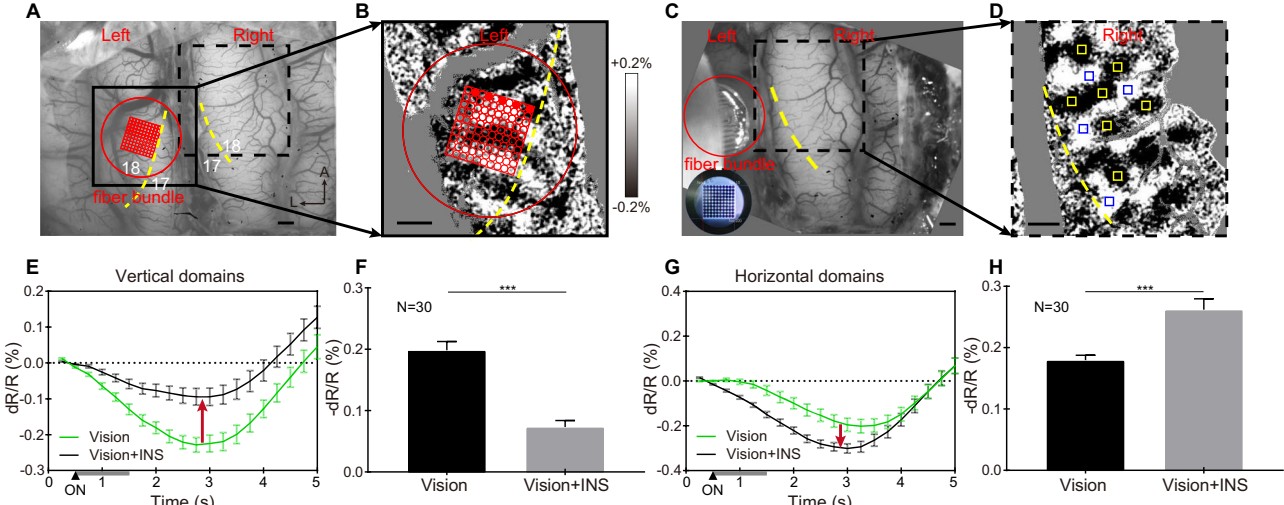

**Fig. 8 | Orientation-selective response of INS stimulation via fiber bundle array.**
**A** vessel map of both hemispheres, red circle, and grid indicate the bundle location with 100 channels (10 × 10) in the left hemisphere. **B** Orientation map generated by visual stimulation, enlarged view of the black box in **A**. Red dots: working channels in this experiment. **C** The actual location of the fiber bundle interface is shown in **A**. inset: photo of 100-channel fiber bundle face. **D** The orientation map generated by visual stimulation, an enlarged view of the black dashed box in **A** and **C**, indicates the imaging area in the right hemisphere. Blue squares: ROIs of vertical domains analyzed in **E**. Red squares: ROIs of horizontal domains analyzed in **G**. Yellow dashed lines in **A**–**D**: 17/18 borders (VM). Scale bar in **A**–**D**: 1 mm. A anterior, L lateral. **E**, **G** Time course of OI signals for 'Vision' (green line) and 'Vision+INS' (black line) stimuli in vertical **E** and horizontal **G** domains. Gray bars at bottom of graph from 0.5–1.5 s: INS ON. Red arrows: suppressive (up arrow) or enhancing (down arrow) effect of adding INS to ongoing visual grating stimulation. See text. **F**, **H** Averaged reflectance changes during 2.5–5 s of the time courses in **E**, **G**. \*\*\*$P < 0.001$ (two-sided Welch's $t$ test, vision vs. vision + INS). $N = 30$ trials in **E**–**H**, data are presented as mean values ±SEM. INS: 1870 nm, 200 Hz, 0.3 J cm$^{-2}$, pulse width 250 µs, pulse train 1 s. Data from Cat 4.

like' experience. As we have demonstrated here, our optical stimulation approach can provide spatially specific effects (diameter of activation area less than 500 µm), fulfilling the requirements for selectively interfacing with column-based features[29]. With lower stimulation intensities, it is feasible to reach even smaller activation sizes (Fig. 2G).

**Neuromodulation.** Inducing a percept by adding external stimulation to an ongoing elevated baseline can have enhancing or inhibiting effects[55]. Here, we have combined a linear arrangement of stimulation points to ongoing visual-oriented gratings, demonstrating an enhancement when the stimulation effect is matched to the orientation of the visual stimulus and a relative suppression when not matched. These orientation-specific effects were seen with both 'slow' and 'fast' array stimulation (Fig. 5; except at high INS intensities in Fig. 7), consistent with the breadth of velocity preferences in cat area 18[56]. Orthogonal INS array stimulation resulted in symmetrically opposite effects, as predicted (Figs. 6 and 9). Random sequences of INS stimulation, which do not induce a salient 'moving dot' percept failed to induce this oriented effect and in fact led to a general reduction of signal (Fig. 5). Previous studies of underlying intracortical circuitry have shown stimulation of single orientation domains in cat area 18 lead to relative enhancement (suppression) of connected matched (non-matched) orientation domains in area 18[14]. For callosal circuits, as Makarov 2008 showed[40], there are circuits in the ferret brain (which is similar to the cat brain[57]) that underlie this type of orientation-selective cross-callosal integration. They concluded that the schematic (shown in Fig. 1B, upper) is the likely circuit. Our demonstration that a VCP can induce the same integrated orientation-selective (enhancement: matched, relative suppression: non-matched) response suggests that it activates the same circuits. Thus, our data suggest that INS applied to cortical columns can predictably access the circuits underlying higher-order orientation response.

**Dynamics.** The ideal brain-machine interface for enhancing vision would be able to deliver time-varying stimuli for eliciting distinct perceptual effects. Here, using a 5-fiber array, we presented 10 stimulation paradigms: stimulation of single fibers, sequential five-fiber stimulation in two directions and at two speeds, and random (non-directional) five-fiber stimulation. The 10 × 10 fiber bundle array illustrates that different spatial patterns of stimulation can be delivered from a two-dimensional array. Similar capabilities have been developed in other studies using electrical stimulation in humans and monkeys[2,6] and optogenetic stimulation in mice[29].

**Higher order response.** For each oriented fiber array stimulation on the cortex, the fiber tip locations targeted different orientation domains in one hemisphere (Figs. 5 and 6) and achieved selective activation in the contralateral hemisphere. Random sequences did not produce this result. We showed that the contralateral orientation-specific response, despite receiving inputs from multiple different orientations, resulted in topographic multipoint integration. Previous VCP studies have induced oriented and motion[58] percepts from multisite stimulation[39,42,59–62]. Here, we show that a VCP induces activation of predicted inter-areal circuits leading to higher order response, providing a link between neural circuits and previously demonstrated higher order percepts evoked. Future application of INS stimulation in behaving monkeys will further evaluate this relationship. This demonstration lends confidence to the ability of external devices to activate existing brain circuits in a predictable manner, something that may impact the future design of VCPs.

**Linking the contralateral visual hemifield.** This study highlights an approach for linking the two visual hemifields[63]. We find that not only are their responses induced in the contralateral hemisphere, but they are integrated into orientation-domain specific effects. Callosal connections are important for integrating the left and right visual fields (e.g., for reading[64]), tying together the two representations of the vertical meridian[65–67], and the fovea (which is represented on both hemispheres[68]). These aspects of vision and visually guided behaviors are also challenges that must be addressed by VCPs.

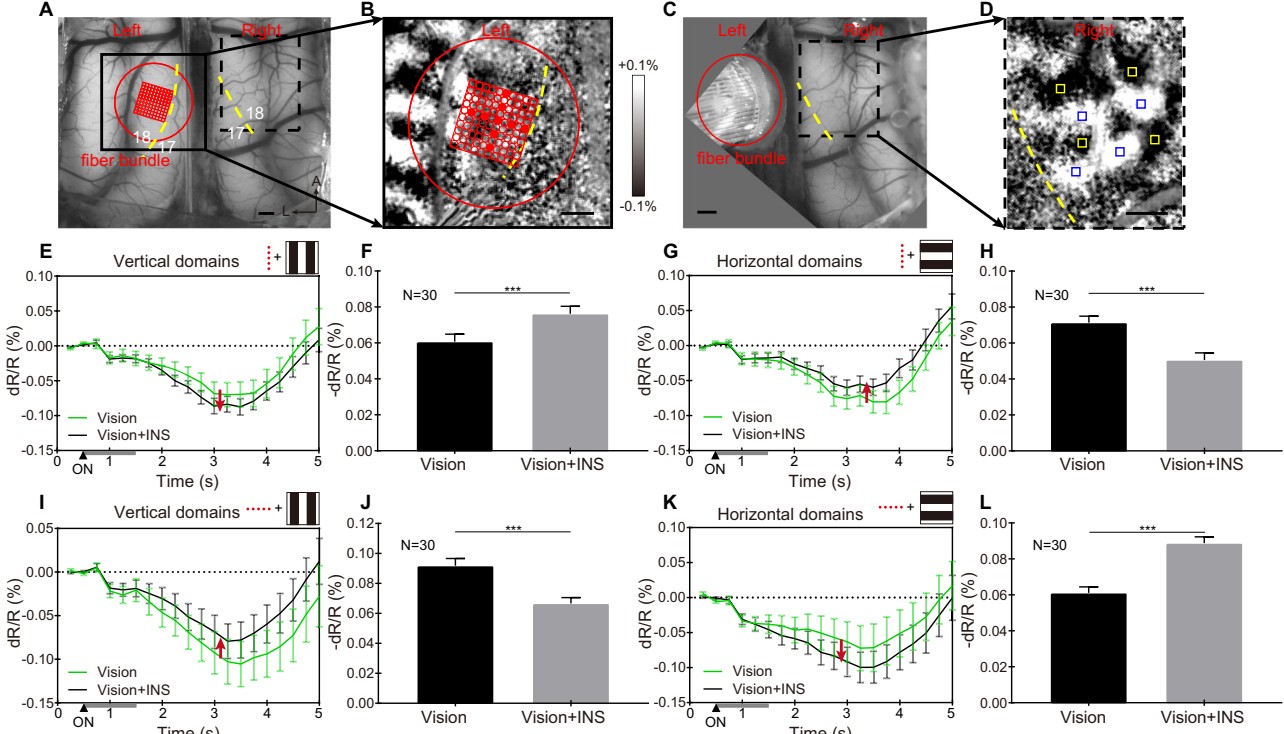

**Fig. 9 | Two orientations of INS stimulation show an orientation-selective effect without moving the fiber bundle. A** Vessel map of both hemispheres. Red circle and grid: 100-channel fiber bundle. **B** Orientation map, an enlarged view of the black box in **A**. Red dots: working channels in this experiment. Spacing between stimulating fibers: 440 μm. **C** Red circle: fiber bundle location, same as in **A**. **D** Orientation map in right hemisphere, an enlarged view of the black dashed box in **A**, **C**. Blue squares: ROIs in vertical domains quantified in **E**, **I**. Yellow squares: ROIs in horizontal domains quantified in **G** and **K**. Yellow dashed lines in **A**–**D** 17/18 border (VM). Scale bar in **A**–**D**: 1 mm. A anterior, L left. **E**–**H** 'Vertical' INS stimulation, parallel to VM in **B**. **E**, **G** Time course of OI signals for 'Vision' and 'Vision+INS'

stimuli in vertical **E** and horizontal **G** domains. Gray bar indicates the duration of INS. **F**, **H** Averaged reflectance changes during 2.5−5 s of the time courses in **E**, **G**. ***$P < 0.001$ (two-sided Welch's $t$ test, vision vs. vision + INS). **I**–**L** 'Horizontal' INS stimulation, orthogonal to VM in **B**. **I**, **K** Time course of OI signals for 'Vision' and 'Vision+INS' stimuli in vertical **I** and horizontal **K** domains. Gray bar indicates the duration of INS. **J**, **L** Averaged reflectance changes during 2.5−5 s of the time courses in **I**, **K**. ***$P < 0.001$ (two-sided Welch's $t$ test, vision vs. vision + INS). $N = 30$ trials in **E**–**L**, data are presented as mean values ±SEM. INS: 1870 nm, 200 Hz, 0.3 J cm$^{-2}$, pulse width 250 μs, pulse train 1 s. Data from Cat 5.

**Intensity dependence and general suppression.** Effects were intensity-dependent: weak effects were seen at 0.1 J cm$^{-2}$; a clear modulating effect was observed at 0.3 J cm$^{-2}$, and at higher intensities of 0.7 J cm$^{-2}$, a general suppression of response was observed, consistent with other reports of suppression at higher INS intensities[17,69,70]. As shown by multiple previous studies[71], such intensity dependence is an important element of developing a reliable VCP.

**Human use.** As there is no need for viral transfection, INS may be more amenable for human use. In fact, some clinical trials using INS have already begun for peripheral nerve stimulation and cochlear stimulation (Jansen NCT04601337, Richter NCT05110183). Recently, its application for intra-operative use in the human cortex has also been demonstrated[33]. Unlike optogenetics, INS is not cell-type specific; however, in human and nonhuman primates, functional specificity can be achieved by stimulating single cortical columns, leading to activation of their associated functionally specific circuits.

**Other considerations.** While fiber optic devices can cause less tissue damage when used for cortical surface stimulation and can be placed in different locations on the brain, the array must be removed when not in use, limiting its capacity for use in freely behaving monkey studies; to address this, it may be possible to engineer a wireless μ-LED stimulation device[16,72–74]. Another limitation is the lack of ability to monitor neural spike activity, which would require the incorporation of additional electrodes. One aim of the 100-channel fiber bundle is to enable selective stimulation of color, form, motion, and depth

columns in awake-behaving monkeys; such feature-specific percepts can be induced to be demonstrated. Finally, we note the possibility for INS to be used for deep brain stimulation[75,76].

We consider whether optical approaches can achieve the stimulation capabilities of electrical VCPs. Evidence from previous studies using electrical stimulation suggests that selective stimulation of the cortex evokes naturally specific percepts. While the specific parameters of different studies vary (intensity, frequency, duration), we summarize a few studies that potentially achieved column-specific stimulation by employing relatively low stimulation intensities. Salzmann & Newsome[7] showed that a monkey's direction discrimination of a moving dot field is biased towards the preferred direction of the electrically stimulated (10 μA in 1 s) column. Similarly, in the disparity axis, DeAngelis and colleagues[8] showed that stimulating (20 μA) patches of cells with selective disparity tuning in MT resulted in a perceptual bias towards the stimulated near-to-far column's disparity preference. In a carefully conducted study in which the visibility of a small color spot was eliminated by presenting a matching background color, the equivalent electrical stimulation (15−55 μA) needed to achieve a similar effect was identified, illustrating that stimulation of V1 sites elicited unsaturated color percepts[9,77,78]. In humans, Schmidt et al.[10] showed that stimulating with low current levels near threshold (e.g., 1.9 μA) in ventral temporal cortex, percepts of color were often evoked and that increasing stimulation led to more washed-out color percepts such as white, gray, or yellow (consistent with the blurring of signals across multiple functional domains). Stimulation of face patches in the temporal cortex led to non-specific distorted face percepts,

perhaps due to the higher electrical current range used (50–300 µA)[11]. Thus, together, these examples may support the hypothesis that stimulation of a single cortical column or patch can, with the lower stimulation intensities, lead to a perception related to the column-encoded feature. Further investigation is needed.

Given the percepts induced by single-site stimulation, what can be elicited by multisite stimulation? A few studies have examined the percepts evoked by multisite electrical stimulation, either sequentially[1,2,5,6] or simultaneously[4,5]. In human studies, surface electrode arrays have evoked striking percepts of shapes and letters in the blind. These percepts can be enhanced by a 'dynamic current steering' stimulation paradigm[6], enhancing the salience of the percept and reducing the number of stimulation sites needed[2,79]. In a study by Fernández et al.[4], a blind subject implanted with a Utah array was able to discriminate evoked sizes and shapes as well as spatially small percepts, marking a significant advance in spatial resolution (see also Yoshor and Bosking[3]). Roelfsema's group implanted an astounding 1024-channel-count electrode interface (16 Utah arrays, 0.3–0.6 MΩ, <100 µA) into monkey V1 and V4, achieving percepts of shapes, letters, and motion direction[5]. We previously showed[28] that INS applied to the visual cortex induces visual phosphenes, leading to predicted saccade behavior in the awake, behaving monkey. These findings indicate the feasibility of INS being used as a cortical stimulation method for VCPs. It remains to be seen whether optical stimulation approaches can achieve the groundbreaking capabilities of electrical VCPs.

Our aim here was to interface with known visual circuits. That is, in our VCP design, we aimed to target known functional columns in the visual cortex and predicted the resulting activation based on known circuitry. We find that our results are quite consistent with previous studies. In a study in the ferret, which recorded local field potentials during presentation of grating stimuli, removal of inputs from one hemisphere by cooling revealed a primary excitation between orientation-matched neurons and a secondary suppression between orthogonally oriented neurons[40]. Here, we find that stimulation through a linear array produced enhanced response in higher order orientation domains matching that of the array and suppressed responses from orthogonal higher order domains. As proposed by Makarov[40], this integration is likely mediated via integration in contralateral area 18 (Fig. 1B, upper). Thus, we show that external activation is capable of activating callosal circuits in a manner very similar to visual stimulation. This demonstration supports the possibility that externally applied focal stimulation can access functional circuitry known to underlie normal visual perception.

## Methods
### Animal procedures
All procedures were in compliance with and approved by the Institutional Animal Care and Use Committee of Zhejiang University and followed the guidelines of the National Institutes of Health Guide for the Care and Use of Laboratory Animals. Experiments were performed in hemispheres of 13 *cats* (weighing 3–4 kg). Three animals were used to develop the stimulation procedure, 2 animals to acquire intensity dependence data, 5 animals to obtain the INS/vision combined data via linear fiber array, and 3 animals to obtain the INS/vision combined data through a 100-channel fiber bundle. Since our study focused on INS modulation of the visual cortex, differences between sexes are unlikely to our findings. For surgical procedures, cats were sedated with ketamine hydrochloride (10 mg/kg)/atropine (0.03 mg/kg) and anesthetized with isoflurane (1.5–2%)[18]. Animals were intubated and artificially ventilated. End-tidal $CO_2$, respiration rate, $SpO_2$, heart rate, and electrocardiogram were monitored and maintained. Temperature (37.5–38.5 °C) was monitored and maintained via a circulating water blanket. A craniotomy (Horsley-Clarke coordinates A10-P5, L7-R7) and durotomy were performed to expose visual cortex areas 17 and 18 of both hemispheres. A piece of transparent acrylic sheet, with an opening for insertion of a linear fiber array, was placed on top of the cortex to suppress physiological motion. After the stabilization of the cortex, for the acquisition of functional neural signals in the brain, cats were anesthetized with propofol (induction 5 mg/kg, maintenance 5–10 mg/kg/h, i.v.) and isoflurane (-0.5%). To prevent the movement of the eyes, a muscle relaxant was used (vecuronium bromide, induction 0.25 mg/kg, maintenance 0.05-0.1 mg/kg/h, i.v.). EEG was monitored to ensure the maintenance of sleep spindles consistent with anesthesia level-2 used in human neurosurgery. The pupils were dilated with atropine sulfate (1%) and the nictitating membranes were retracted with neosynephrine (5%). The eyes were corrected with appropriate contact lenses to focus on a monitor 57 cm in front of the animal. At the end of data collection, the animal was overdosed with pentobarbital sodium (100 mg/kg, i.v.).

### Visual stimuli
Visual stimuli were created using ViSaGe (Cambridge Research Systems Ltd) and presented on a monitor refreshed at 60 Hz and positioned 57 cm from the cat's eyes. Visual stimuli were presented binocularly. Full-screen drifting square gratings were used to obtain orientation maps in visual cortical area 18 (0.2 cycle/degree, 5 Hz) and area 17 (0.5 cycle/degree, 2 Hz). Bidirectionally moving gratings were presented in a randomly interleaved fashion in 1 of 2 orientations (vertical, horizontal).

### Intrinsic signal optical imaging
Imaging was performed with Imager 3001 (Optical Imaging Inc., Israel). Images were obtained from the CCD camera (1308 × 1080 pixels, 10.8 × 8.7 mm) with 630 nm illumination. Frames were acquired at 4 Hz for 4 s or 5 s and were synchronized to respiration. Visual stimuli were presented to obtain orientation maps and determine the border of areas 17/18. Each condition was presented for 20–30 trials, pseudo-randomly interleaved with at least a 7 s intertrial interval. A blood vessel map of the cortex (Fig. 2A) was also acquired under 570 nm illumination for analysis.

### Infrared neural stimulation
We stimulated using an 1870 nm infrared pulsed laser (Changchun New Industries Optoelectronics, FC-W-1870nm-4W, China), coupled to an optical switch (Piezo Jena AG, Germany) and then to a multi-channel linear fiber array (200 µm-diameter per fiber, 500 µm fiber core spacing) (Fig. 3A) or a 100-channel fiber bundle (200 µm diameter per fiber, 220 µm fiber core spacing, 10 × 10 densely packed) (Fig. 8B, C). The fiber array was carefully positioned with a 3D-manipulator (RWD, China, stereotaxic micromanipulator), guided by the optical imaging camera mounted directly above the region of interest (shown as Fig. 2A–C), the angle between the fiber and cortical surface was around 45° (the linear fiber array) or at 90° (the 100-channel fiber bundle), and the fiber tips were just in contact with the cortex. The maximum output power of the laser device at 200 Hz frequency is 200 mW and the output power can be adjusted. Before each experiment, a power meter (Thorlabs, USA, PM320E, with thermal power sensor: S401C) is used to measure the output energy from the fiber tip, to ensure that the peak energy density (radiant exposure) of the laser pulse at the tip of the fiber reaches the target value, such as 0.3 or 0.7 J cm$^{-2}$ (19 or 44 mW), calculated using:

$$\text{Power} = \text{radiant exposure} \times \text{area} \times \text{pulse repetition frequency} \quad (1)$$

The pulse generator (Master-9, Israel) continuously generated an output signal of 200 Hz (INS stimulation frequency). Then, an Arduino control board (Italy) was programmed to determine the duration (16 ms or 200 ms) and order (sequential or random) of each channel of the optical switch. As the channel switching is controlled by the optical

switch while the laser is always on during this process, the output power of the laser is stable.

As shown in Fig. 3B, we generated both fast and slow sequential INS stimulations (Fast: 200 ms per tip, Slow: 16 ms per tip); for each condition, the total duration of laser stimulation across all five tips was 1 s. Each stimulation was also delivered in two directions (Fwd, Rev). The Slow and Fast conditions (200 ms per tip and 16 ms per tip) are similar to a dot moving at ~3 degrees per second and ~30 degrees per second, respectively (typical velocity preference of cat area 18-25 degrees per second[56]). All INS was initiated 0.5 s after the onset of optical imaging data acquisition ($t = 0$) and coincident with the onset of visual stimulation. Conditions were pseudo-randomly interleaved within each trial and repeated 15 times. Image acquisition was synchronized with respiration to reduce variability in OI reflectance signal.

INS imaging experiment comprised 13 conditions at a specific radiant exposure (0.3, 0.7 J cm$^{-2}$). These conditions contain blank condition (no visual stimulation or INS); 2 INS off conditions with visual stimuli (vertical and horizontal grating orientations); and 10 INS on conditions, including two orientations, each paired with five different INS paradigms (two directions: forward, reverse; two speeds: slow, fast; and one rapidly, randomly changing stimulus) (see Fig. 3B).

### Data analysis
Data analysis was performed with Matlab (The Mathworks, Matlab-R2021a). To obtain single condition maps, for each trial, the reflectance change of each pixel was calculated first using the functions: $dR/ = (F_x - F_o)F_o$, where $F_X$ is the reflectance value corresponding to frames × ($X = 1$-20), $F_0$ is the average reflectance value of the first two frames (taken before laser or visual stimulation onset). To identify regions of maximal response, recorded images were first smoothed (Gaussian filter, 3–8 pixels diameter). Low-frequency noise (e.g., gradations in illumination across the cortex, slow 0.1 Hz oscillations, slow physiological fluctuations) was reduced by convolving the maps with a 150–200 pixels diameter (~1.2–1.6 mm) circular filter and subtracted from the smoothed maps[80].

Difference maps between two stimulus conditions (e.g., INS vs. blank or two orthogonal orientations) were obtained by calculating the average difference of filtered single-condition maps. Maps were also clipped as described above. For INS only, maps were calculated from the average of the fourth to the seventh frame (Figs. 2D, 4D); for visual stimulation alone and combined visual stimulation and INS, frames 9–16 were used (Figs. 2B, C, 5A, 6A). For the imaging results, pixel values were clipped by mean ±1.5/2 standard deviation (SD) or mean ±0.05/0.1% values (e.g., Supplementary Fig. 2B, values larger than mean +2 SD/0.05% was truncated to mean +2 SD/0.05%).

A pixel-by-pixel two-sided $t$ test corrected for multiple comparisons was performed to examine the significance of the reflectance changes. Only regions with $p < 0.001$ were selected for further analysis (e.g., right panels in Figs. 5A and 6A, Figs. 8D and 9D, Supplementary Fig. 3). The time course of the changes in intrinsic signal was visualized by taking pixels with significant activation and plotting the mean value as a function of time (See Fig. 5B, C). These significant regions were also used to estimate the activation size. We manually drew the elliptical contour of the activation area and measured the lengths of the long and short axes using Image J (1 pixel = 8 μm). The average value of the long and short axes was calculated as the diameter of the activation area.

### Statistics
Prism (GraphPad Software, Prism 8) was used for statistical analysis. To test whether the five fibers from the linear array evoke similar response amplitude, a one-way ANOVA test was performed on the peak amplitude of OI response curves to determine statistical significance among them (Fig. 4c). To test whether INS significantly modulates the cortical response to visual stimuli, after homogeneity of variance validation, a

Welch's $t$ test (two-sided) was performed on averaged data (typically frames 10–20) to determine statistical significance between two conditions (INS on vs. off).

### Reporting summary
Further information on research design is available in the Nature Portfolio Reporting Summary linked to this article.

## Data availability
All data supporting the findings of this study are available within the article and its supplementary files. Any additional requests for information can be directed to and will be fulfilled by the corresponding authors. Source data are provided in this paper.

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

## Acknowledgements

This work was supported by: STI 2030-Major Projects 2021ZD0200401(to A.W.R.), Key Research and Development Program of Zhejiang Province 2020C03004 (to A.W.R.), National Natural Science Foundation of China U20A20221, 819611280292 (to A.W.R.), and MOE Frontier Science Center for Brain Science & Brain-machine Integration at Zhejiang University. We thank Xinjian Li and Rob Friedman for their helpful comments.

## Author contributions

F.Y.T. and Y.Z. performed the experimental studies. F.Y.T. carried out the analysis. F.Y.T, K.E.S, J.M.H., and A.W.R. designed experiments and wrote the manuscript. A.W.R. supervised the work and provided the funding.

## Competing interests

The authors declare no competing interests.
