## [Peer Review File · Nature Communications]

REVIEWER COMMENTS

Reviewer #1 (Remarks to the Author):

Overview

INS is a very interesting technique, and certainly merits further study and development for clinical applications. The authors demonstrated impressive technical skill in imaging and combining techniques. However, their representation of electrical stimulation is misconstrued and ultimately their proposed method results in an activation radius similar to electrical micro stimulation, despite INS framed as more focal. Furthermore, the authors pose INS as a potential modality to restore functional vision to blind subjects through a VCP, however it is unclear how the specific methods or techniques demonstrated here would be implemented as a VCP. The experiments presented in this manuscript are largely INS coupled with visually evoked activity, but the context is framed around VCPs. Modulation of ongoing activity was demonstrated, however, no activation to drive perception was demonstrated. This is interesting for INS as a scientific tool, but not relevant for VCP as the blind VCP users will have no visually evoked activity. For VCPs, it may be useful to modulate ongoing neural activity, but the fundamental application requires activating cortex in such a way that it drives perception of a visual sensation.

The authors posit three challenges to VCPs. Creation of visual features is certainly a goal, along with creating focal activation patterns at functionally relevant cortical column scales. However, the integration between left and right hemispheres is not presently a fundamental hurdle for VCPs to overcome. This third goal does not fit well with the theme of VCPs, but does correspond to the results presented. The effects in the contralateral hemisphere do not seem relevant to VCP function.

The authors indicate they demonstrated INS induces higher order effects, but that is unclear both what they mean by this and that the results presented demonstrate higher order effects.

Overall, this is original, interesting work with that employed noteworthy techniques, but the relationship of the experiments conducted to VCPs is not well laid out. Ultimately, the manuscript is missing essential context for VCPs and electrical stimulation that are needed to make an appropriate comparison with INS or a valid argument for INS to be used for a VCP.

Details

The authors did not properly cite articles to reflect the activation radius for electrical stimulation (Stoney1968, Asanuma 1976, Tehovnik 2006, Tehovnik 2007) or indicate the expected range for

intracortical microstimulation (ICMS) (Chen 2020, Fernandez 2021, Bak 1990, Torab 2011, ...). Fernandez 2021 reported an average threshold of $66.8 \pm 36.5 \mu\text{A}$ for single electrodes. This puts the estimated activation diameter to be in the range of 0.45 – 0.63 mm, depending on the K value used. The authors did not give exact measurements for the diameter of activation with INS (they should) for each energy level and only stated “submillimeter”. It would be useful to indicate diameter of activation for each energy level tested. Based on the figures, it would seem it is in the range of 0.5 – 1 mm. Based on published data, it does not seem that INS has a significant benefit in terms of activation radius compared to ICMS. However, INS does not puncturing the parenchyma like ICMS. Other benefits of INS over ICMS may also be reasonable and supported by data/literature and would be better used to argue for an INS approach for VCPs over ICMS rather than activation radius. This type of addition to the discussion would be relevant/welcome context if the authors wish to frame INS as an alternative approach to VCPs.

There seems to be a bit of a conflation between ICMS and electrical stimulation with macro surface electrodes in the discussion. ICMS tends to have very low current thresholds to evoke phosphenes (20 – 200 μA), macro surface electrodes (~2 mm) tend to be larger (~mA range, varies depending on whether blind or sighted). Mini surface electrodes tend to be in the middle (~0.5 mA), and micro ECoG electrodes may have thresholds between mini and micro electrodes.

The authors indicate VCPs have “yet to achieve the basic qualitative aspects of vision” yet cite in the following line an example of simple form vision delivered with electrical stimulation. However, there is greater missing context to this citation that is relevant to the discussion on VCPs. While the Chen 2020 paper did demonstrate some limited efficacy for sighted NHPs to perform a 2AFC, the electrodes used were far apart, and the NHPs could have performed the discrimination task based on low level features. Secondly, the Fernandez 2021 study (not cited by authors) delivers a follow-up to the Chen study with a human subject who was implanted with one Utah array in visual cortex. The results from Fernandez 2021 were complicated. While the researchers in that study were able to evoke perception of some simple characters in their blind subject, the results were inconsistent. Some patterns that would be expected to create a character didn't, and other times unexpected spatial patterns results in characters. Sometimes electrical stimulation on neighboring electrodes resulted in perception of separate phosphenes, sometimes electrical stimulation on electrodes across the array resulted in perception of one phosphene. This is further discussed in a commentary response to the paper (Beauchamp 2021). A more substantive critique of ICMS than is presented in the manuscript is possible and could better be used to bolster INS as an alternative approach.

The authors appear to be presenting data that indicates INS delivered to in a linear location on one hemisphere of V1 results in activation in the other hemisphere, in a different part of the visual field, and in different functional representation. It is unclear how this relates to VCPs or illustrates higher order integration.

Presentation of some of the raw imaging data would be useful.

Authors claim each of the following as novel aspects in the discussion, response to each is included. May be better to indicate these as features or attributes rather than calling them novel.

1. Mesoscale stimulation

- o This scale is not novel to electrical stimulation. ICMS at levels that can evoke phosphenes can be more focal than the levels demonstrated with INS

- o This discussion also indicates the need for discriminable individual phosphenes. A “pixel” based approach is an outdated framework for VCPs

2. Neuromodulation

- o Neuromodulation is a type of neural stimulation, there is no role shift.

- o Using directional stimulation on cortex or delivering a linear pattern is not novel

- o High levels of electrical stimulation can also inhibit spread of activation (Bosking 2017, Neuropace RNS)

- o This is not a “demonstration of functionally specific neural modulation”

3. Dynamics

- o Delivering 10 different stimulation patterns is not novel

- o Slow, fast, and random electrical stimulation have been used in other studies

4. Higher order response

- o The authors state “Here, presenting a sequence of point stimuli, which can induce the percept of motion[18] as well as lower or higher order shape[29,46–50]”

- o No demonstration of perception induced by INS was presented in this manuscript

- o The authors state “our linear array fiber tip locations targeted different orientation domains in one hemisphere”. This was not clearly demonstrated. Activation seems to be in ~0.5 – 1 mm chunks and not in specific orientation columns, could use a clear demonstration of this activation

- o Linear patterns resulted on contralateral activation while random did not. This would seem to indicate the tissue is responding to the linear spatiotemporal pattern more so than activation of specific orientation columns.

5. Linking contralateral visual hemifield

- o It is not clear how this is relevant to a VCP

6. Intensity dependence and general suppression

- o Changing effects with intensity of stimulation are not a novel demonstration

Lines 253 – 254. The scale of activation presented is closer to hypercolumn scale than single column scale.

Lines 278 – 291. Feature specific functional activation would not be possible at the scale of the Orion electrodes and did not seem to be a goal given the interface. Beauchamp 2020 did use “form from motion” to evoke perception of simple characters. The authors here used similarly structured presentation of INS to yield their evoked responses.

Line 286. Odd usage of “feelings”, should use perception.

Lines 287 – 291. The authors reverence “knowledge about the precise stimulated column or specific functional neural response” but don’t expand on the what this means in context

Lines 307 – 308. The authors state “Thus, our BMI is capable of activating callosal circuits in a manner very similar to visual stimulation” but this does not seem to be fully supported by their data.

Reviewer #2 (Remarks to the Author):

The current study introduced a new approach to visual cortical prosthetics (VCP), using pulsed infrared neural stimulation (INS) in cat V2 that achieves the spatial precision required to interact with small cortical columns responsible for visual feature perception. Tian and colleagues used an optical fiber array, producing cortical activation similar to a moving visual dot (the trajectory of a point of light is defined as contour orientation). The stimulation enhances responses in orientation-matched domains and reduces responses in non-matched ones in contralateral V2, with the effects varying based on the speed, direction, and pattern of stimulation, emphasizing the orientation-selective nature of this approach. The story sounds interesting. However, my enthusiasm for this research has been hampered by the following shortcomings:

Major :

1. The most crucial technical indicator for the development of visual cortex prostheses (VCP) is to acquire information about the subjective perceptual experiences of the subjects in response to the INS or

electrical stimuli. This includes aspects such as the presence or absence of phosphenes and the location, size, shape, and color of a presented phosphene. In this article, the use of anesthetized cats seems inadequate for obtaining behavioral reports of subjective perceptual experiences elicited by INS stimuli. In contrast, previous research with awake macaques has made it possible to assess the subjective perceptual experience of cortical stimuli using the macaque eye movement selection paradigm (Chen et al., 2020 Science). It would be worthwhile to repeat the INS stimulation experiments with awake cats and use a similar behavioral paradigm as in macaques to assess whether INS stimulation can elicit phosphenes. Otherwise, the current findings may have limited applicability in the development of VCPs.

Chen, X., Wang, F., Fernandez, E., & Roelfsema, P. R. (2020). Shape perception via a high-channel-count neuroprosthesis in monkey visual cortex. *Science*, 370(6521), 1191-1196.

2. I am concerned about the potential for tissue damage from INS, as this could limit the applicability of the INS technique in VCPs. In the Introduction section, the authors assert that INS is a non-damaging approach. However, the fundamental principle of INS is the ability of infrared light to cause a rapid, localized temperature increase within neural tissues. This swift increase in temperature leads to a transient change in the cell membrane's capacitance, which can trigger action potentials. If the authors aim to highlight that INS stimulation is less traumatic than electrical stimulation, they should at least compare the near-threshold intensities of INS and electrical stimulation when the stimulation produces a reportable phosphene. A simpler method might involve *in vitro* measurements of temperature changes. Histological results would provide even stronger evidence.

Moreover, in Beauchamp et al., 2020 *Cell*, dynamic electrical stimulation was shown to generate phosphenes at efficiencies of up to 30 forms per minute. If similar efficiencies can be achieved, would INS cause more damage than electrical stimulation?

Beauchamp, M. S., Oswald, D., Sun, P., Foster, B. L., Magnotti, J. F., Niketeghad, S., ... & Yoshor, D. (2020). Dynamic stimulation of visual cortex produces form vision in sighted and blind humans. *Cell*, 181(4), 774-783.

3. The authors did not clarify in the Results section whether there is a difference between the forward and reverse directions of the sequential INS stimuli. However, Figure S1 clearly shows different effects between the two directions of stimulation. This is confusing because, according to the authors' hypothesis, the forward and reverse direction of the sequential INS stimuli should produce stimuli in the same contour orientation in the cat's cortex, thus having the same effect on the contralateral cortex.

4. Another question that arises is why the authors did not record the response on the same side as the INS stimulus, which could arguably be more direct. An iso-orientation surround suppression effect, but not a cross-orientation suppression effect, should be generated between neurons with adjacent receptive fields. The cross-orientation suppression effect in the current study is supported by fMRI experiments on human subjects (ref28, Brouwer & Heeger, 2008). However, electrophysiological data usually show iso-orientation suppression. Are there any electrophysiological data that support the cross-orientation suppression effect?

5. The authors' method of defining the orientation of the trajectory involves measuring the boundaries of V1 and V2 first and then inferring them from the brain atlas. It would be more convincing if receptive fields were measured before applying INS stimulation experiments.

6. There are two concepts of orientation in the text that can be confusing. One refers to the orientation preference of the column itself, and the other to the contour orientation of the sequential INS, which stimulates the generation of an imaginary slash. It would be more precise to use the terms 'column orientation' and 'contour orientation' respectively.

7. I am not sure whether this cross-hemisphere neuromodulation in V2 is generalizable to humans and non-human primates. The authors' findings regarding callosal connections between the left and right V2 (ref 32-36) are based on studies in ferrets and cats. Are there previous findings in humans or non-human primates that support this?

8. In the Results section, the statement "Under these higher fluence conditions, we observed a reduction in the OI reflectance signal irrespective of the orientation of the fiber optic array" needs to be accompanied by statistical test results.

9. The manuscript addresses two distinct concepts of orientation: the orientation preference of different columns in V2, and the contour orientation induced by sequential multi-fiber stimulation. However, the same term 'orientation' is used for both concepts in the manuscript, leading to confusion. It would be clearer to distinguish between them using terms like 'orientation' and 'contour orientation'.

Minor:

1. The full form of the abbreviation "BMI" is not provided in the manuscript.

2. The radiant energy of INS is not specified in Figure 5 and 6.

3. The term "higher order callosal response" can be confusing. The phrase "high-order visual processing facilitated by callosal connections" would be clearer.

Reviewer #3 (Remarks to the Author):

The introduction to this paper makes some very broad claims about how this technology will address the shortcomings of bionic vision systems to meet requirements such as: (i) color, resolution, depth, contour orientations, motion directions; (ii) higher-order features; (iii) complete visual field and integrating left-right information. Line 65 states: "Here we present an interface that addresses these three challenges". However, the presented results are extremely preliminary and at best suggest that such things may be possible, but the system is far from practical: a few spots under a linear array of optical fibers (connected to large equipment) have been stimulated, and this stimulation shows a response in the opposite hemisphere.

My first concern is with the description of the optical system and energies. It appears that a 4-W laser is being used (though I could not find this exact laser in the company catalog), which is passed through a

low-loss piezo-mechanical optical switch to select a delivery fiber. The fibers are 200 um diameter (though the core size is not given). The reported energies are up to 0.7 J/cm² (presumable for a complete stimulation event?).

So, the energy out of the core of the fiber could be calculated as 0.7 J/cm² * Area, where the area is $\pi * 100 \text{ (um)}^2 = 0.7 * 1e-8 * \pi * 1e4 = 0.22 \text{ mJ}$. If we say this heats a 200-um deep 200-um diameter cylinder, then the temperature rise would be about 2 degC. Is this about right? Obviously any rise above this would be forbidden (usually rises <2degC are acceptable for RF heating).

However, taking a different route for calculations, we have a 4-W average power laser delivered by a 200-um fiber (perhaps a smaller core?), so a 16 ms stimulation would deliver 64 mJ, resulting in a 300x - higher temperature rise, which would be destructive.

My point is that there needs to be a calibrated measurement of the optical energy delivered, using say an integrating sphere and calibrated detector. Also, the waveform of the pulses needs to be captured, because it's not clear whether the switch is acting fast enough (there were no speed specifications on the manufacturer's site). Only then can some claim of "non-destructive" be made. Indeed, the temperature rise should be measured to confirm the volume being heated, which would help justify the specificity claim.

The results mainly rely on very small changes in reflectance of the cortical surface (<0.1 %) to indicate neural activity, convolution with a spatial filter (1.2 mm diameter), and a low-pass filter (surely not 3-8 pixels in diameter, but some frequency in Hz?), truncation of extraneous results, and scaling by root-N divided by the standard deviation of the difference between stimulated and unstimulated measurements (line 400), which probably needs a better justification. Some calibration bars are required on the greyscale images, to identify the actual change in reflectivity. The definition of activation size should be made (is it a measure of the area of >50% change compared with the peak change?).

Features of the waveforms need explaining. Fig. 2E has fluctuations greater than the error bars (which are very difficult to see, presumable due to jpg encoding). Fig. 4B shows that the peaks are all the same, but the traces diverge towards 5 s: I would expect them to re-converge – why not? The later waveforms cut off at 5 s, long before reconvergence, which is disturbing. Fig. C has 2Hz oscillations, which is very strange.

A potential uncertainty in the experiment might be the mechanical pressure from the array. A single cross (+) shaped array, placed once and not moved would be more convincing.

The mechanism behind an array positioned V or H across many columns (where some are responsive to V optical stimulation, and others H) causing an increase or suppression of response in the contralateral cortex seem counter-intuitive. It might be more intuitive if a single V or H column was selected and a contra-lateral response observed. I know the present experiment is designed to show higher-order integration, but it seems a very complex effect.

I wonder whether the out-of-band energy from the 4-W heating laser directly affects the CCD camera, and whether any steps were taken to eliminate this hypothesis, such as using a piece of paper rather than a brain to check for cross-talk?

Overall, there are some very interesting observations in this work, but I would like more clarification on the experimental technique, and how it was verified and calibrated, as the signals are very tiny, and some of the responses not what I would expect (e.g. not returning to the resting state after stimulation, which would indicate some damage). In some cases, it would be more convincing to plot (every measurement, rather than presenting averages).

Response to reviewer comments:

We thank the Reviewers for their constructive comments and feedback and the Editor for giving us the opportunity to submit this revision. We have implemented all of the recommended changes and responded to the stated concerns. Changes in the revised manuscript have been marked in bold typeface with the corresponding referee and comment number (e.g., R1-1). We feel that the revised manuscript has been improved by the recommended changes. Below we respond to each point raised.

Reviewer #1 (Remarks to the Author):

R1-1. INS is a very interesting technique, and certainly merits further study and development for clinical applications. The authors demonstrated impressive technical skill in imaging and combining techniques. (1) However, their representation of electrical stimulation is misconstrued and ultimately their proposed method results in an activation radius similar to electrical micro stimulation, despite INS framed as more focal. (2) Furthermore, the authors pose INS as a potential modality to restore functional vision to blind subjects through a VCP, however it is unclear how the specific methods or techniques demonstrated here would be implemented as a VCP. The experiments presented in this manuscript are largely INS coupled with visually evoked activity, but the context is framed around VCPs. (3) Modulation of ongoing activity was demonstrated, however, no activation to drive perception was demonstrated. This is interesting for INS as a scientific tool, but not relevant for VCP as the blind VCP users will have no visually evoked activity. For VCPs, it may be useful to modulate ongoing neural activity, but the fundamental application requires activating cortex in such a way that it drives perception of a visual sensation.

Response: We thank the reviewer for the thoughtful evaluation of the INS approach.

(1) “...*their representation of electrical stimulation is misconstrued.*” The reviewer states that electrical stimulation and INS stimulation can be equally focal. We agree with the reviewer on this (Histed et al. 2009, Schiller et al. 2011). Despite this, targeted stimulation of cortical columns has not been an explicit strategy of electrical VCPs (Roe et al. 2020, Roe 2022). *That is, electrical VCP for column-specific activation may be feasible but has not been developed.* We ourselves developed quite focal methods for electrical stimulation of single cortical columns and mapping their associated circuits (Hu et al. 2020, Hu and Roe 2022). However, we decided that application of optical fiber bundles would be a more flexible and non-damaging approach for VCP, as it can be applied to the cortical surface without tissue penetration (which can cause damage) and can be moved around to different cortical locations as needed. Although this may not be optimal for all brain areas, for area V1 and parts of V2 and V4, it is an attractive approach. For deep areas, we have used designs with single optical fiber stimulation approaches (Xu et al. 2019, Shi et al. 2021, Yao et al. 2023). **We have now added text in the Introduction and substantial paragraph in the Discussion on this topic.**

(2) “...*it is unclear how the specific methods or techniques demonstrated here would be implemented as a VCP.*” The reviewer is correct. We did not explain this. We now refer to our plan (described in our paper entitled “A roadmap to a columnar visual cortical prosthetic” Roe et al. 2020) in the Introduction and describe it in the Discussion. Previously, we targeted single 200um fibers to cortex via an implanted optical window (through which optical imaging of cortical functional organization was first conducted) and then showed this focal INS stimulation leads to activation of highly specific brainwide mesoscale circuits as well as activation of laminar specific connectivity, replicating known anatomical feedforward and feedback connections (Xu et al. 2019). **We plan to apply multiple fiber bundles through windows on the brain.** We have constructed a 100-fiber bundle prototype capable of delivering different patterns and dynamic sequences of columnar stimulation applied to anesthetized monkeys in 7T MRI to map brainwide columnar networks (Schriver et al. 2022 FENS). Using our imaging methods capable of mapping columnar organizations in the dorsal and ventral pathways (Wang et al. under view), we are now applying this approach to stimulate imaged columnar maps in awake monkeys performing discrimination tasks in the 7T MRI, aiming to shift behavioral discrimination curves, columnar neural (BOLD) response curves, and study effects on neural circuitry. **Confirming that higher order activation can be evoked by multifiber stimulation is a key step in establishing the utility of this VCP approach on brainwide circuits. This is the aim of the study here.**

(3) “...*no activation to drive perception was demonstrated*” We agree with the reviewer’s important comment. *INS induced phosphenes:* We present the reviewer’s evidence that single fiber

INS stimulation generates phosphenes to which monkeys saccade. **Please see our response to Reviewer 2 Comment #1.** For the study in this manuscript, the goal is confined to demonstrating that, for an effective VCP, a multipoint stimulation can produce functionally specific higher order activations. *Neuromodulation in the blind*: We agree with the reviewer that the aim of visual cortical prosthetics (VCP) is to enable vision in the blind. Blind subjects include those with no vision, low vision, and specific vision deficits (e.g., akinetopsia, prosopagnosia). Thus, visual cortical prosthetics encompass a spectrum of activation strengths, paradigms, and technologies for the purpose of vision enhancement. While the blind may have low retinally induced visual cortical response, studies have shown that visual cortical response of similar magnitude and specificity can be induced from internally generated influences such as memory or imagery (Podgorny and Shephard 1978, Senden et al. 2019). Thus, even in the blind, columnar maps would still likely be accessible and potentially mappable via imagery or memory paradigms (e.g., www.sciencedaily.com/releases/2007/02/070220021337 “Those who once were blind can learn to see”) and would thus be candidates for prosthetic intervention.

References:

- Histed, M. H., Bonin, V., & Reid, R. C. (2009). Direct activation of sparse, distributed populations of cortical neurons by electrical microstimulation. *Neuron*, 63(4), 508-522.
- Hu, J. M., Qian, M. Z., Tanigawa, H., Song, X. M., & Roe, A. W. (2020). Focal electrical stimulation of cortical functional networks. *Cerebral Cortex*, 30(10), 5532-5543.
- Hu, J. M., & Roe, A. W. (2022). Functionally specific and sparse domain-based micro-networks in monkey V1 and V2. *Current Biology*, 32(13), 2797-2809.
- Podgorny, P., & Shepard, R. N. (1978). Functional representations common to visual perception and imagination. *Journal of Experimental Psychology: Human Perception and Performance*, 4(1), 21.
- Roe, A. W., Chen, G., Xu, A. G., & Hu, J. (2020). A roadmap to a columnar visual cortical prosthetic. *Current Opinion in Physiology*, 16, 68-78.
- Roe, A. W. (2022). Neuromorphic model of human intelligence: Brain neuromorphic needs an architectural framework. *IEEE BrainInsight* ([http:// brain.ieee.org/publications/braininsight/](http://brain.ieee.org/publications/braininsight/)). Dec 2022. Viewpoint.
- Schiller, P. H., Slocum, W. M., Kwak, M. C., Kendall, G. L., & Tehovnik, E. J. (2011). New methods devised specify the size and color of the spots monkeys see when striate cortex (area V1) is electrically stimulated. *Proceedings of the National Academy of Sciences*, 108(43), 17809-17814.
- Schriner KE, Zhang Y, Liu Y, Roe AW (2022) Illuminating the Mesoscale Connectome: A 100-fiber Infrared Neural Stimulation System. *Fed Eur Neurosci Soc. Paris, France. July 9-13, 2022.*
- Senden, M., Emmerling, T. C., Van Hoof, R., Frost, M. A., & Goebel, R. (2019). Reconstructing imagined letters from early visual cortex reveals tight topographic correspondence between visual mental imagery and perception. *Brain Structure and Function*, 224, 1167-1183.
- Shi, S., Xu, A. G., Rui, Y. Y., Zhang, X., Romanski, L. M., Gothard, K. M., & Roe, A. W. (2021). Infrared neural stimulation with 7T fMRI: A rapid in vivo method for mapping cortical connections of primate amygdala. *NeuroImage*, 231, 117818.
- Wang JB, ..., Roe AW (2023) Mesoscale organization of ventral and dorsal visual pathways in macaque monkey revealed by 7T fMRI. *Prog Neurobiol*, under re-review.
- Xu, A. G., Qian, M., Tian, F., Xu, B., Friedman, R. M., Wang, J., ... & Roe, A. W. (2019). Focal infrared neural stimulation with high-field functional MRI: a rapid way to map mesoscale brain connectomes. *Science advances*, 5(4), eaau7046.
- Yao, S., Shi, S., Zhou, Q., Wang, J., Du, X., Takahata, T., & Roe, A. W. (2023). Functional topography of pulvinar-visual cortex networks in macaques revealed by INS-fMRI. *Journal of Comparative Neurology*, 531(6), 681-700.

R1-2. The authors posit three challenges to VCPs. Creation of visual features is certainly a goal, along with creating focal activation patterns at functionally relevant cortical column scales. However, the integration between left and right hemispheres is not presently a fundamental hurdle for VCPs to overcome. This third goal does not fit well with the theme of VCPs, but does correspond to the results presented. The effects in the contralateral hemisphere do not seem relevant to VCP function.

Response:

Thanks for raising this issue. Callosal connections are important for integrating the left and right visual fields (e.g. for reading, Hunter et al. 2007), tying together the vertical meridian (Blakemore et al. 1983, Payne 1990, Payne and Siwek 1991), and the fovea (which is represented on both hemispheres, Kennedy et al. 1986). The coordination of visually guided bimanual behavior also requires the integration of the two hemispheres. Split-brain individuals exhibit deficits resulting from lack of callosal connections, revealing severe sensory, motor, and cognitive deficits of left and right hemifield coordination. These aspects of vision and visually guided behaviors are also challenges that must be addressed by VCPs, and is a goal introduced here.

References:

- Blakemore, C., Diao, Y. C., Pu, M. L., Wang, Y. K., & Xiao, Y. M. (1983). Possible functions of the interhemispheric connexions between visual cortical areas in the cat. *The Journal of Physiology*, 337(1), 331-349.
- Hunter, Z. R., Brysbaert, M., & Knecht, S. (2007). Foveal word reading requires interhemispheric communication. *Journal of Cognitive Neuroscience*, 19(8), 1373-1387.
- Kennedy, H., Dehay, C., & Bullier, J. (1986). Organization of the callosal connections of visual areas V1 and V2 in the macaque monkey. *Journal of Comparative Neurology*, 247(3), 398-415.
- Payne, B. R. (1990). Function of the corpus callosum in the representation of the visual field in cat visual cortex. *Visual neuroscience*, 5(2), 205-211.
- Payne, B. R., & Siwek, D. F. (1991). The visual map in the corpus callosum of the cat. *Cerebral Cortex*, 1(2), 173-188.

R1-3. The authors indicate they demonstrated INS induces higher order effects, but that is unclear both what they mean by this and that the results presented demonstrate higher order effects.

Response:

Thanks for raising this question. We have added a schematic (Fig. 1AB) to describe the integration. Here, we provide further explanation. What is meant by ‘higher order’ is that the response preference results from an integration of inputs. For example, orientation in V1 (orientation columns) results from an integration of unoriented inputs from the LGN (Hubel and Wiesel 1977); orientation in V2 (higher order orientation domains) results from an integration of different types of cues from V1 that lead to perception of a visual contour (see Fig. 1B, e.g. illusory contour, Kaniza contour, motion contour, depth contour) (Roe et al. 2009); ‘orientation’ in V4 (curvature domains) results from an integration of contour segments in V1 or V2 that lead to perception of curvature (Hu et al. 2020). Similar to primate V2, in cat Area 18, neurons are also sensitive to higher order contour responses (Song and Baker 2006).

In our experiment, each optic fiber was targeted to an orientation domain, resulting in integration of multiple orientations (example b in Fig. 1A). This suggests that the contralateral oriented response was due to **a topographical integration** (example e in Fig. 1A and not d as shown by the random control condition). While there is large aggregate receptive field size (2-8 degrees at -5 eccentricity along VM) in a single column in area 18, the 2-3 deg shift in spatial location over 2mm cortical distance of the fiber array could give rise to the resulting orientation selectivity (Hubel and Wiesel 1965, Tusa et al. 1979, Ferreiro et al. 2021), thus resulting in a ‘higher orientation-selective effect’. These results actually mirror findings from Makarov’s 2008 study (Makarov et al. 2008) which showed (see their Fig. 7) that, in anesthetized ferrets, (1) cooling diminishes contralateral response to oriented gratings, but does not affect the response to small flashing squares, supporting the integrative role of callosal inputs and (2) a primary excitation between orientation matched responses and a secondary suppression between orthogonally oriented responses. In sum, they show that callosal interactions are higher order in nature; this is also consistent with anatomical results in cat (Tettoni et al. 1998). Our results parallel both of their findings, and suggest that the multipoint INS fiber array induced activation of contralateral (higher order) orientation domains.

References:

- Ferreiro, D. N., Conde-Ocazonez, S. A., Patriota, J. H., Souza, L. C., Oliveira, M. F., Wolf, F., & Schmidt, K. E. (2021). Spatial clustering of orientation preference in primary visual cortex of the large rodent agouti. *Iscience*, 24(1).
- Hu, J. M., Qian, M. Z., Tanigawa, H., Song, X. M., & Roe, A. W. (2020). Focal electrical stimulation of cortical functional networks. *Cerebral Cortex*, 30(10), 5532-5543.
- Hubel, D. H., & Wiesel, T. N. (1965). Receptive fields and functional architecture in two striated visual areas (18 and 19) of the cat. *Journal of neurophysiology*, 28(2), 229-289.
- Hubel, D. H., & Wiesel, T. N. (1977). Ferrier lecture-Functional architecture of macaque monkey visual cortex. *Proceedings of the Royal Society of London. Series B. Biological Sciences*, 198(1130), 1-59.

Response Figure #1-1 (now Manuscript Figure 1AB). Experimental scheme. (A) Perception of 90° contours induced by higher order illusory contours. a. single orientation b. mixed orientation c. abutting line illusory contour d. aligned points e. sequential aligned points. (B) Above: Schematic of ‘high order projection’ via callosal connection (Makarov 2008). Below: Predicted activation in right hemisphere following stimulation of multiple aligned domains in left hemisphere.

Makarov, V. A., Schmidt, K. E., Castellanos, N. P., Lopez-Aguado, L., & Innocenti, G. M. (2008). Stimulus-dependent interaction between the visual areas 17 and 18 of the 2 hemispheres of the ferret (*Mustela putorius*). *Cerebral Cortex*, 18(8), 1951-1960.

Roe, A. W., Chen, G., Lu, H. D., & Squire, L. R. (2009). Visual system: Functional architecture of area V2. *Encyclopedia of neuroscience*, 10, 331-349.

Song, Y., & Baker, C. L. (2006). Neural mechanisms mediating responses to abutting gratings: luminance edges vs. illusory contours. *Visual neuroscience*, 23(2), 181-199.

Tettoni, L., Gheorghita-Baechler, F., Bressoud, R., Welker, E., & Innocenti, G. M. (1998). Constant and variable aspects of axonal phenotype in cerebral cortex. *Cerebral cortex (New York, NY: 1991)*, 8(6), 543-552.

Tusa, R. J., Rosenquist, A. C., & Palmer, L. A. (1979). Retinotopic organization of areas 18 and 19 in the cat. *Journal of Comparative Neurology*, 185(4), 657-678.

R1-4. Overall, this is original, interesting work with that employed noteworthy techniques, but the relationship of the experiments conducted to VCPs is not well laid out. Ultimately, the manuscript is missing essential context for VCPs and electrical stimulation that are needed to make an appropriate comparison with INS or a valid argument for INS to be used for a VCP.

Response:

Thanks for your comments. We agree that context regarding electrical stimulation is needed. We now include a section in the Discussion on previous electrical stimulation studies and how this design differs. We explain the rationale for developing a VCP capable of fine scale functional activation of early visual cortex at columnar level.

Note that we have made efforts to compare the effectiveness and ease of use of electrical, optogenetic, and INS stimulation methods for the purpose of columnar VCP development. We have tried different electrical stimulation arrays (e.g. Utah array, Chernov et al. 2016; NeuroNexus array, Wang et al. 2013); however, we moved away from electrical approaches because the electrode array geometries are fixed and they caused tissue damage.

Response Figure #1-2. Left: List of Roe papers on electrical, optogenetic, and INS stimulation of single cortical columns. Red text: studies in monkeys. Black text: rats, cats or humans. Right: Study of columnar circuits using resting state fMRI, anatomical tracing, and electrophysiological cross correlation.

References:

Chernov, M. M., Chen, G., Torre-Healy, L. A., Friedman, R. M., & Roe, A. W. (2016). Microelectrode array stimulation combined with intrinsic optical imaging: A novel tool for functional brain mapping. *Journal of neuroscience methods*, 263, 7-14.

Wang, Z., Chen, L. M., Négyessy, L., Friedman, R. M., Mishra, A., Gore, J. C., & Roe, A. W. (2013). The relationship of anatomical and functional connectivity to resting-state connectivity in primate somatosensory cortex. *Neuron*, 78(6), 1116-1126.

R1-5. The authors did not properly cite articles to reflect the activation radius for electrical stimulation (Stoney1968, Asanuma 1976, Tehovnik 2006, Tehovnik 2007 or indicate the expected range for intracortical microstimulation (ICMS) (Chen 2020, Fernandez 2021, Bak 1990, Torab 2011, ...).

Response:

Thanks for making this important point. We apologize for failing to cite some important papers. We have now limited our discussion to stimulation of visual cortex and focus the discussion towards low levels of stimulation that might approximate column-specific stimulation (see *Previous Studies with Electrical Stimulation*). We now incorporate more of the relevant electrical stimulation literature (Tehovnik et al. 2006, Tehovnik et al. 2007, Schiller et al. 2011) this in the manuscript. There are a number of impressive VCP studies based on electrical stimulation. We were particularly motivated by Schmidt et al who conducted extensive study of a blind subject implanted with an electrical stimulator and examined psychophysical effects of single site stimulation (such as brightness, color, distance, duration, stability of the percept); these studies suggested that some feature specific percepts (such as color) could be achieved with low stimulation current levels. Beauchamp et al, 2020; Tehovnik's work, while very impressive, systematic, and quantitative, was focused largely on perception of phosphenes, without orientation or strong color percepts (Schiller et al. 2011). We considered multiple reasons for why electrical stimulation failed to activate the visual features well known to populate area V1: (1) overly high stimulus intensity leading to activation of a region of cortex that includes multiple functional domains and therefore lack of feature selectivity, (2) cortical area, where V1 has the smallest and most low level receptive fields, which alone potentially limits the complexity of perception, and (3) electrical stimulation without any visual stimulation (either from sensory stimuli or from internally generated imagery) may add to the 'unnaturalness' of the percept. Thus, our rationale for INS via fiber optic array: (1) INS because of the columnar specificity offered (no viral transfection needed), (2) area 18 (visual area V2) to examine lower to higher level integration, and (3) neuromodulation by adding stimulation to ongoing visual stimulation or internally generated imagery.

References:

- Beauchamp, M. S., Oswald, D., Sun, P., Foster, B. L., Magnotti, J. F., Niketeghad, S., ... & Yoshor, D. (2020). Dynamic stimulation of visual cortex produces form vision in sighted and blind humans. *Cell*, 181(4), 774-783.
- Schiller, P. H., Slocum, W. M., Kwak, M. C., Kendall, G. L., & Tehovnik, E. J. (2011). New methods devised specify the size and color of the spots monkeys see when striate cortex (area V1) is electrically stimulated. *Proceedings of the National Academy of Sciences*, 108(43), 17809-17814.
- Tehovnik, E. J., Toliás, A. S., Sultan, F., Slocum, W. M., & Logothetis, N. K. (2006). Direct and indirect activation of cortical neurons by electrical microstimulation. *Journal of neurophysiology*, 96(2), 512-521.
- Tehovnik, E. J., & Slocum, W. M. (2007). Phosphene induction by microstimulation of macaque V1. *Brain research reviews*, 53(2), 337-343.

R1-6. Fernandez 2021 reported an average threshold of $66.8 \pm 36.5 \mu\text{A}$ for single electrodes. This puts the estimated activation diameter to be in the range of 0.45 – 0.63 mm, depending on the K value used. The authors did not give exact measurements for the diameter of activation with INS (they should) for each energy level and only stated “submillimeter”. It would be useful to indicate diameter of activation for each energy level tested. Based on the figures, it would seem it is in the range of 0.5 – 1 mm.

Response:

Thanks for raising this question. As the reviewer states, our Figure 2 and 4 indicates that, with the stimulation parameters used, the diameter of INS stimulated activation diameter ranges from 200-500um. This is consistent with Monte Carlo simulations of these energies delivered via a 200um fiber (Response Figure #1-3. Thompson et al. 2012, 2013). These values thus are appropriate for the size of orientation columns in cat cortical area 18 (300-700um, Shumel and Grinvald 1996, Hubener et al. 1997).

We have now added the diameters of activation from 3 intensities in 3 cats (Response Figure #1-4, Manuscript Figure 2G). This data reveals the increasing size of activation with increasing stimulation intensity; note that across cats the absolute values differ due to the slightly different angles of the fiber array (from 45-60 deg) (Richter et al. 2013). To further strengthen the relationship between stimulation intensity and activation size. Also, activation diameters have been added each panel of Figure 4D. We have also previously shown, using INS (Cayce et al. 2014), optogenetics (Chernov et al. 2018), and electrical stimulation (Hu et al. 2020) in cats and monkeys, that the INS stimulation evokes response enhancement in the matched functional domains and relative suppression in non-matched domains, consistent with the known anatomy and functional specificity of columnar connectivity in V1. These provide strong supporting evidence for the specificity of stimulation and confinement of the effect to within single columns.

Response Figure #1-3. Monte Carlo simulation of single fiber INS stimulation at 3 radiant exposures: 0.3J/cm², 0.5J/cm², 0.7J/cm². Core diameter of the fiber: 200um, NA: 0.22. Size of activation is 330-420um in diameter (labelled in Figure).

Response Figure #1-4 (now Manuscript Figure 2G). Measured widths (average long and short axes) of INS activations at 3 intensities. Dot colors: Cats 1-3.

References:

Cayce, J. M., Friedman, R. M., Chen, G., Jansen, E. D., Mahadevan-Jansen, A., & Roe, A. W. (2014). Infrared neural stimulation of primary visual cortex in non-human primates. *Neuroimage*, 84, 181-190.

Chernov, M. M., Friedman, R. M., Chen, G., Stoner, G. R., & Roe, A. W. (2018). Functionally specific optogenetic modulation in primate visual cortex. *Proceedings of the National Academy of Sciences*, 115(41), 10505-10510.

Hu, J. M., Qian, M. Z., Tanigawa, H., Song, X. M., & Roe, A. W. (2020). Focal electrical stimulation of cortical functional networks. *Cerebral Cortex*, 30(10), 5532-5543.

Hübener, M., Shoham, D., Grinvald, A., & Bonhoeffer, T. (1997). Spatial relationships among three columnar systems in cat area 17. *Journal of Neuroscience*, 17(23), 9270-9284.

Richter, C. P., Rajguru, S., Stafford, R., & Stock, S. R. (2013, March). Radiant energy during infrared neural stimulation at the target structure. In *Photonic Therapeutics and Diagnostics IX* (Vol. 8565, pp. 649-655). SPIE.

Shmuel, A., & Grinvald, A. (1996). Functional organization for direction of motion and its relationship to orientation maps in cat area 18. *Journal of Neuroscience*, 16(21), 6945-6964.

Thompson, A. C., Wade, S. A., Brown, W. G., & Stoddart, P. R. (2012). Modeling of light absorption in tissue during infrared neural stimulation. *Journal of biomedical optics*, 17(7), 075002-075002.

Thompson, A. C., Wade, S. A., Pawsey, N. C., & Stoddart, P. R. (2013). Infrared neural stimulation: influence of stimulation site spacing and repetition rates on heating. *IEEE Transactions on Biomedical Engineering*, 60(12), 3534-3541.

R1-7. Based on published data, it does not seem that INS has a significant benefit in terms of activation radius compared to ICMS. However, INS does not puncturing the parenchyma like ICMS. Other benefits of INS over ICMS may also be reasonable and supported by data/literature and would be better used to argue for an INS approach for VCPs over ICMS rather than activation radius. This type of addition to the discussion would be relevant/welcome context if the authors wish to frame INS as an alternative approach to VCPs.

Response:

Thanks for this important comment. Indeed, this is why we have turned to INS. This approach can be used on the cortical surface and can, via an array, access different selected patterns of functional domains. We have shown that INS can be applied to the surface of the cortex to evoke (1) cortical responses mimicking a moving dot (e.g. Chernov et al. 2021), (2) mesoscale local and brainwide circuits in cats and monkeys mapped in ultrahigh field fMRI (Xu et al. 2019), and local functional connectivity in human cortex in vivo (Pan et al. 2023). We are currently conducting INS in visual cortex of awake behaving monkeys in the 7T fMRI so that we can simultaneously map the effects on visual discrimination curves and neural (BOLD) response curves of brainwide circuit nodes. The current study in cat is a precursor to the monkey studies showing proof of principle that array stimulation can achieve functionally selective higher order effects in other cortical areas.

References:

Chernov, M. M., Friedman, R. M., & Roe, A. W. (2021). Fiberoptic array for multiple channel infrared neural stimulation of the brain. *Neurophotonics*, 8(2), 025005-025005.

Pan, L., Ping, A., Schriver, K. E., Roe, A. W., Zhu, J., & Xu, K. (2023). Infrared neural stimulation in human cerebral cortex. *Brain Stimulation*, 16(2), 418-430.

Xu, A. G., Qian, M., Tian, F., Xu, B., Friedman, R. M., Wang, J., ... & Roe, A. W. (2019). Focal infrared neural stimulation with high-field functional MRI: a rapid way to map mesoscale brain connectomes. *Science advances*, 5(4), eaau7046.

R1-8. There seems to be a bit of a conflation between ICMS and electrical stimulation with macro surface electrodes in the discussion. ICMS tends to have very low current thresholds to evoke

phosphenes (20 – 200 μ A), macro surface electrodes (~2 mm) tend to be larger (~mA range, varies depending on whether blind or sighted). Mini surface electrodes tend to be in the middle (~0.5 mA), and micro EcoG electrodes may have thresholds between mini and micro electrodes.

Response:

Thanks for pointing this out. In the Discussion “*Previous studies with electrical stimulation*”, we now focus primarily on low current (<100uA) microelectrode stimulation. Use of surface electrodes in human studies is also mentioned and specified.

R1-9. The authors indicate VCPs have “yet to achieve the basic qualitative aspects of vision” yet cite in the following line an example of simple form vision delivered with electrical stimulation. However, there is greater missing context to this citation that is relevant to the discussion on VCPs. While the Chen 2020 paper did demonstrate some limited efficacy for sighted NHPs to perform a 2AFC, the electrodes used were far apart, and the NHPs could have performed the discrimination task based on low level features. Secondly, the Fernandez 2021 study (not cited by authors) delivers a follow-up to the Chen study with a human subject who was implanted with one Utah array in visual cortex. The results from Fernandez 2021 were complicated. While the researchers in that study were able to evoke perception of some simple characters in their blind subject, the results were inconsistent. Some patterns that would be expected to create a character didn’t, and other times unexpected spatial patterns results in characters. Sometimes electrical stimulation on neighboring electrodes resulted in perception of separate phosphenes, sometimes electrical stimulation on electrodes across the array resulted in perception of one phosphene. This is further discussed in a commentary response to the paper (Beauchamp 2021). A more substantive critique of ICMS than is presented in the manuscript is possible and could better be used to bolster INS as an alternative approach.

Response:

Thanks for this important point. We have carefully studied the Fernandez paper. It produces exactly what we would expect. That is, the array was implanted near the V1/V2 border. V1 contains organization for eye dominance, orientation, and color (blobs); V2 contains interleaved stripe organization (thin, pale, thick, pale), each of which has internal organization for hues and brightness, higher order orientation, and ocular disparity, respectively. Thus, within the 4mm Utah array expanse, many of these domains would have been sampled, each subserving the same or nearby locations in space but very different functionalities; it is quite likely that each domain functionality would have differing thresholds and requirements for perceptual activation. We now add this to the Discussion “*Previous studies with electrical stimulation*”.

Reference:

Fernández, E., Alfaro, A., Soto-Sánchez, C., Gonzalez-Lopez, P., Lozano, A. M., Peña, S., ... & Normann, R. A. (2021). Visual percepts evoked with an intracortical 96-channel microelectrode array inserted in human occipital cortex. *The Journal of clinical investigation*, 131(23).

R1-10. The authors appear to be presenting data that indicates INS delivered to in a linear location on one hemisphere of V1 results in activation in the other hemisphere, in a different part of the visual field, and in different functional representation. It is unclear how this relates to VCPs or illustrates higher order integration.

Response:

In both cats and monkeys, callosal connections link similar topographic locations at or near the vertical meridian. The callosal fibers in essence stitch together the two halves of the visual field (Innocenti et al. 1995). In addition to linking topographically matched points, callosal connections also mediate integration of inputs from the contralateral hemisphere to achieve a higher order functional integration (e.g. higher orientation: Makarov et al. 2008, Rochefort et al. 2009). This type of integration serves to strongly link and emphasize the information at the vertical meridian. This type of integration and accompanying overrepresentation of vertical is hypothesized to underlie perceptual enhancements at the vertical meridian (Coppola et al. 1998). There are also studies that indicate callosal interactions are important for binocular vision (Watroba et al. 2001). As evidenced by studies in split brain patients, callosal function is a critical aspect of vision that needs to be addressed in VCPs.

References:

Coppola, D. M., White, L. E., Fitzpatrick, D., & Purves, D. (1998). Unequal representation of cardinal and oblique contours in ferret visual cortex. *Proceedings of the National Academy of Sciences*, 95(5), 2621-2623.

Innocenti, G. M., Aggoun-Zouaoui, D., & Lehmann, P. (1995). Cellular aspects of callosal connections and their development. *Neuropsychologia*, 33(8), 961-987.

Makarov, V. A., Schmidt, K. E., Castellanos, N. P., Lopez-Aguado, L., & Innocenti, G. M. (2008). Stimulus-dependent interaction between the visual areas 17 and 18 of the 2 hemispheres of the ferret (*Mustela putorius*). *Cerebral Cortex*, 18(8), 1951-1960.

Rochefort, N. L., Buzás, P., Quenech'Du, N., Koza, A., Eysel, U. T., Milleret, C., & Kisvárdy, Z. F. (2009). Functional selectivity of interhemispheric connections in cat visual cortex. *Cerebral Cortex*, 19(10), 2451-2465.

Watroba, L., Buser, P., & Milleret, C. (2001). Impairment of binocular vision in the adult cat induces plastic changes in the callosal cortical map. *European Journal of Neuroscience*, 14(6), 1021-1029.

R1-11. Presentation of some of the raw imaging data would be useful.

Response:

Thanks for raising this question. The figure below (now Supplemental Fig. 2) illustrates the steps involved in standard OI imaging which includes collection of single condition maps (A), clipping of the grayscale range to remove large vascular-related reflection artifacts (B), generation of difference map (C), highpass spatial filtering to remove illumination gradients across the field of view and lowpass spatial filtering to remove very small pixelation (primarily for clarity) (D), vessel mask to remove big vessels. These are standard procedures in intrinsic signal optical imaging.

Response Figure #1-5 (now Manuscript Supplemental Figure 2). Steps in processing of OI data. (A) Single condition maps or horizontal (left) and vertical (right) gratings. (B) In the grayscale distribution of an image, we 'clip' the distribution to remove large artifactual (e.g. $dR/R > 2\%$) reflectances which are often due to large vascular pulsations or locations of specularly. A clip of median $\pm 2SD$ is typical. (C) Difference map of two single-condition maps. Dark: horizontal preferring. Light: vertical preferring. (D) Filtered map based on difference map, with both low and high pass (see text). (E) Following removal of large vessel pixels, the filtered map is clipped.

R1-12. Authors claim each of the following as novel aspects in the discussion, response to each is included. May be better to indicate these as features or attributes rather than calling them novel.

Response:

We agree with the reviewer. We now simply say "We summarize key aspects of this visual brain-machine interface design." We have also added limitations of this approach.

Limitations: While fiber optic designs can cause less tissue damage when used for cortical surface stimulation and can be placed on different locations on the brain, the current design must be removed when not in use, limiting its capacity for use in freely behaving monkey studies; however, in the future, it may be possible to engineer a u-LED stimulation design that is wireless (Zaraza et al. 2022, Kim et al. 2013, 2019, Zhang et al. 2019). A major limitation of fiber optic arrays compared to electrode array interfaces is that they are unable to, alone, record neuronal spike activity. Our design does incorporate neural signals via OI or fMRI; however, for direct spiking activity, we are now approaching this with a separate session of column-targeted electrical recording. In addition, unlike optogenetics, INS is not cell type specific, a limitation that could be overcome by conducting

column targeted optogenetic stimulation (Ruiz et al. 2013, Chernov et al. 2018). The current linear fiber design is useful for some applications but is limited in the spatial patterning. To increase flexibility of stimulation patterns, we have developed a 100-fiber bundle array and have applied this to cortex in anesthetized monkeys, with plans to do so in trained awake monkeys. Note that some multi-fiber designs can also be inserted to deep brain locations (Roe et al in prep).

References:

- Chernov, M. M., Friedman, R. M., Chen, G., Stoner, G. R., & Roe, A. W. (2018). Functionally specific optogenetic modulation in primate visual cortex. *Proceedings of the National Academy of Sciences*, 115(41), 10505-10510.
- Kim, T. I., McCall, J. G., Jung, Y. H., Huang, X., Siuda, E. R., Li, Y., ... & Bruchas, M. R. (2013). Injectable, cellular-scale optoelectronics with applications for wireless optogenetics. *Science*, 340(6129), 211-216.
- Kim, A., Zhou, J., Samaddar, S., Song, S. H., Elzey, B. D., Thompson, D. H., & Ziaie, B. (2019). An implantable ultrasonically-powered micro-light-source (μ light) for photodynamic therapy. *Scientific reports*, 9(1), 1395.
- Ruiz, O., Lustig, B. R., Nassi, J. J., Cetin, A., Reynolds, J. H., Albright, T. D., ... & Roe, A. W. (2013). Optogenetics through windows on the brain in the nonhuman primate. *Journal of neurophysiology*, 110(6), 1455-1467.
- Zaraza, D., Chernov, M. M., Yang, Y., Rogers, J. A., Roe, A. W., & Friedman, R. M. (2022). Head-mounted optical imaging and optogenetic stimulation system for use in behaving primates. *Cell Reports Methods*, 2(12).
- Zhang, Y., Castro, D. C., Han, Y., Wu, Y., Guo, H., Weng, Z., ... & Rogers, J. A. (2019). Battery-free, lightweight, injectable microsystem for in vivo wireless pharmacology and optogenetics. *Proceedings of the National Academy of Sciences*, 116(43), 21427-21437.

R1-13. Mesoscale stimulation: 1. This scale is not novel to electrical stimulation. ICMS at levels that can evoke phosphenes can be more focal than the levels demonstrated with INS. 2. This discussion also indicates the need for discriminable individual phosphenes. A “pixel” based approach is an outdated framework for VCPs.

Response:

We agree that the scale is not novel, but the column based approach is. Please see our response in R1-7 and R1-9 (regarding functional organization). We now explicitly summarize this point in the Discussion.

R1-14. Neuromodulation: 1. Neuromodulation is a type of neural stimulation, there is no role shift. 2. Using directional stimulation on cortex or delivering a linear pattern is not novel. 3. High levels of electrical stimulation can also inhibit spread of activation (Bosking 2017, Neuropace RNS) 4. This is not a “demonstration of functionally specific neural modulation”.

Response:

We agree that neuromodulation is a type of neural stimulation. 2. We have removed ‘novel’ from our descriptions. 3. We agree that stimulation can induce inhibition (e.g., as illustrated in Figure 7) and cite additionally Bosking et al. 2017 and Logothetis et al. 2010. 4. Our demonstration of higher order columnar integration due to external focal stimulation is very functionally specific.

References:

- Bosking, W. H., Beauchamp, M. S., & Yoshor, D. (2017). Electrical stimulation of visual cortex: relevance for the development of visual cortical prosthetics. *Annual review of vision science*, 3, 141-166.
- Logothetis, N. K., Augath, M., Murayama, Y., Rauch, A., Sultan, F., Goense, J., ... & Merkle, H. (2010). The effects of electrical microstimulation on cortical signal propagation. *Nature neuroscience*, 13(10), 1283-1291.

R1-15. Dynamics: 1. Delivering 10 different stimulation patterns is not novel. 2. Slow, fast, and random electrical stimulation have been used in other studies.

Response:

We have removed ‘novel’ and have added refer to these other studies (Oswalt et al. 2021, Beauchamp et al. 2020).

References:

- Beauchamp, M. S., Oswalt, D., Sun, P., Foster, B. L., Magnotti, J. F., Niketeghad, S., ... & Yoshor, D. (2020). Dynamic stimulation of visual cortex produces form vision in sighted and blind humans. *Cell*, 181(4), 774-783.
- Oswalt, D., Bosking, W., Sun, P., Sheth, S. A., Niketeghad, S., Salas, M. A., ... & Yoshor, D. (2021). Multi-electrode stimulation evokes consistent spatial patterns of phosphenes and improves phosphene mapping in blind subjects. *Brain stimulation*, 14(5), 1356-1372.

R1-16. Intensity dependence and general suppression: Changing effects with intensity of stimulation are not a novel demonstration.

Response:

Thanks for pointing this out. We have removed ‘novel’ and modified the description in manuscript accordingly.

R1-17. Higher order response:

The authors state “Here, presenting a sequence of point stimuli, which can induce the percept of motion[18] as well as lower or higher order shape[29,46-50]”.

No demonstration of perception induced by INS was presented in this manuscript.

The authors state “our linear array fiber tip locations targeted different orientation domains in one hemisphere”. This was not clearly demonstrated. Activation seems to be in ~0.5-1 mm chunks and not in specific orientation columns, could use a clear demonstration of this activation.

Linear patterns resulted on contralateral activation while random did not. This would seem to indicate the tissue is responding to the linear spatiotemporal pattern more so than activation of specific orientation columns.

Response:

The reviewer is correct that we have not provided evidence of inducing perception, but have referred to other studies that relate higher order contour perception to similar cortical responses (Chen et al. 2020). The difference here is in the targeting of columns and the spacing (which only spans 1-2 deg). Cat orientation domains are about 0.5mm in size. If one moves an electrode 0.5mm across the cortex, it will record neurons from a distinct orientation domain and have different orientation tunings (Shmuel and Grinvald 1996; Hubener et al. 1997). In our experiment, each optic fiber was targeted to a different orientation domain (90°, 0°, 135°, 100°, 20°), but the response transmitted to contralateral cortex generated a selective (enhancement and suppression) oriented response; this is an example of a “higher order effect”. Here, using INS stimulation, we have replicated the effect by Makarov et al (Makarov et al. 2008) using visual stimulation to show that area 18 receives callosal inputs to generate higher order orientation response.

References:

Chen, X., Wang, F., Fernandez, E., & Roelfsema, P. R. (2020). Shape perception via a high-channel-count neuroprosthesis in monkey visual cortex. *Science*, 370(6521), 1191-1196.

Hübener, M., Shoham, D., Grinvald, A., & Bonhoeffer, T. (1997). Spatial relationships among three columnar systems in cat area 17. *Journal of Neuroscience*, 17(23), 9270-9284.

Makarov, V. A., Schmidt, K. E., Castellanos, N. P., Lopez-Aguado, L., & Innocenti, G. M. (2008). Stimulus-dependent interaction between the visual areas 17 and 18 of the 2 hemispheres of the ferret (*Mustela putorius*). *Cerebral Cortex*, 18(8), 1951-1960.

Shmuel, A., & Grinvald, A. (1996). Functional organization for direction of motion and its relationship to orientation maps in cat area 18. *Journal of Neuroscience*, 16(21), 6945-6964.

R1-18. Linking contralateral visual hemifield: It is not clear how this is relevant to a VCP.

Response:

Please see **R1-2**, **R1-7**, and **R1-9**.

R1-19. Lines 253 - 254. The scale of activation presented is closer to hypercolumn scale than single column scale.

Response:

The size of cat orientation column in area 18 is bigger than that in monkey and is about 500 um. The cat hypercolumns is 1-2mm (Hubener et al., 1997). Please see Fig 2G, the size of activation is ~0.3mm in diameter, so the activation is limited to a single column.

Reference:

Hübener, M., Shoham, D., Grinvald, A., & Bonhoeffer, T. (1997). Spatial relationships among three columnar systems in cat area 17. *Journal of Neuroscience*, 17(23), 9270-9284.

R1-20. Lines 278 – 291. Feature specific functional activation would not be possible at the scale of the Orion electrodes and did not seem to be a goal given the interface. Beauchamp 2020 did use “form from motion” to evoke perception of simple characters. The authors here used similarly structured presentation of INS to yield their evoked responses.

Response:

‘Form from motion’ percepts can be achieved by stimulating topographically positioned points, such as perceptions of letters and shapes 5-10 deg in size (Beauchamp et al. 2020) or 2-4 deg in size (Chen et al. 2020). Our study also achieved this, but, distinct from previous studies, the integrated result (1) illustrated the circuit by which this was achieved from one cortical area to another (Response Figure #1-1B), (2) identified the cortical functional columnar basis of this effect, i.e. showed that the resulting orientation selective result involved both enhancement of vertical

(horizontal) and suppression of horizontal (vertical) domains and (3) showed that the integrated orientation-selective response was invariant across the multiple types of input orientations. Thus, this study demonstrates an example of a known circuit by which a VCP induces higher order percepts.

References:

Beauchamp, M. S., Oswalt, D., Sun, P., Foster, B. L., Magnotti, J. F., Niketeghad, S., ... & Yoshor, D. (2020). Dynamic stimulation of visual cortex produces form vision in sighted and blind humans. *Cell*, 181(4), 774-783.

Chen, X., Wang, F., Fernandez, E., & Roelfsema, P. R. (2020). Shape perception via a high-channel-count neuroprosthesis in monkey visual cortex. *Science*, 370(6521), 1191-1196.

R1-21. Line 286. Odd usage of “feelings” , should use perception.

Response:

We have revised this as suggested.

R1-22. Lines 287 - 291. The authors reverence “knowledge about the precise stimulated column or specific functional neural response” but don’t expand on the what this means in context

Response:

This refers to our work on stimulation of single columns in primate visual cortex and single digit sites in primate somatosensory cortex. We have shown using electrical, optogenetic, and infrared neural stimulation that such single column stimulation activates functionally specific columnar circuits. This work has now extended from the local (intrinsic cortical connections) to the global (brainwide mesoscale networks) scale. Please see our response above in **R1-4**. We have now revised this to clarify its meaning.

R1-23. Lines 307 - 308. The authors state “Thus, our BMI is capable of activating callosal circuits in a manner very similar to visual stimulation” but this does not seem to be fully supported by their data.

Response:

Please see our response above in **R1-3**.

Reviewer#2 (Remarks to the Author):

The current study introduced a new approach to visual cortical prosthetics (VCP), using pulsed infrared neural stimulation (INS) in cat V2 that achieves the spatial precision required to interact with small cortical columns responsible for visual feature perception. Tian and colleagues used an optical fiber array, producing cortical activation similar to a moving visual dot (the trajectory of a point of light is defined as contour orientation). The stimulation enhances responses in orientation-matched domains and reduces responses in non-matched ones in contralateral V2, with the effects varying based on the speed, direction, and pattern of stimulation, emphasizing the orientation-selective nature of this approach. The story sounds interesting. However, my enthusiasm for this research has been hampered by the following shortcomings:

Major:

R2-1. The most crucial technical indicator for the development of visual cortex prostheses (VCP) is to acquire information about the subjective perceptual experiences of the subjects in response to the INS or electrical stimuli. This includes aspects such as the presence or absence of phosphenes and the location, size, shape, and color of a presented phosphene. In this article, the use of anesthetized cats seems inadequate for obtaining behavioral reports of subjective perceptual experiences elicited by INS stimuli. In contrast, previous research with awake macaques has made it possible to assess the subjective perceptual experience of cortical stimuli using the macaque eye movement selection paradigm (Chen et al., 2020 *Science*). It would be worthwhile to repeat the INS stimulation experiments with awake cats and use a similar behavioral paradigm as in macaques to assess whether INS stimulation can elicit phosphenes. Otherwise, the current findings may have limited applicability in the development of VCPs.

Chen, X., Wang, F., Fernandez, E., & Roelfsema, P. R. (2020). Shape perception via a high-channel-count neuroprosthesis in monkey visual cortex. *Science*, 370(6521), 1191-1196.

Response:

Thanks for raising this question. Reviewer 2 asked that whether INS induces any perceptual

experience. While we have not yet conducted INS-induced orientation selective perception, we show below that INS applied to V4 in a known visuotopic location induces phosphenes to which a monkey makes accurate and reliable saccades. We first determined the visual map in V4 by conducting optical imaging in response to a small (1°) grating patch in an awake fixating macaque monkey (Response Figure #2-1, dotted lines, cf. Tanigawa et al. 2010). We predicted that placing the INS fiber optic on a specific location on the cortex would induce a phosphene that would elicit a saccade to the corresponding location in visual space (panel A). We chose V4 as a starting point because receptive fields there are larger, potentially making a single site stimulation more visible.

Response Figure #2-1. (A) Left: location of chronic optical window implanted over visual cortex. Is, lunate sulcus. sts, superior temporal sulcus. A, anterior; D, dorsal. Middle: View of V1, V2, V4 in window. White dashed lines: iso-eccentricities $0.5^\circ, 1.5^\circ, 2.5^\circ, 3.5^\circ$; yellow lines: iso-polarities $22^\circ, 45^\circ, 67^\circ$. Right: Fiber optic (white arrow) and fiber tip (black arrow) targets map near 3.5° isoeccentricity and 22° - 45° isopolarity in V4. (B-C) Eye saccades induced by infrared neural stimulation from two example trials. (D) An example trial without INS stimulation. Gray shading: visual field corresponding to the INS stimulated cortex. (E) The average eye positions for all 37 trials with INS stimulation. x (upper) and y (bottom) positions, gray shadings: SD. Latency to onset of eye saccade from INS stimulation onset: 300ms. (F) Average of 37 INS induced eye saccades in the visual field.

We compared the monkey's eye saccade response to the presence and absence of INS stimulation (panel B). In both types of trials, the monkey was rewarded for maintaining good fixation (3.5 sec) on a small fixation point. After fixation point offset, fluid reward was delivered (0.3 sec), and then INS stimulation was presented for 0.5 s. Our expectation was that, if INS induces visual phosphenes, the monkey would (1) make a saccade, (2) saccade in the direction towards the visual location corresponding to the cortical activation spot, (3) saccade reliably to the visual location represented by the activated site, and (4) saccade with a latency to target of about 300 ms (i.e., the time of a typical eye saccade, Oz et al. 2023). In addition, (5) absence of INS stimulation would not result in an eye movement biased towards any single direction.

Panels B-C show examples of 2 trials with INS. INS was delivered through a sterile 200 μm fiber (white arrow in panel A) that was placed on the visual cortex (red dot). Placement of the fiber tip was guided with the help of a red-light laser guide (black arrow in panel A). INS stimulation, with parameters described in this manuscript (1875nm wavelength, 200 Hz, pulse width 200 μs , pulse train 500ms, and a radiant exposure of 0.6 J/cm²), was applied to a cortical location in V4 representing isopolar angle 40° and isoeccentricity 3.5°. As shown in panels B and C, the eye saccaded to a location in the lower right quadrant of the visual field, with isopolar coordinate between polar angle 22.5° and 45° (endpoint polar angle of 30.4°) and eccentricity between 2.5° and 3.5°. Panel D shows the result of one trial without INS stimulation. All trials (both INS and no INS) were rewarded at offset of fixation spot.

Panels E and F illustrates the averaged eye position in the x and y directions of all 37 INS trials within 1.5 s after fixation point offset. As revealed by the eye positions in x direction (upper panel in E, black line, gray shading: SD) and in y direction (bottom panel in E, black line, gray shading: SD), the monkey maintained good fixation on the fixation point (see the x and y lines during the fixation ON period, panel E). After the offset of the fixation point, INS stimulation was applied for 0.5 sec, resulting in initiation of a saccade in 300 ms. On average, the monkey made saccades with a direction of 40.4° polar angle and 2.5°-3.5° isoeccentricity, consistent with the visuotopic location of the INS fiber in V4.

Our results suggest that INS applied to visual cortex induces visual phosphenes, leading to predicted saccade behavior in the awake, behaving monkey. These findings demonstrate feasibility for INS to be used as a cortical stimulation method that can be incorporated into visual cortical prosthetics. Combined with our aim of targeting cortical columns, we aim to enhance or restore spatially and featurally specific visual percepts.

Similar to these results, previous studies have also evoked saccades using electrical stimulation or optogenetics (Tehovnik et al. 2005, Jazayeri et al. 2012, Oz et al. 2023).

Reference:

Jazayeri, M., Lindbloom-Brown, Z., & Horwitz, G. D. (2012). Saccadic eye movements evoked by optogenetic activation of primate V1. *Nature neuroscience*, 15(10), 1368-1370.

Oz, R., Edelman-Klapper, H., Nivinsky-Margalit, S., & Slovlin, H. (2023). Microstimulation in the primary visual cortex: activity patterns and their relation to visual responses and evoked saccades. *Cerebral Cortex*, 33(9), 5192-5209.

Tanigawa, H., Lu, H. D., & Roe, A. W. (2010). Functional organization for color and orientation in macaque V4. *Nature neuroscience*, 13(12), 1542-1548.

Tehovnik, E. J., Slocum, W. M., Carvey, C. E., & Schiller, P. H. (2005). Phosphene induction and the generation of saccadic eye movements by striate cortex. *Journal of neurophysiology*, 93(1), 1-19.

R2-2. I am concerned about the potential for tissue damage from INS, as this could limit the applicability of the INS technique in VCPs. In the Introduction section, the authors assert that INS is a non-damaging approach. However, the fundamental principle of INS is the ability of infrared light to cause a rapid, localized temperature increase within neural tissues. This swift increase in temperature leads to a transient change in the cell membrane's capacitance, which can trigger action potentials. If the authors aim to highlight that INS stimulation is less traumatic than electrical stimulation, they should at least compare the near-threshold intensities of INS and electrical stimulation when the stimulation produces a reportable phosphene. A simpler method might involve in vitro measurements of temperature changes. Histological results would provide even stronger evidence.

Moreover, in Beauchamp et al., 2020 *Cell*, dynamic electrical stimulation was shown to generate phosphenes at efficiencies of up to 30 forms per minute. If similar efficiencies can be achieved, would INS cause more damage than electrical stimulation?

Beauchamp, M. S., Oswalt, D., Sun, P., Foster, B. L., Magnotti, J. F., Niketeghad, S., ... & Yoshor, D. (2020). Dynamic stimulation of visual cortex produces form vision in sighted and blind humans. *Cell*, 181(4), 774-783.

Response:

Thanks for asking this question. We and others have conducted temperature and damage threshold studies. Findings indicate that, for stimulation paradigms similar to that described here, thresholds, estimated from histological studies fall between $0.6\text{J}/\text{cm}^2 - 1\text{J}/\text{cm}^2$ (rat cortex $0.4\text{J}/\text{cm}^2$ and nonhuman primate cortex $\sim 0.6\text{J}/\text{cm}^2$ [Chernov et al. 2014], human cortex $0.6\text{J}/\text{cm}^2$ [Pan et al. 2023], human spinal roots $1.09\text{J}/\text{cm}^2$ [Cayce et al. 2015], cochlea $25\mu\text{J}/\text{pulse}$ [Goyal et al. 2012]).

Studies have also examined the temperature rise due to INS. Thompson et

Response Figure #2-2. MR thermometry from ex vivo rat brain. A. Rat brain (white arrow) embedded in agar (yellow arrow) with $200\ \mu\text{m}$ INS optic fiber inserted (red arrow). B. Anatomical image of the rat brain, showing optic fiber location. C. INS stimulation ($1\ \text{J}/\text{cm}^2$) induces phase shifts at fiber tip. Numbers are the peak temperature change values of different voxels at the fiber tip calculated from the measured phase shifts. E. Temperature changes calculated from measured phase shifts. Each point is one measurement at one INS intensity. The evolution of the temperature change at peak voxels show initial rise over 6-8 pulse trains followed by plateau and rapid fall off at offset of stimulation (63 sec timepoint). Curves show intensity dependence (see color code in inset). Note maximum temperature rise at highest INS intensity does not exceed 2°C . Black line: single pulse train paradigm. At cessation of pulse trains, the temperature falls off rapidly. From [Xi et al, 2023, accepted].

al. modelled temperature rise in rat peripheral nerve and found that 250Hz stimulation with comparable parameters ($200\mu\text{m}$ fiber, pulse width $100\mu\text{sec}$, 1850nm , $25\mu\text{J}$) produced a temperature rise of 2.3°C (Thompson et al. 2013). We have also addressed this issue using MRI thermometry (which measures the shift in proton resonance frequency caused by temperature increase). Using the same parameters used in this study for INS stimulation in ex vivo rat brains (Response Fig. #2-2AB), MRI thermometry, reveals that the spatial extent of temperature increase is quite confined (Response Fig. #2-2C, a single to several voxels) and the highest ΔT (measured at the strongest voxel in response to the highest intensity of $1\text{J}/\text{cm}^2$, red line) plateaus below 2°C (for optogenetic stimulation see Luo et al. 2023). Importantly, after the session of pulse trains, the temperature immediately falls to less than 0.5°C within 3sec. This demonstrates that, with these parameters, INS-induced temperature increase is dissipated rapidly (Response Figure #2-2E, black vertical line). In sum, assessments of heat induced damage via histological methods, thermometry, and modelling, all indicate that INS delivered with these parameters are non-damaging. Specifically, the temperature rise for 0.1 , 0.3 , and $0.5\ \text{J}/\text{cm}^2$ does not exceed 0.3°C and even at the highest value of $1\text{J}/\text{cm}^2$ does not exceed 1°C (first dot 'after first pulse train' of the 20 dots on graph), suggesting all stimulation conditions conducted here (Manuscript Figure 3) were non-damaging (the single and longest pulse train was 500msec delivered once per $>10\text{sec}$).

Note also that, based on these studies, we have conducted INS in human cortex (Pan et al. 2023) and there are currently clinical trials using INS (cochlea: Richter NCT05110183, peripheral nerves: Jansen NCT04601337).

References:

- Chernov, M. M., Chen, G., & Roe, A. W. (2014). Histological assessment of thermal damage in the brain following infrared neural stimulation. *Brain Stimulation*, 7(3), 476-482.
- Goyal, V., Rajguru, S., Matic, A. I., Stock, S. R., & Richter, C. P. (2012). Acute damage threshold for infrared neural stimulation of the cochlea: functional and histological evaluation. *The Anatomical Record: Advances in Integrative Anatomy and Evolutionary Biology*, 295(11), 1987-1999.
- Jonathan M. Cayce, Jonathon D. Wells, Jonathan D. Malphrus, Chris Kao, Sharon Thomsen, Noel B. Tulipan, Peter E. Konrad, E. Duco Jansen, and Anita Mahadevan-Jansen (2015) Infrared neural stimulation of human spinal nerve roots in vivo. *Neurophotonics* 2(1), 015007
- Luo, H., Yang, Z., Yang, P. F., Wang, F., Reed, J. L., Gore, J. C., ... & Chen, L. M. (2023). Detection of laser-associated heating in the brain during simultaneous fMRI and optogenetic stimulation. *Magnetic Resonance in Medicine*, 89(2), 729-737.
- Pan, L., Ping, A., Schriver, K. E., Roe, A. W., Zhu, J., & Xu, K. (2023). Infrared neural stimulation in human cerebral cortex. *Brain Stimulation*, 16(2), 418-430.
- Thompson, A. C., Wade, S. A., Pawsey, N. C., & Stoddart, P. R. (2013). Infrared neural stimulation: influence of

stimulation site spacing and repetition rates on heating. *IEEE Transactions on Biomedical Engineering*, 60(12), 3534-3541.

Xi Y*, Shrivastava D*, Tian J, Xu AG, Xi W, Roe AW**, Schriver KE**, and Zhang XT** (2023) Quantifying Tissue Temperature Changes Induced by Infrared Neural Stimulation: Numerical Simulation and MR Thermometry. *Biomed Optics Express*, Provisionally accepted.

R2-3. The authors did not clarify in the Results section whether there is a difference between the forward and reverse directions of the sequential INS stimuli. However, Figure S1 clearly shows different effects between the two directions of stimulation. This is confusing because, according to the authors' hypothesis, the forward and reverse direction of the sequential INS stimuli should produce stimuli in the same contour orientation in the cat's cortex, thus having the same effect on the contralateral cortex.

Response:

Thanks for raising this question. We do not find it surprising that there are differences.

Differences between forward and reverse directions may be due to the domain from which the sequence is initiated. Such sequence dependent effects have been observed to impact psychophysical perception (see Fischer et al. 2014, Motala et al. 2020, Salas et al. 2022). One possible rationale of this phenomenon is that primacy facilitates perceptual stability, perhaps via subthreshold effects (Jancke et al. 2004, Bosking et al. 2000, Sincich and Blasdel 2001). An alternative explanation relates to perceptual bias during centripetal vs centrifugal directions of motion (e.g. in situations of self-motion, Harris et al. 1981). Thus, in motion sensitive areas such as area 18, sequential stimulation may need to take into account such biases in cortical representation. Such differences may be important to study for VCP.

References:

Bosking, W. H., Kretz, R., Pucak, M. L., & Fitzpatrick, D. (2000). Functional specificity of callosal connections in tree shrew striate cortex. *Journal of Neuroscience*, 20(6), 2346-2359.

Fischer, J., & Whitney, D. (2014). Serial dependence in visual perception. *Nature neuroscience*, 17(5), 738-743.

Harris, L. R., Morgan, M. J., & Still, A. W. (1981). Moving and the motion after-effect. *Nature*, 293(5828), 139-141.

Jancke, D., Chavane, F., Naaman, S., & Grinvald, A. (2004). Imaging cortical correlates of illusion in early visual cortex. *Nature*, 428(6981), 423-426.

Motala, A., Zhang, H., & Alais, D. (2020). Auditory rate perception displays a positive serial dependence. *I-Perception*, 11(6), 2041669520982311.

Salas, M. A., Bell, J., Niketeghad, S., Oswald, D., Bosking, W., Patel, U., ... & Pouratian, N. (2022). Sequence of visual cortex stimulation affects phosphene brightness in blind subjects. *Brain Stimulation*, 15(3), 605-614.

Sincich, L. C., & Blasdel, G. G. (2001). Oriented axon projections in primary visual cortex of the monkey. *Journal of Neuroscience*, 21(12), 4416-4426.

R2-4. Another question that arises is why the authors did not record the response on the same side as the INS stimulus, which could arguably be more direct. An iso-orientation surround suppression effect, but not a cross-orientation suppression effect, should be generated between neurons with adjacent receptive fields. The cross-orientation suppression effect in the current study is supported by fMRI experiments on human subjects (ref28, Brouwer & Heeger, 2008). However, electrophysiological data usually show iso-orientation suppression. Are there any electrophysiological data that support the cross-orientation suppression effect?

Response:

Thanks for raising this question. (1) Why we did not record from the same side: Technically, it was spatially easier to place the optical imaging camera away from the INS stimulation array and not be concerned about the camera bumping into the camera. The glass array itself could lead to reflection artifacts that interfered with evaluation of the cortical reflectance signals. (2) The iso-orientation suppression arises from stimulating the surround region of an oriented receptive field (Blakemore and Tobin 1972, Fries et al. 1977, Nelson and Frost 1978, Allman et al. 1985). This surround suppression is distinct because the surround is being stimulated; this situation might be analogous to simultaneously stimulating a vertical domain in the center and multiple nearby vertical domains in the surround, resulting in the surround effect outbalancing the central activation. In our case, we are focally stimulating the center only. This central stimulation results in enhancement of like oriented domains and relative suppression of orthogonal domains, consistent with previous studies following stimulation of single orientation columns in monkey V1 (Chernov et al. 2018) and cat area 18 (Hu et al. 2020). Thus, similar circuits effects are present in intra-areal and inter-areal interactions. Other electrophysiological studies also support a cross-orientation suppression effect (Allison et al. 2001, Bishop et al. 1973, DeAngelis et al. 1992, Morrone et al. 1982).

References:

- Allison, J. D., Smith, K. R., & Bonds, A. B. (2001). Temporal-frequency tuning of cross-orientation suppression in the cat striate cortex. *Visual neuroscience*, 18(6), 941-948.
- Allman, J., Miezin, F., & McGuinness, E. (1985). Stimulus specific responses from beyond the classical receptive field: neurophysiological mechanisms for local-global comparisons in visual neurons. *Annual review of neuroscience*, 8(1), 407-430.
- Blakemore, C., & Tobin, E. A. (1972). Lateral inhibition between orientation detectors in the cat's visual cortex. *Experimental brain research*, 15, 439-440.
- Chernov, M. M., Friedman, R. M., Chen, G., Stoner, G. R., & Roe, A. W. (2018). Functionally specific optogenetic modulation in primate visual cortex. *Proceedings of the National Academy of Sciences*, 115(41), 10505-10510.
- DeAngelis, G. C., Robson, J. G., Ohzawa, I., & Freeman, R. D. (1992). Organization of suppression in receptive fields of neurons in cat visual cortex. *Journal of Neurophysiology*, 68(1), 144-163.
- Fries, W., Albus, K., & Creutzfeldt, O. D. (1977). Effects of interacting visual patterns on single cell responses in cat's striate cortex. *Vision research*, 17(9), 1001-1008.
- Henry, G. H., Bishop, P. O., Tupper, R. M., & Dreher, B. (1973). Orientation of specificity and response variability of cells in the striate cortex. *Vision research*.
- Hu, J. M., Qian, M. Z., Tanigawa, H., Song, X. M., & Roe, A. W. (2020). Focal electrical stimulation of cortical functional networks. *Cerebral Cortex*, 30(10), 5532-5543.
- Morrone, M. C., Burr, D. C., & Maffei, L. (1982). Functional implications of cross-orientation inhibition of cortical visual cells. I. Neurophysiological evidence. *Proceedings of the Royal Society of London. Series B. Biological Sciences*, 216(1204), 335-354.
- Nelson, J. I., & Frost, B. J. (1978). Orientation-selective inhibition from beyond the classic visual receptive field. *Brain research*, 139(2), 359-365.
- RM, H. G. B. P. T., & Dreher, B. (1973). Orientation specificity and response variability of cells in the striate cortex. *Vision res*, 13, 17711779.

R2-5. The authors' method of defining the orientation of the trajectory involves measuring the boundaries of V1 and V2 first and then inferring them from the brain atlas. It would be more convincing if receptive fields were measured before applying INS stimulation experiments.

Response:

Thanks for raising this question. The optical imaging method for determining the boundary of area 17 and 18 (and in monkey V1/V2) is a reliable and effective way to quickly reveal the border (Bonhoeffer et al. 1995; Hung et al. 2001; Li et al. 2017). Our fiber array was placed parallel to the border which is known to represent the vertical meridian (Tusa et al. 1978, 1979); this is quite straightforward way to determine the orientation. Receptive field mapping can be used to determine the locations of fiber placement; however, the presence of significant receptive scatter in such a small 1-2 degree area of area 18 makes determining the border less certain. In addition, optical imaging reveals the entire vertical meridian representation, thereby offering flexibility in array placement.

References:

- Bonhoeffer, T., Kim, D. S., Maloney, D., Shoham, D., & Grinvald, A. (1995). Optical imaging of the layout of functional domains in area 17 and across the area 17/18 border in cat visual cortex. *European Journal of Neuroscience*, 7(9), 1973-1988.
- Hung, C. P., Ramsden, B. M., Chen, L. M., & Roe, A. W. (2001). Building surfaces from borders in Areas 17 and 18 of the cat. *Vision research*, 41(10-11), 1389-1407.
- Li, Z., Meng, J., Li, H., Jin, A., Tang, Q., Zhu, J., & Yu, H. (2017). The feature-specific propagation of orientation and direction adaptation from areas 17 to 21a in cats. *Scientific Reports*, 7(1), 390.
- Tusa, R. J., Palmer, L. A., & Rosenquist, A. C. (1978). The retinotopic organization of area 17 (striate cortex) in the cat. *Journal of Comparative Neurology*, 177(2), 213-235.
- Tusa, R. J., Rosenquist, A. C., & Palmer, L. A. (1979). Retinotopic organization of areas 18 and 19 in the cat. *Journal of Comparative Neurology*, 185(4), 657-678.

R2-6. There are two concepts of orientation in the text that can be confusing. One refers to the orientation preference of the column itself, and the other to the contour orientation of the sequential INS, which stimulates the generation of an imaginary slash. It would be more precise to use the terms 'column orientation' and 'contour orientation' respectively.

Response:

We thank and accept this suggestion. We now use the suggested terms 'column orientation' and 'contour orientation' in the manuscript.

R2-7. I am not sure whether this cross-hemisphere neuromodulation in V2 is generalizable to humans and non-human primates. The authors' findings regarding callosal connections between the left and right V2 (ref 32-36) are based on studies in ferrets and cats. Are there previous findings in

humans or non-human primates that support this?

Response:

Yes, there are multiple studies in nonhuman primates and humans on topography and organization of callosal connectivity. For example, Kennedy et al. and Abel et al. revealed the topographic organization of the callosal connection of V2 in macaque (Kennedy et al. 1986, Abel et al. 2000). A review from Schmidt pointed out that not only in cats or ferrets, visual callosal connection also have similar functions in humans and non-human primates, forming clusters with neurons in the same hemisphere and establishing connections with neurons with similar orientation preferences (Schmidt 2013). Perhaps most relevant is a study from Manabu Tanifuji's group in which they used optical imaging and optogenetic stimulation to study callosal connectivity of macaque monkey V1/V2 border (Nakamichi et al. 2019). Focal stimulation produced submillimeter patchy activations in topographically similar locations on the contralateral side. Although to our knowledge, there are no parallel stimulation studies revealing higher order response, it is well established that monkey and human extrastriate cortex exhibit illusory contour responses (e.g. monkey: Peterhans and von der Heydt 1989, Ramsden et al. 2001; human: Montaser-Kouhsari et al. 2007, de Haas and Schwarzkopf. 2018). In monkeys, in particular, the thick and pale stripes contain higher order functional domains that respond to 'illusory contours' (Roe et al. 2009).

References:

- Abel, P. L., O'Brien, B. J., & Olavarria, J. F. (2000). Organization of callosal linkages in visual area V2 of macaque monkey. *Journal of Comparative Neurology*, 428(2), 278-293.
- de Haas, B., & Schwarzkopf, D. S. (2018). Spatially selective responses to Kanizsa and occlusion stimuli in human visual cortex. *Scientific reports*, 8(1), 611.
- Kennedy, H., Dehay, C., & Bullier, J. (1986). Organization of the callosal connections of visual areas V1 and V2 in the macaque monkey. *Journal of Comparative Neurology*, 247(3), 398-415.
- Montaser-Kouhsari, L., Landy, M. S., Heeger, D. J., & Larsson, J. (2007). Orientation-selective adaptation to illusory contours in human visual cortex. *Journal of Neuroscience*, 27(9), 2186-2195.
- Nakamichi, Y., Okubo, K., Sato, T., Hashimoto, M., & Tanifuji, M. (2019). Optical intrinsic signal imaging with optogenetics reveals functional cortico-cortical connectivity at the columnar level in living macaques. *Scientific Reports*, 9(1), 6466.
- Peterhans, E., & von der Heydt, R. (1989). Mechanisms of contour perception in monkey visual cortex. II. Contours bridging gaps. *Journal of Neuroscience*, 9(5), 1749-1763.
- Ramsden, B. M., Hung, C. P., & Roe, A. W. (2001). Real and illusory contour processing in area V1 of the primate: a cortical balancing act. *Cerebral Cortex*, 11(7), 648-665.
- Roe, A. W., Chen, G., Lu, H. D., & Squire, L. R. (2009). Visual system: Functional architecture of area V2. *Encyclopedia of neuroscience*, 10, 331-349.
- Schmidt, K. E. (2013). The visual callosal connection: a connection like any other? *Neural plasticity*, 2013.

R2-8. In the Results section, the statement “Under these higher fluence conditions, we observed a reduction in the OI reflectance signal irrespective of the orientation of the fiber optic array” needs to be accompanied by statistical test results.

Response:

Thanks for pointing this out. We have now added bar graphs for each of the vertical and horizontal timecourse comparisons in Case 1 (A, B) and Case 2 (C, D) to show the statistical significance in Figure 7. This shows that, at higher intensity (0.7J/cm²), both horizontal and vertical domains are suppressed by the array stimulation, in both positions of the array.

Response Figure #2-3 (now Manuscript Figure 7). Reduction of OI response at high INS intensity. (A-B) Upper row is the time course of OI signal for three vision / INS stimulation conditions, measures at ROIs in vertical (A) and horizontal (B) domains shown in Fig.5A in manuscript. Bottom row is the average reflectance changes during 2.5-5s of the time courses in the upper row. ***: $P < 0.001$, *: $P < 0.1$ (Welch's t-test, vision vs. vision + slow/fast INS). (C) The location and direction of fiber array placement in case 1. (D-E) Upper row is the time course of OI signal for three vision / INS stimulation conditions (Vertical, D vs. Horizontal, E domains shown in Fig.5A in manuscript). Bottom row is the average reflectance changes during 2.5-5s of the time courses in the upper row. (F) The location and orientation of the fiber array placement in case 2. Radiant exposure of INS in this figure: 0.7 J/cm^2 . Gray bars at x-axis: laser ON. Error bar: SEM

R2-9. The manuscript addresses two distinct concepts of orientation: the orientation preference of different columns in V2, and the contour orientation induced by sequential multi-fiber stimulation. However, the same term 'orientation' is used for both concepts in the manuscript, leading to confusion. It would be clearer to distinguish between them using terms like 'orientation' and 'contour orientation'.

Response:

Please see response to **R2-6**.

R2-10. Minor points:

1. The full form of the abbreviation "BMI" is not provided in the manuscript.
We have added the full form of the abbreviation "BMI" (Brain-machine interface) in the abstract.
2. The radiant energy of INS is not specified in Figure 5 and 6.
We have added the energy of INS in Figure legend (0.3 J/cm^2).
3. The term "higher order callosal response" can be confusing. The phrase "high-order visual processing facilitated by callosal connections" would be clearer.
We have revised as suggested by the reviewer. Thank you.

Reviewer #3 (Remarks to the Author):

The introduction to this paper makes some very broad claims about how this technology will address the shortcomings of bionic vision systems to meet requirements such as: (i) color, resolution, depth, contour orientations, motion directions; (ii) higher-order features; (iii) complete visual field and

integrating left-right information. Line 65 states: “Here we present an interface that addresses these three challenges” . However, the presented results are extremely preliminary and at best suggest that such things may be possible, but the system is far from practical: a few spots under a linear array of optical fibers (connected to large equipment) have been stimulated, and this stimulation shows a response in the opposite hemisphere.

R3-1. My first concern is with the description of the optical system and energies. It appears that a 4-W laser is being used (though I could not find this exact laser in the company catalog), which is passed through a low-loss piezo-mechanical optical switch to select a delivery fiber. The fibers are 200 um diameter (though the core size is not given). The reported energies are up to 0.7 J/cm² (presumable for a complete stimulation event?).

Response:

The laser is a custom CW laser purchased from CNI (1870nm, max 200mW at 200Hz, rise time 3.23 usec, 200um fiber), tailored to deliver 250usec pulses. We provide detailed description of the optical system and energies in the new Methods: We used 0.1-0.7J/cm², which is equal to 6.28mW-43.96mW (calculation function below). The energies we provided are *peak fluence values*, not the averaged power of the 0.5s pulse train. The energy delivery is adjustable from 0-200mW (0-3.18J/cm²).

Calculation formula: intensity*size / (1/frequency)

$$0.1\text{J}/\text{cm}^2 = 0.1\text{J}/\text{cm}^2 \cdot \text{s} \rightarrow (0.1 * \pi * 1\text{e}-4) / (0.5 * 1\text{e}-3) = 6.28\text{mW}$$

The optical switch is a 1x9 switch purchased from PiezoJena with <2msec switch time (electrical + optical) for all channels. Fiber NA 0.22, core size 200um, with cladding 220um.

R3-2. So, the energy out of the core of the fiber could be calculated as 0.7 J/cm² * Area, where the area is $\pi * 100 \text{ (um)}^2 = 0.7 * 1\text{e}-8 * \pi * 1\text{e}4 = 0.22 \text{ mJ}$. If we say this heats a 200-um deep 200-um diameter cylinder, then the temperature rise would be about 2 degC. Is this about right? Obviously any rise above this would be forbidden (usually rises <2degC are acceptable for RF heating).

Response:

Thanks for raising this question. See **R2-2 and Response Figure #2-2**. Our temperature rises are well below 1°C.

R3-3. However, taking a different route for calculations, we have a 4-W average power laser delivered by a 200-um fiber (perhaps a smaller core?), so a 16 ms stimulation would deliver 64 mJ, resulting in a 300x -higher temperature rise, which would be destructive.

Response:

Thanks for raising this question. 4-W is the maximum power of the laser. The maximum output power of the device at 200Hz frequency is 200mW. In addition, as shown in **R3-1**, no higher than 43.96mW (peak power of the pulse, equivalent to 0.7J/cm²) with very short pulses, while as **R2-2** showed, the temperature rise caused by INS stimulation will not cause tissue damage.

R3-4. My point is that there needs to be a calibrated measurement of the optical energy delivered, using say an integrating sphere and calibrated detector. Also, the waveform of the pulses needs to be captured, because it's not clear whether the switch is acting fast enough (there were no speed specifications on the manufacturer's site). Only then can some claim of “non-destructive” be made. Indeed, the temperature rise should be measured to confirm the volume being heated, which would help justify the specificity claim.

Response:

Thanks for raising this question. As written in R3-1, we used optical switch (the switching speed on the manufacturer's site is 2ms, <https://www.piezosystem.com/products/fiber-switches/>) to control the laser switching from channel to channel, while the laser device operates continuously at the power we set throughout the experiment session.

Response Figure #3-1. Intensities delivered by each of the 5 channels of fiber array measured by the power meter. (A) Average peak value per channel (\pm SD, the SD cross channels is 0.8). Radiant intensity was set to 15mW in this test. (B) Five-channel sequential (from ch1-5) intensity measurement (0.5s pulse train per channel, the cycle from ch1-5 repeating 20 times) through optical switch, which is the entire measurement result of (A). The graph in the top right corner: schematic diagram of 0.5s pulse trains (each contains 100 250us-pulses), the interval between adjacent pulse train is 0.5s. The numbers below indicate different channels.

At the beginning of each experiment session, we used calibrated detector and optical power meter from Thorlabs (PM320E, dual-channel benchtop power and energy meter console) to measure the output energy from each fiber tip. Below is the energy of the pulse trains we captured by Optical power monitor connected with optical power meter.

R3-5. The results mainly rely on very small changes in reflectance of the cortical surface ($<0.1\%$) to indicate neural activity, convolution with a spatial filter (1.2 mm diameter), and a low-pass filter (surely not 3-8 pixels in diameter, but some frequency in Hz?), truncation of extraneous results, and scaling by root-N divided by the standard deviation of the difference between stimulated and unstimulated measurements (line 400), which probably needs a better justification. Some calibration bars are required on the greyscale images, to identify the actual change in reflectivity. The definition of activation size should be made (is it a measure of the area of $>50\%$ change compared with the peak change?).

Response:

Thanks for raising this question. These are standard optical imaging procedures. Please see our response in **R1-11** (Grinvald et al. 1986, 1988, Roe and Ts'o 1995, Tanigawa et al. 2010, Zhang et al. 2023). We have added the grayscale bars. We have explicitly defined the method of measuring the activation size (area and diameter). The figure below illustrates the equivalence of using \pm SD and $\pm\%dR/R$.

References:

- Grinvald, A., Lieke, E., Frostig, R. D., Gilbert, C. D., & Wiesel, T. N. (1986). Functional architecture of cortex revealed by optical imaging of intrinsic signals. *Nature*, 324(6095), 361-364.
- Grinvald, A., Frostig, R. D., Lieke, E. D. M. U. N. D., & Hildesheim, R. I. N. A. (1988). Optical imaging of neuronal activity. *Physiological reviews*, 68(4), 1285-1366.
- Roe, A. W., & Ts'o, D. Y. (1995). Visual topography in primate V2: multiple representation across functional stripes. *Journal of Neuroscience*, 15(5), 3689-3715.
- Tanigawa, H., Lu, H. D., & Roe, A. W. (2010). Functional organization for color and orientation in macaque V4. *Nature neuroscience*, 13(12), 1542-1548.
- Zhang, Y., Schriver, K. E., Hu, J. M., & Roe, A. W. (2023). Spatial frequency representation in V2 and V4 of macaque monkey. *Elife*, 12, e81794.

Response Figure #3-2. Comparison of result displayed with different calibration bar. (A) Original figure clipped with $\pm 1.5SD$ (also shown in R1-11). (B) Original figure clipped with $\pm 0.05\%$ dR/R. (C) T-test map when $p < 0.001$, which is the threshold we chose to indicate the activation region. The area is determined from the number of significant pixels and the diameter is the average of the long and short axes.

R3-6. Features of the waveforms need explaining. Fig. 2E has fluctuations greater than the error bars (which are very difficult to see, presumably due to jpg encoding). Fig. 4B shows that the peaks are all the same, but the traces diverge towards 5 s: I would expect them to re-converge – why not? The later waveforms cut off at 5 s, long before reconvergence, which is disturbing. Fig. C has 2Hz oscillations, which is very strange.

Response:

Thanks for raising this question. These signals are not unusual for optical imaging signal timecourses obtained in rats, cats, monkeys, and humans. (1) *5 sec acquisition*: The intrinsic signal peaks early between 1-3 sec (called the initial dip) and returns to baseline within about 10sec (Grinvald et al. 1988), often with a slight undershoot, over the next several seconds (see Lu et al. 2017 for movie of full spatiotemporal timecourse). As primary response is in the first few seconds, 4-5 seconds is the signal that many optical imagers capture (as we have done in most of our studies over the past 30 years). To ensure that the next trial starts from baseline, we use an ISI of at least 7 seconds (which provides well over 10sec between trial onsets). These are also parameters used by others in the field (Ts'o et al. 1990, Shmuel and Grinvald 1990, Wang et al. 2007). (2) *Fluctuations in the timecourse*: There are multiple noise sources in the acquisition of these signals, including that from the camera electronics, illumination shot noise, physiological noise, and motion. Electronics noise and shot noise are minimized by purchasing good equipment. Physiological noise is minimized by synchronizing image acquisition with respiration and heartbeat. A glass coverslip helps to dampen noise due to brain and vessel pulsations. In this preparation, it was a challenge to completely dampen cortical motion with the glass coverslip because of the large field of view across two hemispheres and because the brain could only be covered by the glass up to the location of the array. Thus some of the irregular bumps in the timecourse in Fig. 2 may be due to remaining brain pulsation noise. In Figure 4, the variability of the late part of the signal is not unusual. The early darkening part of the signal is due to deoxygenation of the blood induced by neural activity; the late lightening part is due to influx of newly oxygenated blood. This late influx is often not as synchronized to the neural event and contains more variability (Chen-Bee et al. 2007, Suh et al. 2006). For this data set, we collected 10 trials per condition. Increasing trial number can reduce the noise; however, total acquisition time can be limited by the duration of a stable fiber array/agar/coverslip preparation.

Reference:

- Chen-Bee, C. H., Agoncillo, T., Xiong, Y., & Frostig, R. D. (2007). The triphasic intrinsic signal: implications for functional imaging. *Journal of Neuroscience*, 27(17), 4572-4586.
- Grinvald, A., Frostig, R. D., Lieke, E. D. M. U. N. D., & Hildesheim, R. I. N. A. (1988). Optical imaging of neuronal activity. *Physiological reviews*, 68(4), 1285-1366.
- Lu, H. D., Chen, G., Cai, J., & Roe, A. W. (2017). Intrinsic signal optical imaging of visual brain activity: tracking of fast cortical dynamics. *NeuroImage*, 148, 160-168.
- Shmuel, A., & Grinvald, A. (1996). Functional organization for direction of motion and its relationship to orientation maps in cat area 18. *Journal of Neuroscience*, 16(21), 6945-6964.
- Suh, M., Bahar, S., Mehta, A. D., & Schwartz, T. H. (2006). Blood volume and hemoglobin oxygenation response following electrical stimulation of human cortex. *Neuroimage*, 31(1), 66-75.
- Ts'o, D. Y., Frostig, R. D., Lieke, E. E., & Grinvald, A. (1990). Functional organization of primate visual cortex revealed by high resolution optical imaging. *Science*, 249(4967), 417-420.

Wang, Y. I., Xiao, Y., & Felleman, D. J. (2007). V2 thin stripes contain spatially organized representations of achromatic luminance change. *Cerebral Cortex*, 17(1), 116-129.

R3-7. A potential uncertainty in the experiment might be the mechanical pressure from the array. A single cross (+) shaped array, placed once and not moved would be more convincing.

Response:

Thanks for raising this question. We have shown in previous studies that the fiber on the cortex does not damage the cortex (Chernov 2014 Brain Stim, Pan et al 2023 Brian Stim) and is not the source of neural activation (Chernov, 2021, Neurophotonics). The fiber is very stable as the cortex is stabilized by a glass to reduce the pulsation or vascular noise (Chen, 2002, J Neurosci Methods; Arieli and Grinvald, 2002, J Neurosci Methods). The cross-shaped array is a good idea; we intend to deliver different patterns of fiber stimulation using a 100-fiber array (Schrivier et al 2022, FENS).

References:

Arieli, A., & Grinvald, A. (2002). Optical imaging combined with targeted electrical recordings, microstimulation, or tracer injections. *Journal of neuroscience methods*, 116(1), 15-28.

Chen, L. M., Heider, B., Williams, G. V., Healy, F. L., Ramsden, B. M., & Roe, A. W. (2002). A chamber and artificial dura method for long-term optical imaging in the monkey. *Journal of neuroscience methods*, 113(1), 41-49.

Chernov, M. M., Chen, G., & Roe, A. W. (2014). Histological assessment of thermal damage in the brain following infrared neural stimulation. *Brain Stimulation*, 7(3), 476-482.

Chernov, M. M., Friedman, R. M., & Roe, A. W. (2021). Fiberoptic array for multiple channel infrared neural stimulation of the brain. *Neurophotonics*, 8(2), 025005-025005.

Pan, L., Ping, A., Schriver, K. E., Roe, A. W., Zhu, J., & Xu, K. (2023). Infrared neural stimulation in human cerebral cortex. *Brain Stimulation*, 16(2), 418-430.

Schrivier KE, Zhang Y, Liu Y, Roe AW (2022) Illuminating the Mesoscale Connectome: A 100-fiber Infrared Neural Stimulation System. Fed Eur Neurosci Soc. Paris, France. July 9-13, 2022.

R3-8. The mechanism behind an array positioned V or H across many columns (where some are responsive to V optical stimulation, and others H) causing an increase or suppression of response in the contralateral cortex seem counter-intuitive. It might be more intuitive if a single V or H column was selected and a contra-lateral response observed. I know the present experiment is designed to show higher-order integration, but it seems a very complex effect.

Response:

The orientation-matched enhancement and orthogonal orientation suppression is predicted from previous studies. Please see our response in **R1-3**. Note the fibers in our array placements contact orientation domains of several different orientations, not just H and V. So the integrated result on the contralateral side is truly higher order, as the resulting orientation is independent or integrated across these different inputs.

R3-9. I wonder whether the out-of-band energy from the 4-W heating laser directly affects the CCD camera, and whether any steps were taken to eliminate this hypothesis, such as using a piece of paper rather than a brain to check for cross-talk?

Response:

Thanks for your concern. The reviewer is concerned that the out-of-band energy at 1875nm will influence the CCD camera to generate cross-talk. 1. We used no higher than 43.96mW; 2. The spectral range of our camera model (photon focus MV1-D1312-160-CL) is from 320-1000nm and is not sensitive to 1875nm wavelength.

Response Figure #3-3. Quantum efficiency image sensor of our CCD camera.

R3-10. Overall, there are some very interesting observations in this work, but I would like more clarification on the experimental technique, and how it was verified and calibrated, as the signals are very tiny, and some of the responses not what I would expect (e.g. not returning to the resting state after stimulation, which would indicate some damage). In some cases, it would be more convincing to plot (every measurement, rather than presenting averages).

Response:

Thanks for raising this question. As shown in **R3-6**, the intrinsic signal peaks early between 1-3 sec (called the initial dip) and returns to baseline within about 10sec, often with a slight undershoot, over the next several seconds. As primary response is in the first few seconds, 4-5 seconds is the signal that many optical imagers capture. To ensure that the next trial starts from baseline, we use an ISI of at least 7 seconds (which provides well over 10sec between trial onsets), but we did not record the whole ISI period. The signal not returning to the resting state 4-5 seconds after stimulation is typical and does not indicate damage on the cortex (e.g. see visual response in Manuscript Figure 5 and 6). Below, we provide one example to show every measurement of each trial (Response Figure #3-4). Response Figure #3-5 below shows an example of published OI timecourses.

Response Figure #3-4. Response example of each trial (case1: horizontal domain). (A) The time-courses of each trial (green lines: Vision, x trials; gray lines: Vision + INS). (B) Averaged reflectance changes during 2.5–5s of the time courses in A. (Welch's t-test, ****: $p < 0.0001$, Vision vs. Vision+INS).

Response Figure #3-5. From Tanigawa et al. 2010 Nat Neurosci.

REVIEWER COMMENTS

Reviewer #1 (Remarks to the Author):

Overall, the authors did make notable improvements to the manuscript, including additional contextual information, improved data presentation, and figure clarity. Despite these improvements and the interesting techniques, significant issues remain with the manuscript, particularly in appropriately contextualizing INS compared to ICMS.

The authors indicate in their response and introduction that “targeted stimulation of cortical columns has not been an explicit strategy of electrical VCPs” and that is not the case. The columnar scale and functional unit framework is not a new conceptual approach. Researchers employing ICMS and micro scale surface electrodes have long since framed the scale of their approach with respect to the functional units of visual cortex. Modeling studies as well as animal studies have been conducted that use and describe electrical micro stimulation at the cortical column scale. Just to cite a few:

Christie BP, Ashmont KR, House PA, Greger B. Approaches to a cortical vision prosthesis: implications of electrode size and placement. *J Neural Eng.* 2016 Apr;13(2):025003. doi: 10.1088/1741-2560/13/2/025003. Epub 2016 Feb 23. PMID: 26905379.

Allison-Walker T, Hagan MA, Price NSC, Wong YT. Microstimulation-evoked neural responses in visual cortex are depth dependent. *Brain Stimul.* 2021 Jul-Aug;14(4):741-750. doi: 10.1016/j.brs.2021.04.020. Epub 2021 May 8. PMID: 33975054.

Pavloski, R. Progress in Developing an Emulation of a Neuromorphic Device That Is Predicted to Enhance Existing Cortical Prosthetic Vision Technology by Engaging Desired Visual Geometries. *Prosthesis* 2022, 4, 600-623. <https://doi.org/10.3390/prosthesis4040049>

Bradley DC, Troyk PR, Berg JA, Bak M, Cogan S, Erickson R, Kufta C, Mascaro M, McCreery D, Schmidt EM, Towle VL, Xu H. Visuotopic mapping through a multichannel stimulating implant in primate V1. *J Neurophysiol.* 2005 Mar;93(3):1659-70. doi: 10.1152/jn.01213.2003. Epub 2004 Sep 1. PMID: 15342724.

Roy A, Osik JJ, Ritter NJ, Wang S, Shaw JT, Fiser J, Van Hooser SD. Optogenetic spatial and temporal control of cortical circuits on a columnar scale. *J Neurophysiol.* 2016 Feb 1;115(2):1043-62. doi: 10.1152/jn.00960.2015. Epub 2015 Dec 2. PMID: 26631152; PMCID: PMC4839491.

The authors state “Percepts elicited have ranged from 2-10 or more degrees in size...4-8

However, the percepts evoked in NHP studies and human studies using Utah arrays (Chen, Torab, Fernandez, etc) have evoked individual phosphenes notably smaller than 1 degree. For example, taken from Fernandez 2021, “The estimated phosphene size evoked by single electrode stimulation was usually

very small (0.8 ± 0.8 ; mean \pm SD of subject's size estimates) and resembled "pin points" of light at arm's length".

The authors have improved the clarity and presentation of data showing single fiber INS activation. In the results section Single Fiber Stimulation, the authors report the data they are presenting is "similar to activations evoked by visual spot stimulations³⁹", which were conducted in macaques. The case for this similarity would be significantly strengthened by showing the activation profile for visual spot stimulation in the same animals undergoing the INS stimulation, and if the authors reported the size in degrees of the visual spot stimulation that evoked similar responses.

Sequential Multi-Fiber Stimulation

The authors report an enhanced response to vertical visual stimulation. The addition of error bars is an improvement, but it is unclear what the error bars represent (standard deviation, standard error, 95% CI, ...). The data presented in 5B-E do not appear to demonstrate differences between vision alone and the vision+INS conditions (B) or differences between the vision+INS conditions (B&C), when comparing individual time points. Are the differences significant at each time point or only when pooling and averaging across multiple time points from 2.5-5s. There seems to have been a large change in the response to visual stimulation along vertical or horizontal domains, but the response with INS seems quite similar regardless of domain.

Discussion

The authors make the claim "For two sites to be distinguishable, traditional BMIs have had to place stimulation points relatively far apart (e.g. 2-3mm); this results in low spatial acuity and crude percepts^{4,5}". This is inaccurate. Fernandez 2021 clearly demonstrated cases where two electrodes spaced 400 μ m apart were able to produce separate phosphenes. Bosking 2022 also showed cases with 1-2mm separation can evoke separate phosphenes.

The authors state "this is the first time that such an inter-areal, higher order orientation (contour orientation) selective effect has been shown with focal columnar stimulation. This suggests that columnar VCPs can effectively engage existing cortical circuits." This is not clear based on the results presented. There seems to have been some small interaction between natural visual processing and INS and potentially some disruption, but the data do not seem to support this strong of a claim.

The authors state "Here, using a single five fiber array, we presented 10 stimulation paradigms: stimulation of single fibers, sequential five-fiber stimulation in two directions and at two speeds, and random (non-directional) five-fiber stimulation. Similar capabilities have been developed in other

studies^{5,7}; however, not at dense columnar scale.” Their optical fibers are spaced 500 μ m, this is larger than electrode spacing on Utah arrays (400 μ m) used in Fernandez 2021. It is inaccurate to claim this report demonstrates improved density.

Response to reviewers:

The authors responded to a criticism that the first submission did not sufficiently explain how their technique would be used as a VCP or how the authors would account for an inability to collect feature maps in blind individuals by saying that it is their opinion that “even in the blind, columnar maps would still likely be accessible and potentially mappable via imagery or memory paradigms”. There is no sufficient evidence to suggest that visual field mapping with mental imagery will be possible in blind patients, especially at the cortical column scale. Their response included a pop science article from 2007, which is not a good example and in no way shows that visual field mapping through mental imagery in the blind would be possible, especially at the scale and specificity feasible with modern electrode arrays.

The authors reference split-brained individuals in their response, but have not made clear how deficits observed in split-brain individuals are relevant to creating visual sensations through direct brain stimulation. The authors have not sufficiently argued the benefit of demonstrating callosal connections nor indicated how this compares to or represents an improvement over electrical stimulation.

The authors comment on the need for callosal connections to perform visually guided behaviors.

It is relevant to point out, based on early VCP clinical trials, this does not appear to be relevant. All trials have thus far been single hemisphere implants. Each study has demonstrated patients performing simple, visually guided tasks.

The authors state in their response, “Confirming that higher order activation can be evoked by multifiber stimulation is a key step in establishing the utility of this VCP approach on brainwide circuits. This is the aim of the study here”. While eliciting higher order activity is certainly a goal of VCPs, the goal is to stimulate visual cortex (or higher level areas) in a way that propagates throughout the visual processing hierarchy without visual input, rather than modulating visual sensations evoked through photic stimulation of the retina.

Reviewer #2 (Remarks to the Author):

The authors have addressed all of my concerns. I thank them for their hard work and I now think this manuscript will make a great addition to the literature!

Reviewer #3 (Remarks to the Author):

Thank you for your detailed responses.

I guess you have chosen not to respond to my disapproval of the claims about advantages being overblown and not proven?

R3-1) Optical powers. The calculation seems wrong as $1/f = 5 \text{ ms}$ for 200 Hz. However it seems to evaluate correctly. That is, if the area is the area of the fiber ($\pi \times 0.01 \text{ cm} \times 0.01 \text{ cm}$), and the energy density of a pulse is 0.1 J, and the repetition rate is 200 pulses/s, then the average power at the tip of the fiber would be

$$0.1 \times \pi \times 0.01 \times 0.01 \times 200 = 2\pi \text{ mW.}$$

The methods section is confusing in its incorrect use of power (average, peak), energy and intensity.

E.G: The peak power of (a) laser pulse is 0.3 J/cm^2 or 0.7 J/cm^2 . I would call this "the peak energy density of the pulse", and at "at the tip of the fiber".

The average power will be the Energy Density (J/cm^2) x area (not size) of tip x pulse repetition frequency.

Now, Fig #1.3 in the Response, shows a volume under the tip of approx. the fiber diameter x 0.1 cm (depth), which is $1e-5 \times \pi \text{ cm}^3$.

A single pulse has an energy $0.1 \times \pi \times 0.01 \times 0.01 = 3.14e-5$ J, which means that the average temperature rise in this volume will be $1/4.186$ degC, as the brain is mainly water. For $0.7\text{J}/\text{cm}^2$, it will be just under 2 degC. Of course, near the fiber's tip, the rise will be much greater, which is concerning. In your Response Figure #2-2, the rise is < 2 degC, but this is averaged power a voxel. It would be very useful to know the volume of a voxel.

Thank you for Response Figure #3-1. However, the y-axis should be power, not intensity.

Response 3.9. I can see that you have provided Quantum Efficiency graphs up to 1100 nm, and you have assumed the QE falls at higher wavelengths. This is reasonable due to Einstein's theory of photodetection. I was just worried that there might be a thermal effect at longer wavelengths, which you could disprove with a simple experiment.

In the MS Fig. 7 D, I see a periodic "up-down-up-down" fluctuation in the results. Something deterministic is happening here. Have you any idea what could be causing this systematic error?

Thank you again for submitting your manuscript "A novel interface for cortical columnar neuromodulation with multi-point infrared neural stimulation" to Nature Communications. We have now received reports from 3 reviewers. As you can read in their comments below, Ref 1 and 3 are not completely satisfied with how you addressed their concerns. Ref. 1 calls you out on a number of claims and points to relevant literature that you should cite and discuss. They also point out that it is unclear the advantage over electrical stimulation devices. I and the editorial team decided to offer an additional revision. However, in the new version, you should lay down clearly the advantages over intracortical electrical stimulation as suggested by Rev. 1 and 3.

Please add in the Methods or SI the calculations to obtain the temperature rise, after taking into consideration the comments of Rev. 3.

Calculation of heat rise in thermometry. We have added this to the Supplementary Methods:

“In the thermometry study, we converted the direct proton resonance frequency phase shift data in acquired images to temperature change according to (Rieke and Pauly 2008, Luo et al 2022):

$$\Delta T = \frac{\phi(T) - \phi(T_0)}{\gamma\alpha B_0 TE} \quad (1)$$

where $\phi(T)$ is the phase map at current time point, $\phi(T_0)$ is the phase map of the baseline image which is measured at room temperature (before INS), γ here represents the gyromagnetic ratio of hydrogen (2.67×10^8 rads/Tesla, constant), α is the PRF shift coefficient of water (-0.01 ppm/ $^{\circ}$ C, constant), B_0 is the magnetic field strength (7 Tesla here), and TE is the echo time of imaging sequence (1.65 msec here). And in our condition, 1° C \cong 0.031 rads.

The temperature of the voxel with greatest phase shift at the location of the optic fiber tip was selected, and the temperature increase in the 9 isolated measurements was averaged for further analysis. ANOVA and paired-sample t-test were conducted to statistically analyze the temperature changes before and after INS, as well as the differences across different power intensities at the same time point. The data was analyzed using MATLAB.”

When resubmitting, you must provide a point-by-point response to the reviewers' comments. Please show all changes in the manuscript text file with track changes or colour highlighting. If you are unable to address specific reviewer requests or find any points invalid, please explain why in the point-by-point response.

REVIEWER COMMENTS

Reviewer #1 (Remarks to the Author):

Overall, the authors did make notable improvements to the manuscript, including additional contextual information, improved data presentation, and figure clarity. Despite these improvements and the interesting techniques, significant issues remain with the manuscript, particularly in appropriately contextualizing INS compared to ICMS.

R1-1: The authors indicate in their response and introduction that “targeted stimulation of cortical columns has not been an explicit strategy of electrical VCPs” and that is not the case. The columnar scale and functional unit framework is not a new conceptual approach. Researchers employing ICMS and micro scale surface electrodes have long since framed the scale of their approach with

respect to the functional units of visual cortex. Modeling studies as well as animal studies have been conducted that use and describe electrical micro stimulation at the cortical column scale. Just to cite a few:

Christie BP, Ashmont KR, House PA, Greger B. Approaches to a cortical vision prosthesis: implications of electrode size and placement. *J Neural Eng.* 2016 Apr;13(2):025003. doi: 10.1088/1741-2560/13/2/025003. Epub 2016 Feb 23. PMID: 26905379.

Allison-Walker T, Hagan MA, Price NSC, Wong YT. Microstimulation-evoked neural responses in visual cortex are depth dependent. *Brain Stimul.* 2021 Jul-Aug;14(4):741-750. doi: 10.1016/j.brs.2021.04.020. Epub 2021 May 8. PMID: 33975054.

Pavloski, R. Progress in Developing an Emulation of a Neuromorphic Device That Is Predicted to Enhance Existing Cortical Prosthetic Vision Technology by Engaging Desired Visual Geometries. *Prosthesis* 2022, 4, 600-623. <https://doi.org/10.3390/prosthesis4040049>

Bradley DC, Troyk PR, Berg JA, Bak M, Cogan S, Erickson R, Kufta C, Mascaro M, McCreery D, Schmidt EM, Towle VL, Xu H. Visuotopic mapping through a multichannel stimulating implant in primate V1. *J Neurophysiol.* 2005 Mar;93(3):1659-70. doi: 10.1152/jn.01213.2003. Epub 2004 Sep 1. PMID: 15342724.

Roy A, Osik JJ, Ritter NJ, Wang S, Shaw JT, Fiser J, Van Hooser SD. Optogenetic spatial and temporal control of cortical circuits on a columnar scale. *J Neurophysiol.* 2016 Feb 1;115(2):1043-62. doi: 10.1152/jn.00960.2015. Epub 2015 Dec 2. PMID: 26631152; PMCID: PMC4839491.

Response:

We thank the reviewer for pointing this out and for these references. We apologize for these statements. What we meant to say is “placement of stimulation sites was not guided by the known organization of functional columns in visual cortex.” We have now substantially revised the introduction by summarizing more of the recent advances, removing the statement that “many percepts elicited have ranged from 2-10 or more degrees in size”, adding the studies pointed out by the reviewer relevant to columnar stimulation, removing “this is a new technological and conceptual approach”, and rewording the purpose of this study, “to demonstrate that higher order representations can be accessed by designed columnar stimulation of lower order representations.”

[Removed these statements in Introduction, and revised the third to last paragraph of Discussion]: “Given the percepts induced by single site stimulation, what can be elicited by multisite stimulation? A few studies have examined the percepts evoked by multisite electrical stimulation, either sequentially^{1,40,41,69} or simultaneously^{69,70}. In human studies, surface electrodes (0.5-3mm diameter, impedance <18k Ω , stimulation frequency 60/120/200 Hz, and current amplitudes 0.3-4/1.2-1.5 mA, several millimeters spacing, spanning a few centimeters in V1 and V2, over 10° of visual space) have evoked percepts of shapes and letters. These percepts can be enhanced by a ‘dynamic current steering’ stimulation paradigm⁴¹, enhancing the salience of the percept and reducing the number of stimulation sites needed^{40,71}. In a study by Fernández et al.⁷⁰, a blind subject implanted with a Utah array was able to discriminate evoked sizes and shapes as well as spatially small percepts, marking a significant advance in spatial resolution (see also Yoshor and Bosking⁷²). Roelfsema’s group implanted an astounding 1024-channel-count electrode interface (16 Utah arrays, 0.3-0.6M Ω , <100uA) into monkey V1 and V4, achieving percepts of shapes, letters, and motion direction⁶⁹. Although presumably higher order circuits associated with these sites of stimulation were activated, the associated brain circuits were not studied.”

R1-2: The authors state “Percepts elicited have ranged from 2-10 or more degrees in size...4-8 However, the percepts evoked in NHP studies and human studies using Utah arrays (Chen, Torab, Fernandez, etc) have evoked individual phosphenes notably smaller than 1 degree. For example, taken from Fernandez 2021, “The estimated phosphene size evoked by single electrode stimulation

was usually very small (0.8 ± 0.8 ; mean \pm SD of subject' s size estimates) and resembled “pin points” of light at arm' s length” .

Response:

Thanks for pointing this out. We were referring to the size of the overall percept evoked by a phosphene pattern, not a single phosphene. We have now removed our description, “percepts elicited have ranged from 2-10 or more degrees in size”. The size of single phosphene elicited by electrical stimulation can be very small (e.g. phosphene evoked in monkey V1, from Schiller 2011 PNAS, is between 9 and 26 min of arc). We have now also cited studies (Bak et al. 1990, Fernandez et al. 2021), describing the percepts as “spatially small percepts” (the third to last paragraph of Discussion).

Reference:

Bak, M., Girvin, J. P., Hambrecht, F. T., Kufta, C. V., Loeb, G. E., & Schmidt, E. M. (1990). Visual sensations produced by intracortical microstimulation of the human occipital cortex. *Medical and Biological Engineering and Computing*, 28, 257-259.

Fernández, E., Alfaro, A., Soto-Sánchez, C., Gonzalez-Lopez, P., Lozano, A. M., Peña, S., ... & Normann, R. A. (2021). Visual percepts evoked with an intracortical 96-channel microelectrode array inserted in human occipital cortex. *The Journal of clinical investigation*, 131(23).

Schiller, P. H., Slocum, W. M., Kwak, M. C., Kendall, G. L., & Tehovnik, E. J. (2011). New methods devised specify the size and color of the spots monkeys see when striate cortex (area V1) is electrically stimulated. *Proceedings of the National Academy of Sciences*, 108(43), 17809-17814.

R1-3: The authors have improved the clarity and presentation of data showing single fiber INS activation. In the results section Single Fiber Stimulation, the authors report the data they are presenting is “similar to activations evoked by visual spot stimulations³⁹”, which were conducted in macaques. The case for this similarity would be significantly strengthened by showing the activation profile for visual spot stimulation in the same animals undergoing the INS stimulation, and if the authors reported the size in degrees of the visual spot stimulation that evoked similar responses.

Response:

Thanks for your suggestion. In macaque V1: (1) spot imaging results in single spot activations that span a few ocular dominance columns (e.g. for 0.25° spot 3 OD columns, Lu et al. 2009), and single 200um fiber INS stimulation also results in activations that span a few OD columns (400um wide, Cayce et al. 2014). To examine whether this is also observed in cat visual cortex, we have conducted a spot imaging experiment in cat visual cortex. As shown in **Fig. R3A**, in cat area 18, we obtained functional maps to a 2° spot and a 1° spot (**Fig. R3B**, 2° spot is slightly lateral to and lower than the 1° spot), imaged monocularly (each spot comprised drifting gratings of 2 orthogonal orientations, SF=0.2 cycles/deg, TF=5 Hz). [We also obtained orientation maps (**Fig. R3E, 3I**, full screen, binocular stimuli).] As there are ocular dominance columns, roughly 1mm in size, in area 18 (Cynader et al. 1987; Rochefort et al. 2007), the activation maps comprised 2-3 monocular domains roughly 1mm in size (2° spot: **Fig. R3F & J**; 1° spot: **Fig. R3G & K**). We defined the total size of the activation region using the width and length of a circle or an ellipse around these domains. This was compared to the size of activation due to INS ($0.3\text{J}/\text{cm}^2$) (**Fig. R3H & L**). The size of the INS activation is well within the activation sizes of the 1° and 2° spot stimuli. Although we did not test smaller spot stimulations, the results indicate that the INS can evoke activation even smaller

than a 1 degree spot.

Fig. R3: Comparison of response size evoked by spot visual stimuli and single fiber INS stimulation. (A) Schematic picture of a cat cerebral cortex. Orange rectangle: approximate imaging field of view. Dotted lines: approximate areal borders. (B) Schematic of spot positions. 2° spot slightly more lateral than 1° spot. (C, D) Blood vessel maps recorded in the visual stimulation and INS conditions. (E, I) orientation map of 'H'-V' full screen grating visual stimulation (binocular stimuli). (F, G) response map of 'vision'-'blank' from 2° (F), 1° (G) visual spot stimulation (monocular stimuli). (H) activation map in response to single fiber INS stimulation ('INS'-'blank', wavelength: 1870nm, frequency: 200Hz, radiant exposure: 0.3J/cm², pulse width: 250us, pulse train: 0.5s). (J-L) significantly ($p < 0.05$) activated pixels of F-H. Numbers at top of each panel: activation size of the cortex in the corresponding condition. Scale bar: 1mm. A: anterior, M: median.

Reference:

- Cayce, J. M., Friedman, R. M., Chen, G., Jansen, E. D., Mahadevan-Jansen, A., & Roe, A. W. (2014). Infrared neural stimulation of primary visual cortex in non-human primates. *Neuroimage*, 84, 181-190.
- Cynader, M. S., Swindale, N. V., & Matsubara, J. A. (1987). Functional topography in cat area 18. *Journal of Neuroscience*, 7(5), 1401-1413.
- Lu, H. D., Chen, G., Ts'o, D. Y., & Roe, A. W. (2009). A rapid topographic mapping and eye alignment method using optical imaging in Macaque visual cortex. *Neuroimage*, 44(3), 636-646.
- Rochefort, N. L., Buzás, P., Kisvárdy, Z. F., Eysel, U. T., & Milleret, C. (2007). Layout of transcallosal activity in cat visual cortex revealed by optical imaging. *Neuroimage*, 36(3), 804-821.

Sequential Multi-Fiber Stimulation

R1-4: The authors report an enhanced response to vertical visual stimulation. The addition of error bars in an improvement, but it is unclear what the error bars represent (standard deviation, standard error, 95% CI, ...). The data presented in 5B-E do not appear to demonstrate differences between vision alone and the vision+INS conditions (B) or differences between the vision+INS conditions (B&C), when comparing individual time points. Are the differences significant at each time point or only when pooling and averaging across multiple time points from 2.5-5s. There seems to have been a large change in the response to visual stimulation along vertical or horizontal domains, but the response with INS seems quite similar regardless of domain.

Response:

The error bars represent standard error mean, SEM.

Each point is an average of multiple (10-15) trials. Some individual timepoints are significantly different and some are not. However, using Welch's t-test (a paired t-test, normal distributions but unequal variances), the timecourses during the peak response (from 2.5-5s) were significantly

different between vision alone (green) and vision+INS conditions (brown: vision+slow INS, black: vision+fast INS, magenta (control): vision+random INS; in addition, vision+slow and vision+fast were significantly greater than vision+random).

Yes, the change in response to visual stimulation along vertical and horizontal domains were different. This could be due to anisotropies in representations of vertical and horizontal cardinal axes orientation arising from the cat's horizontal visual streak in the retina (Stone 1978, Stone et al. 1980, Schall et al. 1986), although we do not know of any optical imaging study that has explicitly compared retinal vs cortical vertical/horizontal anisotropy. Despite this difference in absolute response, the response change of orientation domains to vision/INS 'matched' and 'non-matched' domains is clearly opposite in effect.

References:

- Schall, J. D., Vitek, D. J., & Leventhal, A. G. (1986). Retinal constraints on orientation specificity in cat visual cortex. *Journal of Neuroscience*, 6(3), 823-836.
- Stone, J. (1978). The number and distribution of ganglion cells in the cat's retina. *Journal of Comparative Neurology*, 180(4), 753-771.
- Stone, J., Leventhal, A., Watson, C. R., Keens, J., & Clarke, R. (1980). Gradients between nasal and temporal areas of the cat retina in the properties of retinal ganglion cells. *Journal of Comparative Neurology*, 192(2), 219-233.

Discussion

R1-5: The authors make the claim “For two sites to be distinguishable, traditional BMIs have had to place stimulation points relatively far apart (e.g. 2-3mm); this results in low spatial acuity and crude percepts^{4,5}”. This is inaccurate. Fernandez 2021 clearly demonstrated cases where two electrodes spaced 400 μm apart were able to produce separate phosphenes. Bosking 2022 also showed cases with 1-2mm separation can evoke separate phosphenes.

Response:

Thanks for pointing this out. We have now removed this statement and have described Fernandez and Bosking studies in the Discussion.

In Fernandez 2021, they occasionally can see two discrete phosphenes evoked by electrodes spacing 400μm, but more often, it would be a single large phosphene. And they also said “when stimulation of the 2 electrodes was temporally separated by more than 250ms, the subject perceived 2 different, distinct phosphenes 90% of the time” for electrodes separated by 400μm. Thus, often a separation of 400μm is not sufficient to distinguish 2 phosphenes, but this separation is aided by adding temporal separation between the two sites (Beauchamp et al. 2020). In Bosking 2022, surface electrodes with 2-6mm spacing were used, and the manuscript showed that “simultaneous stimulation of pairs of electrodes separated by greater than 4mm tended to produce perception of two distinct phosphenes”.

References:

- Beauchamp, M. S., Oswalt, D., Sun, P., Foster, B. L., Magnotti, J. F., Niketeghad, S., ... & Yohor, D. (2020). Dynamic stimulation of visual cortex produces form vision in sighted and blind humans. *Cell*, 181(4), 774-783.
- Bosking, W. H., Oswalt, D. N., Foster, B. L., Sun, P., Beauchamp, M. S., & Yohor, D. (2022). Percepts evoked by multi-electrode stimulation of human visual cortex. *Brain stimulation*, 15(5), 1163-1177.
- Fernández, E., Alfaro, A., Soto-Sánchez, C., Gonzalez-Lopez, P., Lozano, A. M., Peña, S., ... & Normann, R. A. (2021). Visual percepts evoked with an intracortical 96-channel microelectrode array inserted in human occipital cortex. *The Journal of clinical investigation*, 131(23).

R1-6: The authors state “this is the first time that such an inter-areal, higher order orientation (contour orientation) selective effect has been shown with focal columnar stimulation. This suggests that columnar VCPs can effectively engage existing cortical circuits.” This is not clear based on the results presented. There seems to have been some small interaction between natural visual processing and INS and potentially some disruption, but the data do not seem to support this strong of a claim.

Response:

In a previous study by Makarov, they showed that inactivation by cooling of visual cortex affects contralateral response to oriented stimuli, but not spot stimuli, suggesting callosal connections mediate a higher order integration generating oriented effects. They also showed that this effect is orientation selective (enhancement: matched, relative suppression: non-matched), and consistent with the excitatory nature of callosal connections (Schmidt et al. 1997, Kiper et al. 1999, Rochefort et al. 2007). They conclude that the schematic below is the likely circuit (**Fig. R4**). Our demonstration that a VCP can induce the same integrated orientation-selective (enhancement: matched, relative suppression: non-matched) response suggests that it activates similar circuits. So, our data suggest that INS can access the circuits underlying callosal integration that results in higher order orientation response.

Fig. R4. A model of the present results. Because small squares do not affect contralateral targets, whereas gratings do, it appears that the output from neurons responsive to the small squares needs summation, to affect the contralateral targets. The cartoon suggests that the summation likely occurs at the target neurons in the receiving hemisphere. (from Makarov 2008)

We have revised this statement:

“For callosal circuits, as Makarov 2008 showed²², there are circuits in the carnivore brain that underlie this type of orientation-selective cross-callosal integration. They concluded that the schematic (Fig. 1B, upper) is the likely circuit. Our demonstration that a VCP can induce the same integrated orientation-selective (enhancement: matched, relative suppression: non-matched) response suggests that it activates the same circuits. Thus, our data suggest that INS applied to cortical columns can predictably access the circuits underlying higher order orientation response.”

References:

- Kiper, D. C., Knyazeva, M. G., Tettoni, L., & Innocenti, G. M. (1999). Visual stimulus-dependent changes in interhemispheric EEG coherence in ferrets. *Journal of Neurophysiology*, 82(6), 3082-3094.
- Rochefort, N. L., Buzás, P., Kisvárdy, Z. F., Eysel, U. T., & Milleret, C. (2007). Layout of transcallosal activity in cat visual cortex revealed by optical imaging. *Neuroimage*, 36(3), 804-821.
- Schmidt, K. E., Kim, D. S., Singer, W., Bonhoeffer, T., & Löwel, S. (1997). Functional specificity of long-range intrinsic and interhemispheric connections in the visual cortex of strabismic cats. *Journal of Neuroscience*, 17(14), 5480-5492.

R1-7: The authors state “Here, using a single five fiber array, we presented 10 stimulation paradigms: stimulation of single fibers, sequential five-fiber stimulation in two directions and at two speeds, and random (non-directional) five-fiber stimulation. Similar capabilities have been developed in other studies^{5,7}; however, not at dense columnar scale.” Their optical fibers are spaced 500um, this is larger than electrode spacing on Utah arrays (400um) used in Fernandez 2021.

It is inaccurate to claim this report demonstrates improved density.

Response:

The reviewer is correct. We have removed the statement “however, not at dense columnar scale”.

Response to reviewers:

R1-8: The authors responded to a criticism that the first submission did not sufficiently explain how their technique would be used as a VCP or how the authors would account for an inability to collect feature maps in blind individuals by saying that it is their opinion that “even in the blind, columnar maps would still likely be accessible and potentially mappable via imagery or memory paradigms” . There is no sufficient evidence to suggest that visual field mapping with mental imagery will be possible in blind patients, especially at the cortical column scale. Their response included a pop science article from 2007, which is not a good example and in no way shows that visual field mapping through mental imagery in the blind would be possible, especially at the scale and specificity feasible with modern electrode arrays.

Response:

We list the available methods that may be able to achieve visual field mapping. These are methods that may be feasible with development, but more work is required to make it systematic and readily implemented.

1. Similar to current electrical approaches, INS arrays can be used to sequentially stimulate different positions of the visual cortex. Based on the visual perception reported from the subjects using densely packed 200um fiber bundles, we can map the visual field at cortical columnar scale.

2. By imaging the spontaneous ongoing activity in the visual cortex, we can map and estimate the functional architectures according to resting state activation patterns in the visual cortex (Omer et al. 2019, Tsodyks et al. 1999). Nasr et al. used resting-state imaging to obtain the ocular dominance column mapping in primate V1 (Nasr 2023 OHBM abstract), and resting-state OI and fMRI has revealed mesoscale functional connectivity within the hand representation of primary somatosensory and motor cortex (areas 3b and 1) in anesthetized squirrel monkeys (Wang et al. 2013, Shi et al. 2017, Card and Gharbawie 2022).

3. Based on Myelination differences (T1/T2 MRI mapping), we can distinguish different functional areas (e.g. V1/V2 border, MT) and stripes in V2 (Helbling and Weiskopf 2023).

4. With the accumulation of data on the functional structures of the normal human brain, we can use machine learning methods to predict the distribution of functional columns in the patient's cortex.

5. Use of visual imagery is a future direction that will require further study.

References:

Card, N. S., & Gharbawie, O. A. (2022). Cortical connectivity is embedded in resting state at columnar resolution. *Progress in neurobiology*, 213, 102263.

Haenelt, D., Trampel, R., Nasr, S., Polimeni, J. R., Tootell, R. B., Sereno, M. I., ... & Weiskopf, N. (2023). High-resolution quantitative and functional MRI indicate lower myelination of thin and thick stripes in human secondary visual cortex. *elife*, 12, e78756.

Nasr, S. (2023). Resting-State Functional Connectivity between Ocular Dominance Columns. OHBM. Abstract.

Omer, D. B., Fekete, T., Ulchin, Y., Hildesheim, R., & Grinvald, A. (2019). Dynamic patterns of spontaneous ongoing activity in the visual cortex of anesthetized and awake monkeys are different. *Cerebral Cortex*, 29(3), 1291-

1304.

Shi, Z., Wu, R., Yang, P. F., Wang, F., Wu, T. L., Mishra, A., ... & Gore, J. C. (2017). High spatial correspondence at a columnar level between activation and resting state fMRI signals and local field potentials. *Proceedings of the National Academy of Sciences*, 114(20), 5253-5258.

Tsodyks, M., Kenet, T., Grinvald, A., & Arieli, A. (1999). Linking spontaneous activity of single cortical neurons and the underlying functional architecture. *Science*, 286(5446), 1943-1946.

Wang, Z., Chen, L. M., Négyessy, L., Friedman, R. M., Mishra, A., Gore, J. C., & Roe, A. W. (2013). The relationship of anatomical and functional connectivity to resting-state connectivity in primate somatosensory cortex. *Neuron*, 78(6), 1116-1126.

R1-9: The authors reference split-brained individuals in their response, but have not made clear how deficits observed in split-brain individuals are relevant to creating visual sensations through direct brain stimulation. The authors have not sufficiently argued the benefit of demonstrating callosal connections nor indicated how this compares to or represents an improvement over electrical stimulation.

The authors comment on the need for callosal connections to perform visually guided behaviors. It is relevant to point out, based on early VCP clinical trials, this does not appear to be relevant. All trials have thus far been single hemisphere implants. Each study has demonstrated patients performing simple, visually guided tasks.

Response:

Callosal connections are important for integrating the left and right visual fields (e.g. for reading, Hunter et al. 2007), tying together the vertical meridian (Blakemore et al. 1983; Payne 1990; Payne and Siwek 1991), and the fovea (which is represented on both hemispheres, Kennedy et al. 1986). The coordination of visually guided bimanual behavior also requires the integration of the two hemispheres. Split-brain individuals provide examples of deficits resulting from lack of callosal connections, revealing severe sensory, motor, and cognitive deficits of left and right hemifield coordination. These aspects of vision and visually guided behaviors are also challenges that must be addressed by VCPs.

In both cats and monkeys, callosal connections link similar topographic locations at or near the vertical meridian. The callosal fibers in essence stitch together the two halves of the visual field (Innocenti et al. 1995). In addition to linking topographically matched points, callosal connections also mediate integration of inputs from the contralateral hemisphere to achieve a higher order functional integration (e.g. higher orientation). This type of integration serves to strongly link and emphasize the information at the vertical meridian. This type of integration and accompanying overrepresentation of vertical is hypothesized to underlie perceptual enhancements at the vertical meridian (Coppola et al. 1998). As evidenced by studies in split brain patients, callosal function is a critical aspect of vision that needs to be addressed in VCPs. There are also studies that indicate callosal interactions are important for binocular vision (Watroba et al. 2001)

Now we have added this in the Discussion.

References:

Blakemore, C., Diao, Y. C., Pu, M. L., Wang, Y. K., & Xiao, Y. M. (1983). Possible functions of the interhemispheric connexions between visual cortical areas in the cat. *The Journal of Physiology*, 337(1), 331-349.

Coppola, D. M., White, L. E., Fitzpatrick, D., & Purves, D. (1998). Unequal representation of cardinal and oblique contours in ferret visual cortex. *Proceedings of the National Academy of Sciences*, 95(5), 2621-2623.

- Hunter, Z. R., Brysbaert, M., & Knecht, S. (2007). Foveal word reading requires interhemispheric communication. *Journal of Cognitive Neuroscience*, 19(8), 1373-1387.
- Innocenti, G. M., Aggoun-Zouaoui, D., & Lehmann, P. (1995). Cellular aspects of callosal connections and their development. *Neuropsychologia*, 33(8), 961-987.
- Kennedy, H., Dehay, C., & Bullier, J. (1986). Organization of the callosal connections of visual areas V1 and V2 in the macaque monkey. *Journal of Comparative Neurology*, 247(3), 398-415.
- Makarov, V. A., Schmidt, K. E., Castellanos, N. P., Lopez-Aguado, L., & Innocenti, G. M. (2008). Stimulus-dependent interaction between the visual areas 17 and 18 of the 2 hemispheres of the ferret (*Mustela putorius*). *Cerebral Cortex*, 18(8), 1951-1960.
- Payne, B. R. (1990). Function of the corpus callosum in the representation of the visual field in cat visual cortex. *Visual neuroscience*, 5(2), 205-211.
- Payne, B. R., & Siwek, D. F. (1991). The visual map in the corpus callosum of the cat. *Cerebral Cortex*, 1(2), 173-188.
- Watroba, L., Buser, P., & Milleret, C. (2001). Impairment of binocular vision in the adult cat induces plastic changes in the callosal cortical map. *European Journal of Neuroscience*, 14(6), 1021-1029.

R1-10: The authors state in their response, “Confirming that higher order activation can be evoked by multifiber stimulation is a key step in establishing the utility of this VCP approach on brainwide circuits. This is the aim of the study here” . While eliciting higher order activity is certainly a goal of VCPs, the goal is to stimulate visual cortex (or higher lever areas) in a way that propagates throughout the visual processing hierarchy without visual input, rather than modulating visual sensations evoked through photic stimulation of the retina.

Response:

Here we are talking about a broader concept of VCP, which refers not only to the triggering of cortical activity, but also to the modulation of ongoing cortical activity. The visual cortical prosthetics encompass a spectrum of activation strengths, paradigms, and technologies for the purpose of vision enhancement. Our demonstration that the optical array can induce a higher order effect is the first step towards developing methods to propagate signals through multiple levels of the hierarchy. We believe this work will contribute understanding that will eventually help the blind.

Reviewer #2 (Remarks to the Author):

The authors have addressed all of my concerns. I thank them for their hard work and I now think this manuscript will make a great addition to the literature!

Reviewer #3 (Remarks to the Author):

Thank you for your detailed responses.

R3-1: I guess you have chosen not to respond to my disapproval of the claims about advantages being over-blown and not proven?

Response:

We apologize for failing to respond to this important comment. We have now substantially revised the introduction. We have expanded and improved our description of previous studies,

removed phrases like “a new technological and conceptual approach”, removed words like “first”, “new”, and “novel”, and revised the purpose of this study “...conduct a proof of principle demonstration of a column-targeted approach and show that higher order representations can be predictably activated by columnar stimulation.”

However, the presented results are extremely preliminary and at best suggest that such things may be possible, but the system is far from practical: a few spots under a linear array of optical fibers (connected to large equipment) have been stimulated, and this stimulation shows a response in the opposite hemisphere.

Response:

We agree with the reviewer, our description was of the ultimate goal of a VCP. This study is the first step towards this goal. We have now revised this.

R3-2: R3-1) Optical powers. The calculation seems wrong as $1/f = 5$ ms for 200 Hz. However it seems to evaluate correctly. That is, if the area is the area of the fiber ($\pi \times 0.01$ cm \times 0.01cm), and the energy density of a pulse is 0.1 J, and the repetition rate is 200 pulses/s, then the average power at the tip of the fiber would be

$$0.1 \times \pi \times 0.01 \times 0.01 \times 200 = 2\pi \text{ mW.}$$

The reviewer’s calculation is correct.

The methods section is confusing in its incorrect use of power (average, peak), energy and intensity. E.G: The peak power of (a) laser pulse is 0.3 J/cm² or 0.7 J/cm². I would call this “the peak energy density of the pulse” , and at “at the tip of the fiber” .

The average power will be the Energy Density (J/cm²) x area (not size) of tip x pulse repetition frequency.

Response:

Thanks for your comment. We do need to distinguish the usage of ‘power, energy and intensity’. Energy (J)= power (W) * time (s). In manuscript, the unit we most commonly used is “J/cm²”, which is radiant exposure. This is the usage that has been established previously in the INS literature and which we followed (Wells et al. 2005, Izzo et al. 2007, Cayce et al. 2014, Richter, Jensen).

Our calculation function is:

$$\text{power} = \text{radiant exposure} * \text{area} / \text{time} = \text{radiant exposure} * \text{area} * \text{pulse repetition frequency.}$$
$$0.1\text{J/cm}^2 \rightarrow 0.1 * \pi * 10^{-4} * 200 = 2\pi \text{ mW} \approx 6.28 \text{ mW}$$

References:

Cayce, J. M., Friedman, R. M., Chen, G., Jansen, E. D., Mahadevan-Jansen, A., & Roe, A. W. (2014). Infrared neural stimulation of primary visual cortex in non-human primates. *Neuroimage*, 84, 181-190.

Izzo, A. D., Walsh, J. T., Jansen, E. D., Bendett, M., Webb, J., Ralph, H., & Richter, C. P. (2007). Optical parameter variability in laser nerve stimulation: a study of pulse duration, repetition rate, and wavelength. *IEEE Transactions on Biomedical Engineering*, 54(6), 1108-1114.

Wells, J., Kao, C., Mariappan, K., Albea, J., Jansen, E. D., Konrad, P., & Mahadevan-Jansen, A. (2005). Optical stimulation of neural tissue in vivo. *Optics letters*, 30(5), 504-506.

R3-3: Now, Fig #1.3 in the Response, shows a volume under the tip of approx. the fiber diameter x 0.1 cm (depth), which is $1e-5 \times \pi \text{ cm}^3$.

A single pulse has an energy $0.1 \times \pi \times 0.01 \times 0.01 = 3.14e-5$ J, which means that the average temperature rise in this volume will be $1/4.186$ degC, as the brain is mainly water. For $0.7\text{J}/\text{cm}^2$, it will be just under 2 degC. Of course, near the fiber's tip, the rise will be much greater, which is concerning. In your Response Figure #2-2, the rise is < 2 degC, but this is averaged power a voxel. It would be very useful to know the volume of a voxel.

Response:

Thanks for this question your concern as it is a very important point to address for this optical technique. Reviewer 3 is concerned that near the fiber tip, the temperature rise will be much greater and the tissue may be damaged.

Summary of previous relevant response: Just as a reminder, in our last response to Reviewer 2-2, we provided some relevant information along with references about the temperature rise question. Briefly summarized, the temperature rise to INS stimulation has been measured using MRI thermometry and modelling. Previous studies in peripheral nerve have shown the temperature rise, using 0.1 - 1.0 J/cm^2 , is estimated not to exceed $\sim 2^\circ\text{C}$ (e.g. 2.3°C from Thompson 2013). In addition, the damage threshold, measured using histological staining of tissue from rodent, monkey, and human cortex following application of the exact same paradigm of INS stimulation at different intensities, ranges from $0.6 - 1$ J/cm^2 . Some of this variability comes from tissue type (peripheral nerve, cortex), angle of the fiber optic to the tissue (the most intense focus of energy delivery is achieved when fiber is orthogonal to tissue surface), and duration of stimulation. Our own studies, using the same parameters as in the current study, provide a threshold of 0.6 J/cm^2 (rodent, monkey: Chernov et al. 2014, human: Pan et al. 2023). Such studies have helped to push forward clinical trials using INS (in cochlea: Claus Richter #NCT05110183; in peripheral nerve: Duco Jansen #NCT04601337).

Current response: **The temperature increases in this study are inferred from previous studies. Below, we provide additional information, including, as requested, the calculation of temperature rise from thermometry. We now incorporate additional information.**

We therefore conducted in-brain thermometry (Xi et al. under review, Quantifying tissue temperature changes induced by infrared neural stimulation: numerical simulation and MR thermometry) as well as simulations using a more precise model (Brown et al. 2020) than previous studies in peripheral nerve (Pennes bioheat equation, Pennes 1948). As Reviewer 3 noted, the calculated temperature depends on the spatial resolution of measurement. Our thermometry methods used **1mm isotropic voxels** (7T MRI coupled with a 5-channel surface coil to enhance SNR). Currently, ~ 1 mm isotropic is state of the art for thermometry from whole brain measurements (cf. $0.86 \times 0.86 \times 1 \text{mm}^3$ in thermometry study by Luo 2022 study in 9.4T MRI). These measures show that the temperature rise timecourses and amplitudes are consistent with, albeit a little lower than, the modelling studies (**compare thermometry in Fig R1A with modelling in R1B upper panel**). For the maximum energy of $0.7\text{J}/\text{cm}^2$ used in this study, the **temperature rise of a single 0.5sec pulse train, does not exceed $\sim 1.3^\circ\text{C}$ (by thermometry: Fig. R1A dotted cyan line) and $\sim 2.9^\circ\text{C}$ (by modelling: Fig R1A, upper panel dotted brown line)**. Our current data from monkeys (e.g. Yao 2023 J Comp Neurol), where we data are collected using values of 0.5 J/cm^2 or less, the temperature rise does not exceed **$\sim 1.0^\circ\text{C}$ (by thermometry: Fig. R1A dotted magenta line) and $\sim 2.1^\circ$ (by modelling in Fig. R1B upper panel dotted green line)**. These radiant exposures are values below the damage threshold (0.6 J/cm^2 , monkey: Chernov et al. 2014, human: Pan et al. 2023), determined using the same stimulation parameters as in this study.

The reason our INS paradigm is not damaging is because the pulses are very brief (5% duty cycle, 0.25msec every 5msec, cf. Thompson et al. 2013 Fig. 8) and the total pulse train short (0.5sec); note the rapid drop in temperature after last pulse train (**Fig. R1A, red arrow**). The transient aspect of the stimulation paradigm allows the heat to dissipate prior to the next pulse or pulse train resulting in a stable max temperature, even if pulse trains are continued (**Fig R1A and R1B**). Note that in this study, each fiber delivers only a single pulse train, resulting in temperature rises that are even lower (**asterisks in Fig. R1A**). The spacing between fibers is 500um, may result in greater temperature rise than a single fiber, but given the mean size of activation (manuscript Fig. 2G < 0.42mm²), it is likely to be minimal; we plan to directly assess this effect. Moreover, we have repeated confirmation that neuronal activity remains healthy and stable over periods of 1-2 years, using intensities of 0.1-0.5 J/cm² and hundreds of pulse train stimulations in a single session. Optical imaging and fMRI maps of INS stimulation in monkeys remain stable, there are no negative effects on animal health or behavior, and tissue at sites of stimulation, when it becomes available, appears histologically normal.

Fig. R1. Temperature rise due to INS stimulation (1875nm wavelength, 0.25msec pulse width, 200hz, 0.5sec, 200um fiber). (A) From MR thermometry in rat brain. (A) Evolution of the temperature change at peak single voxel at laser tip. Paired t-test: 3 sec vs 63 sec (legend, *0.05, **0.01, ***0.001). (B) Simulation of temperature increase using General BioHeat Transfer Model (Brown 2022. Model of increase, for a 200um fiber, over 0.5 sec (upper panel) and 150sec (lower panel) at 5 energies corresponding to 0.1.0.3.0.5.0.7.1.0 J/cm². See text.

Calculation of heat rise in thermometry. We have added this to the Supplemental Methods:

“In the thermometry study, we converted the direct proton resonance frequency phase shift data in acquired images to temperature change according to (Rieke and Pauly 2008, Luo et al. 2022):

$$\Delta T = \frac{\phi(T) - \phi(T_0)}{\gamma \alpha B_0 T E} \quad (1)$$

where $\phi(T)$ is the phase map at current time point, $\phi(T_0)$ is the phase map of the baseline image which is measured at room temperature (before INS), γ here represents the gyromagnetic ratio of hydrogen (2.67×10^8 rads/Tesla, constant), α is the PRF shift coefficient of water (-0.01 ppm/°C, constant), B_0 is the magnetic field strength (7 Tesla here), and TE is the echo time of imaging sequence (1.65 msec here). And in our condition, $1^\circ\text{C} \cong 0.031$ rads.

The temperature of the voxel with greatest phase shift at the location of the optic fiber tip was selected, and the temperature increase in the 9 isolated measurements was averaged for further analysis. ANOVA and paired-sample t-test were conducted to statistically analyze the temperature changes before and after INS, as well as the differences across different power intensities at the same time point. The data was analyzed using MATLAB.”

References:

Brown, W. G., Needham, K., Begeng, J. M., Thompson, A. C., Nayagam, B. A., Kameneva, T., & Stoddart, P. R. (2020). Thermal damage threshold of neurons during infrared stimulation. *Biomedical Optics Express*, 11(4), 2224-2234.

Chernov, M. M., Chen, G., & Roe, A. W. (2014). Histological assessment of thermal damage in the brain following infrared neural stimulation. *Brain Stimulation*, 7(3), 476-482.

Hindman, J. C. (1966). Proton resonance shift of water in the gas and liquid states. *The Journal of Chemical Physics*, 44(12), 4582-4592.

Luo H, Yang Z, Yang P-F, Wang F, Reed RL, Gore JC, Grissom WA, Chen LM (2022) Detection of laser-associated heating in the brain during simultaneous fMRI and optogenetic stimulation, *Magn Reson Med* 89(2):729-737.

Pan, L., Ping, A., Schriver, K. E., Roe, A. W., Zhu, J., & Xu, K. (2023). Infrared neural stimulation in human cerebral cortex. *Brain Stimulation*, 16(2), 418-430.

Pennes, H. H. (1948). Analysis of tissue and arterial blood temperatures in the resting human forearm. *Journal of applied physiology*, 1(2), 93-122.

Rieke, V., & Butts Pauly, K. (2008). MR thermometry. *Journal of Magnetic Resonance Imaging: An Official Journal of the International Society for Magnetic Resonance in Medicine*, 27(2), 376-390.

Thompson, A. C., Wade, S. A., Pawsey, N. C., & Stoddart, P. R. (2013). Infrared neural stimulation: influence of stimulation site spacing and repetition rates on heating. *IEEE Transactions on Biomedical Engineering*, 60(12), 3534-3541.

R3-4: Thank you for Response Figure #3-1. However, the y-axis should be power, not intensity.

Response:

Thanks for your comment, we have modified it.

Fig. R5. Power delivered by each of the 5 channels of fiber array measured by the power meter. (A) Average peak value per channel (\pm SD, the SD cross channels is 0.8). Radiant intensity was set to 15mW in this test. (B) Five-channel sequential (from ch1-5) power measurement (0.5s pulse train per channel, the cycle from ch1-5 repeating 20 times) through optical switch, which is the entire measurement result of (A). The graph in the top right corner: schematic diagram of 0.5s pulse trains (each contains 100 250us-pulses), the interval between adjacent pulse train is 0.5s. The numbers below indicate different channels.

R3-5: Response 3.9. I can see that you have provided Quantum Efficiency graphs up to 1100 nm, and you have assumed the QE falls at higher wavelengths. This is reasonable due to Einstein's theory of photodetection. I was just worried that there might be a thermal effect at longer wavelengths, which you could disprove with a simple experiment.

Response:

Thanks for your concern. The reviewer concerned that whether the out-of-band energy will influence the CCD camera to generate cross-talk. Note that the objective distance of our CCD camera is more than 3 centimeters. As suggested by the reviewer, we imaged INS stimulation of a piece of paper, Fig. R6 shows that there was no significant reflectance change in the field of view; thus, we infer that the CCD camera does not detect the 1870nm infrared light itself.

Fig. R6. INS stimulation on a piece of paper to check for cross-talk. There is no detectable signal. INS radiant exposure: 0.3J/cm². Scale bar: 5mm.

R3-6: In the MS Fig. 7 D, I see a periodic “up-down-up-down” fluctuation in the results. Something deterministic is happening here. Have you any idea what could be causing this systematic error?

Response:

Thanks for pointing out this question. We did not undertake heart rate synchronization in this experiment, after long-term imaging, dehydration of the agar may cause the cerebral cortex to not be firmly compressed by the glass slide, resulting in regular slight cortical pulsation caused by heart rate. Moreover, the number of repetitions performed in this run was insufficient to neutralize this source of biological noise, leading to the appearance of a \sim 2Hz oscillation. However, this oscillation does not impact the overall direction of signal change (enhancement, relative suppression) detected or the final result.

Reviewers' comments:

Reviewer #1 (Remarks to the Author):

The authors of the manuscript continue to make notable improvements, particularly to data presentation and have improved citation of relevant literature. However, fundamental issues still remain in the framing and contextualization of INS to electrical stimulation-based approaches to VCPs that need to be corrected by the authors. The authors still make counterfactual comments throughout that ignore prior columnar scale approaches that use electrical stimulation and continue to misrepresent the scale of visual percepts evoked with electrical stimulation. INS remains an interesting technique, but needs to be accurately compared to electrical stimulation, especially since the latter is the current standard.

Abstract:

"Cutting edge advances in electrical visual cortical prosthetics have evoked perception of shapes, motion, and letters in the blind. However, such prosthetics lack a detailed map of functional cortical columns to guide stimulation. Here, using pulsed infrared neural stimulation, we have developed an approach with the sumillimeter spatial precision needed for interfacing with cortical columns"

VCPs studies that have been conducted in humans lack detailed maps of visual cortex because the implanted population is blind and does not have natural vision to readily allow for feature mapping. This lack is not because of scale, especially since intracortical microelectrodes exist at this scale and have been tested for use in VCPs. Progress has been made with retinotopic atlases and further modifying the maps with resting state electrophysiology. How is this relevant to your manuscript? This manuscript is not about columnar scale mapping for VCPs.

I will once again point out, VCPs using electrical stimulation exist at the submillimeter scale. The authors of this manuscript need to contextualize INS relative to ICMS factually and without denying the scale is similar. Existing at a similar scale does not invalidate INS as a method and continually trying to claim that electrical stimulation does not exist at the "submillimeter" scale is just wrong.

VCPs based on the columnar organization of visual cortex is not conceptually new

Introduction

"However, none of these interfaces have taken advantage of known organization of functional columns in visual cortex."

As pointed out in my last review, many VCP approaches take into account cortical columns. Micro-electrode based approaches are typically framed as columnar scale or targeting the functional subunits of visual cortex. Revising the intro to cover more of the expansive history is good, but the authors still cannot claim to be the first and only researchers to take into account columnar structure. It's just not true.

Use of carnivores is an odd choice here

"technologies were not available to map the submillimeter-sized columns organization of visual cortex (in humans ~500 μm ^{5,6}; monkeys ~200-300 μm ⁷, cat ~500 μm ⁸). "

This not true. Micro electrodes map cortex at this scale. fMRI is not the only way to map visual cortex at the columnar scale

Results

The authors state "Such modulation would potentially make INS an important and useful tool to enhance cortical response under natural sensory and behavioral conditions." Why is this relevant to VCPs? VCP users will be blind and do not have "natural sensory" inputs.

Discussion

Columnar stimulation

Electrical stimulation is not always >1mm. Tehovnik 2006 and Tehovnik 2007 lay out very clearly the diameter of activation resulting from different currents relative to cortical columns, specifically in the context of developing VCPs. They use the equation $r = (I/K)^{1/2}$. As pointed out in previous reviews, depending on electrode type and scale, electrical stimulation can evoke a visual percept with currents as low as 20 μA , with the recent study from Fernandez 2020 reporting an average of 67 μA . Again, these average parameters lead to a diameter of activation of 0.45 - 0.63 mm. On the lower end, at 20 μA , this leads to a diameter of activation of 0.24 - 0.34 mm, well within the columnar scale. Representing electrical stimulation as >1mm compared to INS activating less than 0.5mm like this is wrong.

Tehovnik EJ, Slocum WM. Phosphene induction by microstimulation of macaque V1. *Brain Res Rev.* 2007 Feb;53(2):337-43. doi: 10.1016/j.brainresrev.2006.11.001. Epub 2006 Dec 14. PMID: 17173976; PMCID: PMC1850969.

Tehovnik EJ, Tolias AS, Sultan F, Slocum WM, Logothetis NK. Direct and indirect activation of cortical neurons by electrical microstimulation. *J Neurophysiol.* 2006 Aug;96(2):512-21. doi: 10.1152/jn.00126.2006. PMID: 16835359.

"Carnivore brain" is an odd word choice to make.

"While previous VCP studies have induced oriented and motion42 percepts from multisite stimulation21,24,43–46, we suggest that it is possible to achieve smaller (e.g. "

On what grounds? The percepts made by ICMS are smaller than 1 degree already.

Is this meant to be a heading? "Previous studies with electrical stimulation. Evidence"

The authors state in the discussion that the letters and patterns evoked by VCPs spanned ten degrees of visual space. While the Beauchamp 2020 study did show characters at this scale, the size of percepts, including characters, was on the scale of 1 degree in the Fernandez study.

The authors state: "In contrast to previous VCP designs, we aimed to target known functional columns in visual cortex and predicted the resulting activation based on known circuitry". Targeting known functional columns is not in contrast to previous VCP designs. Studies conducted with sighted cats, NHPs, and humans map features and retinotopy of tissue subtended by electrode arrays.

Reviewer #2 (Remarks to the Author):

The authors have addressed all of my concerns.

Reviewer #3 (Remarks to the Author):

Thank you for the responses and additional information. I am much happier with the manuscript.

My point about the voxels still stands: that the temperature near the fiber tip may be much higher than averaged over a 1-mm cube voxel, but this might be better addressed in another study (perhaps in

Reviewers' comments:

Reviewer #1 (Remarks to the Author):

The authors of the manuscript continue to make notable improvements, particularly to data presentation and have improved citation of relevant literature. However, fundamental issues still remain in the framing and contextualization of INS to electrical stimulation-based approaches to VCPs that need to be corrected by the authors. The authors still make counterfactual comments throughout that ignore prior columnar scale approaches that use electrical stimulation and continue to misrepresent the scale of visual percepts evoked with electrical stimulation. INS remains an interesting technique, but needs to be accurately compared to electrical stimulation, especially since the latter is the current standard.

The reviewer finds that our conceptualization of the electrical stimulation field has fundamental inaccuracies and does not provide the right context for the current study. We agree and apologize for our inadequate characterization of the field. We have now rewritten the abstract, introduction, and discussion to correct this. In addition to addressing the descriptions raised below, we explicitly describe, in both humans and monkeys, (1) the studies of electrical stimulation studies that have targeted cortical columns [“Developing visual cortical prosthetics (VCPs) for the blind (and those with low vision and other vision deficits) has long been a driving force behind vision research. From the days of Dobbelle¹, to current advanced studies in humans and monkeys, groundbreaking work has demonstrated that electrical stimulation delivered through multicontact arrays can elicit perception of shapes, motion, and letters^{2,3}. Advances in electrode technology and stimulation methods have brought significant quantitative and qualitative advances to VCPs. With low to moderate stimulation currents, Utah arrays are now able to evoke perception of fine shapes and features^{4,5}. Development of ‘current steering’ methods in humans demonstrated enhanced saliency of evoked percepts over static stimulation patterns⁶. Evidence from studies in monkeys show that focal low current electrical stimulation of cortex can modulate featurally specific percepts. These include single site stimulation of (1) direction domains in MT to modulate perception of the direction of a field of moving dots (Salzmann & Newsome⁷), (2) disparity tuned (near-to-far) sites in MT to bias depth perception (DeAngelis and colleagues⁸), and (3) presumed color sites in V1 to modulate the visibility of a small color spot in a background color matching task (Tehovnik⁹, in humans, see Schmidt et al¹⁰). Stimulation of face patches in temporal cortex led to non-specific distorted face percepts (non-specificity perhaps due to the higher electrical current range used 50-300 μ A)¹¹. Studies that have employed electrical^{12–14}, optogenetic^{15,16}, and infrared^{17,18} neural stimulation to stimulate single cortical columns lead to activation of local functionally matched orientation and color domain networks, further suggesting that single site stimulation impacts a larger network of functionally selective response. Together, these examples support the hypothesis that the lower stimulation intensities have minimal current spread, allowing selective stimulation of single cortical columns and their associated networks, which may lead to perceptual modulation related to the column-encoded feature. The columnar organization of visual cortex has long been known and provides a beautiful roadmap for accessing different

featural loci underlying perception. Here, the term ‘columnar’ is used to refer to the mesoscale functional domains revealed with electrophysiology¹⁹, optical imaging²⁰, and fMRI²¹. In visual cortex of humans, monkeys, and cats, different cortical areas have distinct spatial layouts representing, for example, features of color, contour orientation, motion, and depth. Previous VCP studies have stimulated single functional columns²² and have recognized the importance of incorporating cortical circuitry in VCP design. Now, with the development of ultrahigh field MRI and high resolution imaging methods, it becomes feasible to obtain columnar maps for targeted stimulation (in humans $\sim 500 \mu\text{m}$ ^{23–25}; monkeys $\sim 200\text{--}300 \mu\text{m}$ ²⁶, cat $\sim 500 \mu\text{m}$ ²⁷). Another important aspect of columnar organization is that, from area to area, information from ‘lower order’ columns are integrated in ‘higher order’ columns in downstream areas, thereby establishing de novo parameter spaces. A columnar VCP must therefore be capable of targeting desired sets of columns, and, ideally, with knowledge of the downstream effects. Here, we conduct a proof of principle demonstration of a column-targeted approach and show that higher order representations can be predictably activated by columnar stimulation.”], (2) the literature on single columns induction of percepts [“Evidence from previous studies using electrical stimulation suggest that selective stimulation of cortex evokes featurally specific percepts. While the specific parameters of different studies vary (intensity, frequency, duration), we summarize a few studies that potentially achieved column-specific stimulation by employing relatively low stimulation intensities. Salzmann & Newsome⁷ showed that a monkey’s direction discrimination of a moving dot field is biased towards the preferred direction of the electrically stimulated ($10\mu\text{A}$ in 1 s) column. Similarly, in the disparity axis, DeAngelis and colleagues⁸ showed that stimulating ($20\mu\text{A}$) patches of cells with selective disparity tuning in MT resulted in a perceptual bias towards the stimulated near-to-far column’s disparity preference. In a carefully conducted study in which visibility of a small color spot was eliminated by presenting a matching background color, the equivalent electrical stimulation ($15\text{--}55\mu\text{A}$) needed to achieve a similar effect was identified, illustrating that stimulation of V1 sites elicited unsaturated color percepts⁹. In humans, Schmidt et al¹⁰ showed that stimulating with low current levels near threshold (e.g. $1.9\mu\text{A}$) in ventral temporal cortex, percepts of color were often evoked and that increasing stimulation led to more washed out color percepts such as white, gray, or yellow (consistent with the blurring of signals across multiple functional domains). Stimulation of face patches in temporal cortex led to non-specific distorted face percepts, perhaps due to the higher electrical current range used ($50\text{--}300\mu\text{A}$)¹¹. Thus, together, these examples may support the hypothesis that stimulation of a single cortical column or patch can, with the lower stimulation intensities, lead to a perception related to the column-encoded feature. Further investigation is needed.”] and (3) on multi-site induction of global percepts [“Given the percepts induced by single site stimulation, what can be elicited by multisite stimulation? A few studies have examined the percepts evoked by multisite electrical stimulation, either sequentially^{1,2,5,6} or simultaneously^{4,5}. In human studies, surface electrode arrays have evoked striking percepts of shapes and letters in the blind. These percepts can be enhanced by a ‘dynamic current steering’ stimulation paradigm⁶, enhancing the salience of the percept and reducing the number of stimulation sites needed^{2,7,77}. In a study by Fernández et al.⁴, a blind subject implanted with a Utah array was able to discriminate evoked sizes and shapes as well as spatially small percepts, marking a significant advance in spatial resolution (see also Yoshor and Bosking³). Roelfsema’s group

implanted an astounding 1024-channel-count electrode interface (16 Utah arrays, 0.3-0.6M Ω , <100uA) into monkey V1 and V4, achieving percepts of shapes, letters, and motion direction⁵. We previously showed²⁸ that INS applied to visual cortex induces visual phosphenes, leading to predicted saccade behavior in the awake, behaving monkey. These findings indicate feasibility for INS to be used as a cortical stimulation method for VCPs. It remains to be seen whether optical stimulation approaches can achieve the groundbreaking capabilities of electrical VCPs.”]. We specifically describe the examples where stimulations are quite focal and induce fine percepts. We highlight all the places in the text in yellow which we have revised relevant to this critique.

Abstract:

"Cutting edge advances in electrical visual cortical prosthetics have evoked perception of shapes, motion, and letters in the blind. However, such prosthetics lack a detailed map of functional cortical columns to guide stimulation. Here, using pulsed infrared neural stimulation, we have developed an approach with the sumillimeter spatial precision needed for interfacing with cortical columns"

The abstract has been rewritten. The incorrect statement regarding lack of detailed maps has been removed.

“Cutting edge advances in electrical visual cortical prosthetics have evoked perception of shapes, motion, and letters in the blind. Here, we present an alternative optical approach using pulsed infrared neural stimulation. To interface with dense arrays of cortical columns with submillimeter spatial precision, both linear array and 100-fiber bundle array optical fiber interfaces were devised. We delivered infrared stimulation through these arrays in anesthetized cat visual cortex and monitored effects by optical imaging in contralateral visual cortex. INS modulation of response to ongoing visual oriented gratings produced enhanced responses in orientation-matched domains and reduced response in non-matched domains, consistent with a known higher order integration mediated by callosal inputs. Controls included dynamically applied speeds, directions and patterns of multipoint stimulation. This provides groundwork for a distinct type of prosthetic targeted to maps of visual cortical columns.”

VCPs studies that have been conducted in humans lack detailed maps of visual cortex because the implanted population is blind and does not have natural vision to readily allow for feature mapping. This lack is not because of scale, especially since intracortical microelectrodes exist at this scale and have been tested for use in VCPs. Progress has been made with retinotopic atlases and further modifying the maps with resting state electrophysiology. How is this relevant to your manuscript? This manuscript is not about columnar scale mapping for VCPs. We have removed these statements.

I will once again point out, VCPs using electrical stimulation exist at the submillimeter scale. The authors of this manuscript need to contextualize INS relative to ICMS factually and without denying the scale is similar. Existing at a similar scale does not invalidate INS as a method and continually trying to claim that electrical stimulation does not exist at the "submillimeter"

scale is just wrong.

We agree. We now describe at length the previous literature on columnar scale stimulation. See Discussion section '*Can optical approaches achieve the capabilities of electrical VCPs?*'.

VCPs based on the columnar organization of visual cortex is not conceptually new

We have removed this statement.

Introduction

"However, none of these interfaces have taken advantage of known organization of functional columns in visual cortex."

As pointed out in my last review, many VCP approaches take into account cortical columns. Micro-electrode based approaches are typically framed as columnar scale or targeting the functional subunits of visual cortex. Revising the intro to cover more of the expansive history is good, but the authors still cannot claim to be the first and only researchers to take into account columnar structure. It's just not true.

We have removed this statement.

Use of carnivores is an odd choice here

We have replaced this with 'cat' or 'ferret'.

"technologies were not available to map the submillimeter-sized columns organization of visual cortex (in humans $\sim 500 \mu\text{m}$ ^{5,6}; monkeys $\sim 200\text{-}300 \mu\text{m}$ ⁷, cat $\sim 500 \mu\text{m}$ ⁸). "

This not true. Micro electrodes map cortex at this scale. fMRI is not the only way to map visual cortex at the columnar scale

We have removed this statement.

Results

The authors state "Such modulation would potentially make INS an important and useful tool to enhance cortical response under natural sensory and behavioral conditions." Why is this relevant to VCPs? VCP users will be blind and do not have "natural sensory" inputs.

Response: In a broad context, VCP users can also include those with low vision and other vision deficits, as mentioned in paragraph 1 of the Introduction.

Discussion

Columnar stimulation

Electrical stimulation is not always $>1\text{mm}$. Tehovnik 2006 and Tehovnik 2007 lay out very clearly the diameter of activation resulting from different currents relative to cortical columns, specifically in the context of developing VCPs. They use the equation $r = (I/K)^{1/2}$. As pointed out in previous reviews, depending on electrode type and scale, electrical stimulation can evoke a visual percept with currents as low as $20\mu\text{A}$, with the recent study from Fernandez 2020 reporting an average of $67\mu\text{A}$. Again, these average parameters lead to a diameter of activation of $0.45 - 0.63 \text{ mm}$. On the lower end, at $20\mu\text{A}$, this leads to a diameter of activation of $0.24 - 0.34 \text{ mm}$, well within the columnar scale. Representing electrical stimulation as $>1\text{mm}$

compared to INS activating less than 0.5mm like this is wrong.

Tehovnik EJ, Slocum WM. Phosphene induction by microstimulation of macaque V1. *Brain Res Rev.* 2007 Feb;53(2):337-43. doi: 10.1016/j.brainresrev.2006.11.001. Epub 2006 Dec 14. PMID: 17173976; PMCID: PMC1850969.

Tehovnik EJ, Tolias AS, Sultan F, Slocum WM, Logothetis NK. Direct and indirect activation of cortical neurons by electrical microstimulation. *J Neurophysiol.* 2006 Aug;96(2):512-21. doi: 10.1152/jn.00126.2006. PMID: 16835359.

We agree and have now written a summary in Discussion of electrical stimulation studies that show focal stimulation. Tehovnik is included.

"Carnivore brain" is an odd word choice to make.

We have changed the "Carnivore" to "cat" or "ferret".

"While previous VCP studies have induced oriented and motion21,24,43–46, we suggest that it is possible to achieve smaller (e.g. "

On what grounds? The percepts made by ICMS are smaller than 1 degree already.

We have now removed this sentence.

Is this meant to be a heading? "Previous studies with electrical stimulation. Evidence"

We have now changed this heading to "Can optical approaches achieve the capabilities of electrical VCPs?"

The authors state in the discussion that the letters and patterns evoked by VCPs spanned ten degrees of visual space. While the Beauchamp 2020 study did show characters at this scale, the size of percepts, including characters, was on the scale of 1 degree in the Fernandez study. Although we stated that "surface electrodes" usually evoked these large-scale percepts, after this sentence, we also pointed out that "In a study by Fernández et al.⁴, a blind subject implanted with a Utah array was able to discriminate evoked sizes and shapes as well as spatially small percepts, marking a significant advance in spatial resolution (see also Yoshor and Bosking³)."

The authors state: "In contrast to previous VCP designs, we aimed to target known functional columns in visual cortex and predicted the resulting activation based on known circuitry". Targeting known functional columns is not in contrast to previous VCP designs. Studies conducted with sighted cats, NHPs, and humans map features and retinotopy of tissue subtended by electrode arrays.

Thanks for these comments. We have removed 'in contrast to VCP designs' and revised the statements about the aims of our design. "In this study, we aimed to target imaged functional columns in visual cortex and predicted the resulting activation based on known circuitry." "We aim to enhance or restore spatially and featurally specific visual percepts."

Reviewer #2 (Remarks to the Author):

The authors have addressed all of my concerns.

Thank you.

Reviewer #3 (Remarks to the Author):

Thank you for the responses and additional information. I am much happier with the manuscript.

My point about the voxels still stands: that the temperature near the fiber tip may be much higher than averaged over a 1-mm cube voxel, but this might be better addressed in another study (perhaps in-silicon?).

We appreciate this comment. As the reviewer states, this needs to be further examined.

There's also a mixture of sec and s for second. The SI unit is s (though Americans prefer sec). Please use s.

Response: Thanks for your comments. We now use 's' to refer to second.

REVIEWERS' COMMENTS

Reviewer #1 (Remarks to the Author):

The authors have done an excellent job revising their manuscript. The new experiments are a good addition and support their claims. INS is much better characterized with respect to other brain stimulation modalities, and the unique benefits of INS are better highlighted by this change.

This manuscript is ready for publication.